# Consistent patterns of distractor effects during decision making

**Bolton KH Chau**[1,2]*, **Chun-Kit Law**[1], **Alizée Lopez-Persem**[3,4],
**Miriam C Klein-Flügge**[3], **Matthew FS Rushworth**[3]

[1]Department of Rehabilitation Sciences, The Hong Kong Polytechnic University, Hong Kong, Hong Kong; [2]University Research Facility in Behavioral and Systems Neuroscience, The Hong Kong Polytechnic University, Hong Kong, Hong Kong; [3]Wellcome Centre for Integrative Neuroimaging (WIN), Department of Experimental Psychology, University of Oxford, Oxford, United Kingdom; [4]FrontLab, Paris Brain Institute (ICM), Inserm U 1127, CNRS UMR 7225, Sorbonne Université, Paris, France

**Abstract** The value of a third potential option or distractor can alter the way in which decisions are made between two other options. Two hypotheses have received empirical support: that a high value distractor improves the accuracy with which decisions between two other options are made and that it impairs accuracy. Recently, however, it has been argued that neither observation is replicable. Inspired by neuroimaging data showing that high value distractors have different impacts on prefrontal and parietal regions, we designed a dual route decision-making model that mimics the neural signals of these regions. Here we show in the dual route model and empirical data that both enhancement and impairment effects are robust phenomena but predominate in different parts of the decision space defined by the options' and the distractor's values. However, beyond these constraints, both effects co-exist under similar conditions. Moreover, both effects are robust and observable in six experiments.

*For correspondence:
boltonchau@gmail.com

**Competing interests:** The authors declare that no competing interests exist.

## Introduction

Independence of irrelevant alternatives is one of the assumptions of decision theory and behavioural economics: optimally, decisions between two options should be made in the same way regardless of whether or not a third option – a distractor – is also present. In practice, however, several lines of evidence suggest distractors impact on the neural mechanisms underlying choice representation and decision making and, as a result, subtly but significantly, alter the choices people and animals take (*Chau et al., 2014*; *Louie et al., 2015*; *Louie et al., 2011*; *Louie et al., 2013*; *Louie et al., 2014*; *Noonan et al., 2017*; *Noonan et al., 2010*).

Two forms of distractor effects have recently received attention. First, it has been reported that the relative accuracy of decisions made between two choosable options – the frequency with which the better option is chosen – decreases as the value of a third distractor increases. This has been interpreted as a consequence of *divisive normalisation* – the representation of any option's value is normalised by the sum of the values of all options present including distractors (*Chau et al., 2014*; *Louie et al., 2015*; *Louie et al., 2011*; *Louie et al., 2013*; *Louie et al., 2014*; *Noonan et al., 2017*; *Noonan et al., 2010*). The argument is bolstered by the observation that in many sensory systems, neural codes are adaptive and the rate of neural activity recorded in response to a given stimulus is related to the range of other stimuli encountered in the same context. For example, in the context of a bright light stimulus that leads to a high rate of neural responding in the visual system, the neural responses to two dimmer stimuli will be more similar to one another than they would otherwise be (*Carandini and Heeger, 2012*). As a result, discerning the brighter of the two dim stimuli

becomes more difficult. The contention is that a similar normalisation process occurs during value-guided decision making when a high value distractor is present.

However, it has also been found that in some circumstances the presence of a high value distractor can have a positive effect and lead to an increase in accuracy when deciding between two choosable options (*Chau et al., 2014*). This finding was explained by reference to a cortical attractor model in which decisions are made when choice representations in a neural network occupy a high firing attractor state (*Wang, 2002*; *Wang, 2008*; *Wong and Wang, 2006*). Competition between representations of choices in the network is mediated by a pool of inhibitory interneurons. In turn the activity levels of the inhibitory interneurons are driven by pools of recurrently interconnected excitatory neurons that each represents a possible choice. Increasing activity in the inhibitory interneuron pool mediating the comparison process improves decision accuracy although it may also slow decisions (*Jocham et al., 2012*). Because higher value distractors lead to more activity in the inhibitory interneuron pool, decisions between the choosable options are more accurate in the presence of high value distractors.

Although the explanation was framed in terms of the cortical attractor model, any model in which comparison mechanisms interact makes similar predictions. For example, if decisions are modelled by a drift diffusion process, then a similar prediction is made if the diffusion process proceeds for longer on trials when there is strong evidence for choosing the distractor instead. This is because initiating a response towards the distractor and then inhibiting it takes a finite amount of time. This allows additional time for the comparison between the choosable options to proceed concurrently and, hence, the better of the two choosable options is more likely to be taken. Other task manipulations that allow more time for the comparison process, for example by simply providing an opportunity for a second decision, after there has been time for more evidence accumulation, also make the 'correct' choice more likely to emerge from the comparison process and be chosen (*Resulaj et al., 2009*; *van den Berg et al., 2016*).

*Gluth et al., 2018*, however, have recently reported a series of experiments in which they claim there is no evidence of either divisive normalisation or positive distractor effects. Here, we therefore review the evidence for divisive normalisation and positive distractor effects. First, we explain how divisive normalisation and positive distractor effects are not 'opposing results' as is sometimes claimed (*Gluth et al., 2018*). We explain how it is possible for both effects to co-exist within the same data set but to predominate in different parts of the decision space. Second, we re-analyse *Gluth et al., 2018* data and show that both divisive normalisation and positive distractor effects are robustly and consistently present. Similarly re-analysis of other previously published data (*Chau et al., 2014*) again confirms both effects are present. In addition we report a new data set, collected at a third site, which again exhibits both effects.

Third, we investigate further the nature of the positive distractor and divisive normalisation effects by examining their manifestation in decisions in which participants make choices between options that lead to losses rather than gains. We find that the impact of larger distractor values flips from being facilitative (positive distractor effect) for gains to being disruptive (negative distractor effect) for losses while the divisive normalisation effect continues to manifest in the same direction (negative distractor effect) as originally shown for gains. This pattern of results suggests that divisive normalisation effects are truly related to the value of the distractor while the positive distractor effect is related to both the distractor's value and salience (its unsigned value – the size of its value regardless of whether it is positive or negative).

Finally, we consider a third consequence of the presence of a distractor; sometimes people choose the distractor itself even if this runs contrary to task instructions. Such choices have been termed attentional capture effects (*Gluth et al., 2018*). Again we show that such attentional capture effects exist in other data sets. However, we also show that the existence of attentional capture effects is not mutually exclusive with either positive distractor or divisive normalisation effects. In fact, whether or not attentional capture by the distractor occurs is itself subject to similar positive distractor and divisive normalisation effects. In a final experiment we use eye tracking data to demonstrate that in fact a relationship exists between the attentional capture effect and the positive distractor effect; positive distractor effects are particularly prominent after attentional capture by the distractor. This makes it possible to link the positive distractor effect to other situations in which the provision of a greater opportunity for evidence accumulation and comparison leads to more

accurate decision making, for example when allowing participants extra time to revise their initial decisions (*Resulaj et al., 2009*; *van den Berg et al., 2016*).

## Results

### Divisive normalisation of value and positive distractor effects should predominate in different parts of the decision space

In order to evaluate the evidence for divisive normalisation and positive distractor effects, it is necessary to realise that reports of each are not 'opposing results' (*Gluth et al., 2018*) that are mutually incompatible. Instead, quite the converse is the case: both effects can theoretically co-exist and do so in practice. This is because the impacts of divisive normalisation and of the positive distractor effect are more likely to be seen in different parts of the 'decision space' defined by the values of the higher value (HV) and lower value (LV) choosable options.

It is equally important to realise that small changes in the organisation of a neural network for making decisions could result in different distractor effects. To demonstrate this, first, we established a mutual inhibition model by simplifying a biophysically plausible model reported elsewhere that exhibits a positive distractor effect (*Chau et al., 2014*). In brief, it involves three pools of excitatory neurons that receive noisy input from the HV, LV and distractor (D) options that compete via a common pool of inhibitory neurons (*Figure 1a*). To visualise the impact of the distractor, we plotted in *Figure 1b* the model's choice accuracy as a function of relative distractor value (i.e. D-HV) and choice difficulty (HV-LV; as HV-LV becomes smaller it means that HV and LV have increasingly similar values and it is harder and harder to select the better option during the decision). When the distractor value is relatively large (left-to-right), the model makes more accurate choices (brighter colors). A similar trend is observed on both hard (*Figure 1c*, bottom) and easy trials (*Figure 1c*, top). In addition, we applied a simple general linear model (GLM) to analyse the simulated choice accuracy data (GLM1a). This GLM involves regressors that describe the difficulty (the difference between HV and LV: HV-LV), the relative value of the distractor (D-HV), and as well as an interaction term (HV-LV)(D-HV) that tests whether the distractor effect is modulated as a function of difficulty. Consistent with the pattern in *Figure 1b*, the results show that the model exhibits a positive D-HV effect ($\beta = 0.296$, $t_{104} = 231.858$, $p<10^{-142}$; *Figure 1d*). There was also a positive HV-LV effect ($\beta = 0.697$, $t_{104} = 663.224$, $p<10^{-189}$) and a positive (HV-LV)(D-HV) effect ($\beta = 0.094$, $t_{104} = 83.377$, $p<10^{-96}$).

It is possible to slightly adapt the mutual inhibition model to produce a divisive normalisation effect. In the divisive normalisation model, there are only two pools of excitatory neurons that are related to the HV and LV options (*Figure 1e*). The HV and LV inputs are normalised by the value sum of all options (HV+LV+D). Hence, instead of 'competing' directly with the HV and LV options, the distractor influences the model's evidence accumulation at the input level. In *Figure 1f and g*, the model shows poorer accuracy when D was relatively larger. When applying GLM1a, the results show that the model exhibits a negative D-HV effect ($\beta = -0.277$, $t_{104} = -229.713$, $p<10^{-141}$; *Figure 1h*). There were also a positive HV-LV effect ($\beta = 0.694$, $t_{104} = 598.596$, $p<10^{-185}$) and a negative (HV-LV)(D-HV) effect ($\beta = -0.094$, $t_{104} = -71.658$, $p<10^{-89}$).

Although the mutual inhibition and divisive normalisation models produce opposite distractor effects, the effects can co-exist with each predominating in different parts of decision space. To demonstrate this, we designed a dual route model, which is inspired by the fact that multiple brain structures have been identified with decision making. It is likely that they compete to select choices with one system predominating in some situations and another system in other situations. For example, in many situations, the intraparietal sulcus (IPS) carries decision-related signals (*Glimcher, 2002*; *Gold and Shadlen, 2007*; *Hanks et al., 2006*; *O'Shea et al., 2007*; *Platt and Glimcher, 1999*; *Shadlen and Kiani, 2013*; *Shadlen and Newsome, 1996*) and unlike decision-related signals elsewhere in the brain they remain present even when there is limited time in which to act (*Jocham et al., 2014*) or when decisions have become easy because of over-training (*Grol et al., 2006*). Moreover, divisive normalisation is present in activity recorded in IPS (*Chau et al., 2014*; *Louie et al., 2011*; *Louie et al., 2014*).

Another region with activity that is similarly decision-related is ventromedial prefrontal cortex (vmPFC) (*Boorman et al., 2009*; *Chau et al., 2014*; *De Martino et al., 2013*; *FitzGerald et al., 2009*; *Hunt et al., 2012*; *Lopez-Persem et al., 2016*; *Noonan et al., 2010*; *Papageorgiou et al.,*

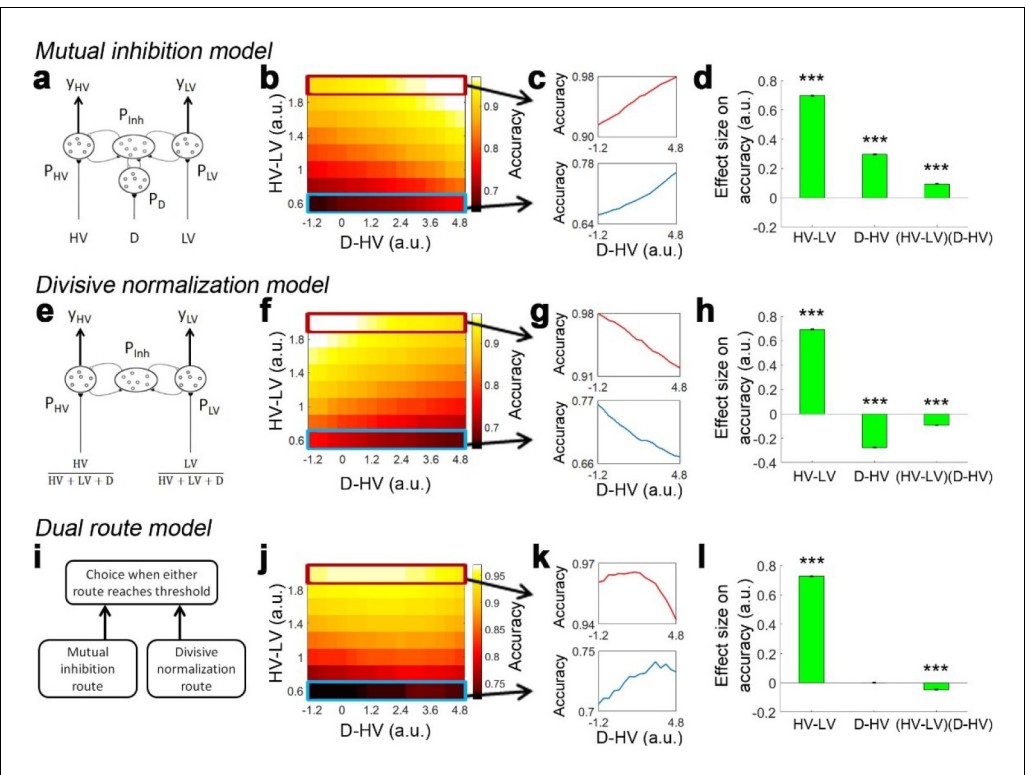

**Figure 1.** Distractor effects predicted by mutual inhibition model, divisive normalisation model, and a dual route model that combines both other models. (a) A mutual inhibition model involves three pools of excitatory neurons that receive excitatory input from the HV, LV or D options ($P_{HV}$, $P_{LV}$ and $P_D$). Concurrently, all excitatory neurons further excite a common pool of inhibitory neurons $P_{Inh}$, which in turn inhibit all excitatory neurons to the same extent. The HV or LV option is chosen once its accumulated evidence ($y_{HV}$ or $y_{LV}$ respectively) reaches a decision threshold. (b) The decision accuracy of the model is plotted across a decision space defined by the difficulty level (i.e. value difference between HV and LV) and the relative distractor value (D–HV). The model predicts a positive distractor effect – the decision accuracy increases (brighter colors) as a function of relative distractor value (left-to-right side). (c) A positive distractor effect is found on both hard (bottom) and easy (top) trials. (d) A GLM analysis shows that the model exhibits a positive HV-LV effect, a positive D-HV effect and a positive (HV-LV)(D–HV) effect. (e) Alternatively, a divisive normalisation model involves only two pools of excitatory neurons that receive input from either the HV or LV option. The input of each option is normalised by the value sum of all options (i.e. HV+LV+D), such that the distractor influences the model's evidence accumulation at the input level. (f) Unlike the mutual inhibition model, the divisive normalisation model predicts that larger distractor values (left-to-right side) will have a negative effect (darker colours) on decision accuracy. (g) A negative distractor effect is found on both hard (bottom) and easy (top) trials. (h) A GLM analysis shows that the model exhibits a positive HV-LV effect, a negative D-HV effect, and a negative (HV-LV)(D–HV) effect. (i) A dual route model involves evidence accumulation via mutual inhibition and divisive normalisation components independently. A choice is made by the model when one of the components accumulates evidence that reaches the decision threshold. (j) The current model predicts that on hard trials (bottom) larger distractor values (left-to-right side) will have a positive effect (brighter colors) on decision accuracy. In contrast, on easy trials (top) larger distractor values will have a negative effect (the colors change from white to yellow from left to right). (k) The opposite distractor effects are particularly obvious when focusing on the hardest (bottom) and easiest (top) trials. (l) A GLM analysis shows that the model exhibits a positive HV-LV effect, a positive D-HV effect and a negative (HV-LV)(D–HV) effect.

The online version of this article includes the following figure supplement(s) for figure 1:

**Figure supplement 1.** In the dual route model the positive D-HV effect on hard trials is mainly contributed by the mutual inhibition component and the negative D-HV effect on easy trials is mainly contributed by the divisive normalisation component.

**Figure supplement 2.** Reaction time (RT) of choices made by the dual route model.

*2017*; *Wunderlich et al., 2012*). However, in contrast to IPS, the impact of divisive normalisation in vmPFC is less prominent or absent (*Chau et al., 2014*), vmPFC activity diminishes with task practice (*Hunt et al., 2012*), and it is only engaged when more time is available to make the decision (*Jocham et al., 2014*). VmPFC lesions particularly disrupt the most difficult decisions but have less impact on easier ones (*Noonan et al., 2010*). The fact that lesions of vmPFC increase the impact of divisive normalisation in decision making (*Noonan et al., 2017*; *Noonan et al., 2010*) suggests divisive normalisation effects are mediated by a different region of the brain, such as IPS, and are perhaps even mitigated by the operation of vmPFC.

Based on these observations from vmPFC and IPS, the dual route model comprises both mutual inhibition and divisive normalisation models as components (*Figure 1i*). The evidence is accumulated independently in the two component models. A choice is made once one of the component models has accumulated sufficient evidence. Interestingly, when the model's decision accuracy is plotted in *Figure 1j*, the pattern looks very similar to the empirical data produced by human participants reported in Figure 2c of *Chau et al., 2014*. On hard trials (*Figure 1k*, bottom) the decision accuracy *increases* as a function of the relative distractor value, whereas on easy trials (*Figure 1k*, top) the decision accuracy *decreases* as a function of the relative distractor value. This pattern should be best captured in GLM1a by the (HV-LV)(D-HV) interaction term, as it reflects how the distractor effect changes as a function of the trial difficulty level (i.e. HV-LV). The results of GLM1a indeed revealed a significant negative (HV-LV)(D-HV) interaction effect on decision accuracy ($\beta = -0.047$, $t_{104} = -32.195$, $p<10^{-55}$). In addition, the distractor effect is slightly biphasic on hard trials and even more so on trials with intermediate difficulty levels – the accuracy increases from low to medium D-HV values and then decreases from medium to large D-HV values. This is due to the partial effects from the divisive normalisation route and mutual inhibition route. The biphasic pattern of the distractor effect is also reported in an alternative model (*Li et al., 2018*). Finally, there was a positive HV-LV effect ($\beta = 0.728$, $t_{104} = 526.591$, $p<10^{-179}$) and an absence of D-HV main effect ($\beta <0.001$, $t_{104} = 0.218$, $p=0.828$; *Figure 1i*).

In the dual route model positive and negative distractor effects predominate in different parts of the decision space. It is possible to understand the underlying reasons by analysing the choices made by the mutual inhibition and divisive normalisation components separately (*Figure 1—figure supplement 1*). On hard trials, when the distractor value becomes larger, the errors made by the mutual inhibition component decrease more rapidly in frequency than the increase in errors made by the divisive normalisation component, resulting in a net positive distractor effect. In contrast, on easy trials when the distractor value becomes larger the decrease in errors made by the mutual inhibition model is much less than the increase in errors made by the divisive normalisation model. *Figure 1—figure supplement 2* shows the reaction time of choices made by each component when the other component is switched off.

## Both divisive normalisation of value and positive distractor effects co-exist in data sets from three sites

These predictions are borne out by the available data from several laboratories using a multi-attribute decision making task (*Figure 2*; Materials and methods: multi-attribute decision-making task). First, as in Figure 2c of *Chau et al., 2014*, we visualised how the relative accuracy of decisions between HV and LV varied as a function of the HV-LV difference and the relative value of a distractor. Chau and colleagues manipulated distractor value with respect to HV because part of their investigation was concerned with comparing the HV-LV and HV-D signals present in neuroimaging data. However, the HV-D term has a negative relationship with the value of the distractor itself: D. This makes it less intuitive for understanding how the distractor value, D, influences choices. Here we present relative distractor value using the more intuitive D-HV term. Larger values of this term are correlated with larger values of the distractor, D.

*Figure 3a and c* show the data from the fMRI experiment (*Experiment 1 fMRI2014*; n = 21) reported by *Chau et al., 2014* and *Gluth et al., 2018* experiment 4 (*Experiment 2 Gluth4*; n = 44) respectively. It is important to consider these two experiments first because they employ an identical schedule. Specifically, Chau and colleagues reported both divisive normalisation effects and positive distractor effects, while Gluth and colleagues claimed they were unable to replicate these effects in their own data and when they analysed this data set from Chau and colleagues. Here we found that both data sets show a positive D-HV distractor effect. In both data sets, when decisions are difficult

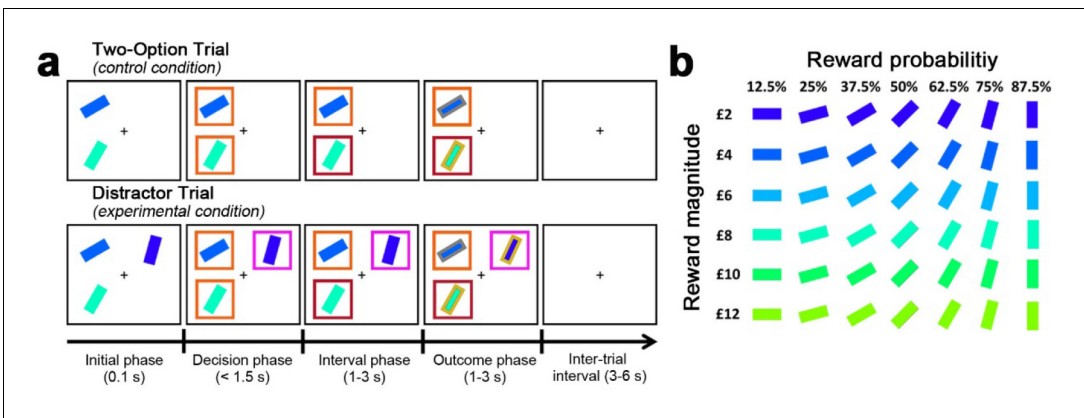

**Figure 2.** Behavioural task in Experiments 1–7. (**a**) The behavioural task was first described by *Chau et al., 2014* as follows. In the initial phase of two-option trials participants saw two stimuli indicating two choices. These were immediately surrounded by orange squares, indicating that either might be chosen. A subsequent color change in one box indicated which choice the participant took. In the outcome phase of the trial the outline color of the chosen stimulus indicated whether the reward had been won. The final reward allocated to the participant on leaving the experiment was calculated by averaging the outcome of all trials. Distractor trials unfolded in a similar way but, in the decision phase, one stimulus, the distractor, was surrounded by a purple square to indicate that it could not be chosen while the presentation of orange squares around the other options indicated that they were available to choose. (**b**) Prior to task performance participants learned that stimulus orientation and color indicated the probability and magnitude of rewards if the stimulus was chosen.

(HV-LV is small) then high value D-HV is associated with higher relative accuracy in choices between HV and LV; for example, the bottom rows of *Figure 3a and c* turn from black/dark red to yellow moving from left to right, indicating decisions are more accurate. However, when decisions were easy (HV-LV is large) then the effect is much less prominent or even reverses as would be predicted if divisive normalisation becomes more important in this part of the decision space. As in the predictions of the dual route model (*Figure 1j,k*), on easy trials although there was an overall decreasing trend in accuracy as a function of D-HV, there was an increasing trend at very low HV-LV levels. Overall, a combination of positive and negative D-HV effects on hard and easy trials respectively suggests that there should be a negative (HV-LV)(D-HV) interaction effect on choice accuracy.

When the behavioural data from human participants are analysed with the same GLM (GLM1a) that was used to analyse model data, the results are consistent with the illustrations in *Figure 1j and k*. In *Experiment 1 fMRI2014*, the results showed that the critical (HV-LV)(D-HV) effect was negative ($\beta = -0.243$, $t_{20} = -3.608$, p=0.002; *Figure 3b*). Just as in the dual route model, the negative (HV-LV)(D-HV) interaction term suggested that that the D-HV effect was particularly positive on hard trials where HV-LV was small. There was also a positive main effect of HV-LV ($\beta = 0.738$, $t_{20} = 8.339$, p<$10^{-7}$) and no main effect of D-HV ($\beta = 0.046$, $t_{20} = 0.701$, p=0.491). Similarly in *Experiment 2 Gluth4*, there was a negative (HV-LV)(D-HV) effect ($\beta = -0.068$, $t_{43} = -2.043$, p=0.047; *Figure 3d*). Interestingly, there was also a strong positive D-HV effect ($\beta = 0.122$, $t_{43} = 5.067$, p<$10^{-5}$), suggesting that even though the distractor effect varied as a function of difficulty level, the effect was generally positive in these participants. There was a positive HV-LV effect ($\beta = 0.571$, $t_{43} = 15.159$, p<$10^{-18}$).

In addition to being present in the data reported by *Chau et al., 2014* and *Gluth et al., 2018* the same effect emerged in a third previously unreported data set (*Experiment 3 Hong Kong*; n = 40) employing the same schedule but collected at a third site (Hong Kong). The results were highly comparable not only when the choice accuracy data were visualised using the same approach (*Figure 3e*), but also when the same GLM was applied to analyse the choice accuracy data (*Figure 3f*). There was a significant (HV-LV)(D-HV) effect ($\beta = -0.089$, $t_{39} = -2.242$, p=0.031). Again there was a positive D-HV effect ($\beta = 0.207$, $t_{39} = 5.980$, p<$10^{-6}$) and a positive HV-LV effect

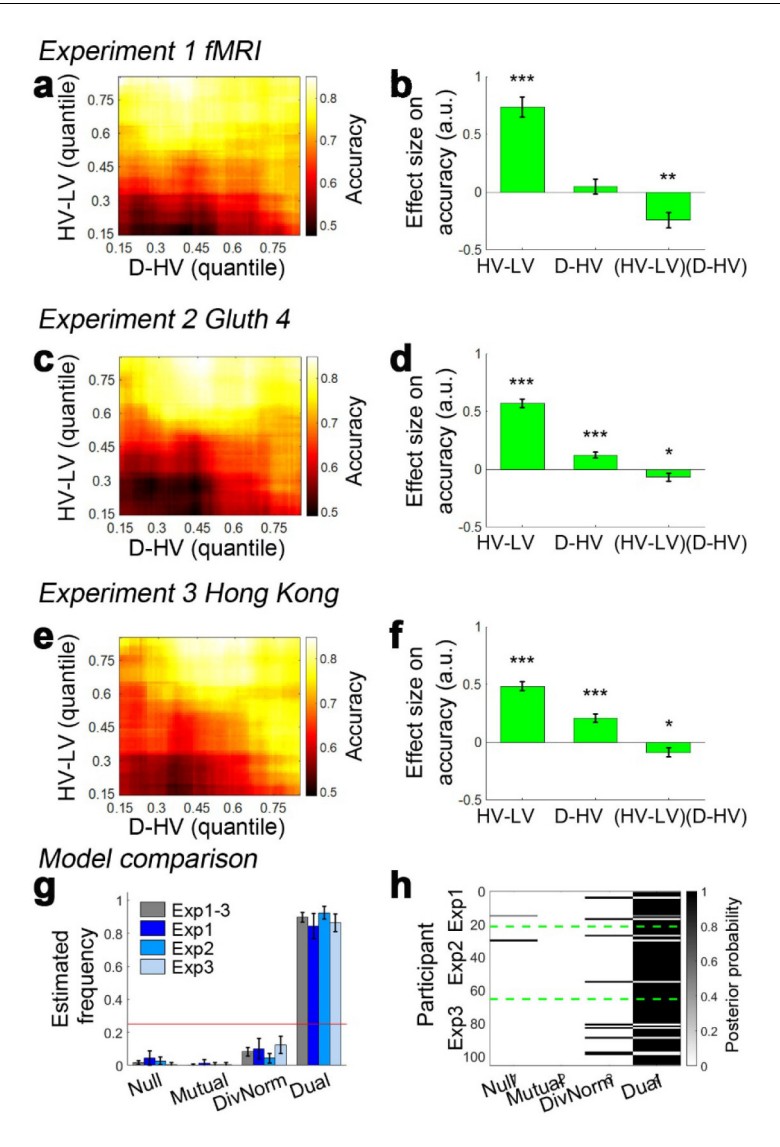

**Figure 3.** Decision accuracy across the decision space. Accuracy (light-yellow indicates high accuracy, dark-red indicates low accuracy) is plotted across the decision space defined by decision difficulty (HV-LV) and relative distractor value (D–HV) from (a) Experiment 1 fMRI2014, (c) Experiment 2 Gluth4, (e) Experiment 3 Hong Kong. In the case of each experiment, GLM analysis indicates that similar factors influence accuracy. The difference in value between the better and worse choosable option (HV-LV) is a major influence on accurately choosing the better option HV. However, accurate choice of HV is also more likely when the distractor is high in value (D-HV is high) and this effect is more apparent when the decision is difficult (negative interaction of (HV-LV)(D–HV)) in the data from (b) Experiment 1 fMRI2014, (d) Experiment 2 Gluth4, (f) Experiment 3 Hong Kong. (g) A model comparison shows that participants' behaviour in Experiments 1 to 3 is best described by the dual route model, as opposed to the null, mutual inhibition, or divisive normalisation models. (h) Posterior probability of each model in accounting for the behaviour of individual participants. Null: null model; Mutual: mutual inhibition model; DivNorm: divisive normalisation model; Dual: dual route model. *p<0.05, **p<0.01, ***p<0.001. (a–f) Error bars indicate standard error. (g–h) Error bars indicate standard deviation.

The online version of this article includes the following figure supplement(s) for figure 3:

**Figure supplement 1.** A similar (HV-LV)(D–HV) effect was observed when the HV+LV term was added to GLM1a which was done in GLM1b.

**Figure supplement 2.** The dual route model is better than the mutual inhibition, divisive normalisation and null models in predicting participants' accuracy and reaction time.

*Figure 3 continued on next page*

*Figure 3 continued*

**Figure supplement 3.** The dual route model provides the best account of participants' behaviour regardless of the parameterisation of non-decision time *Tnd* and inhibition level *f*.

($\beta$ = 0.485, $t_{39}$ = 12.448, p<10$^{-14}$). The pattern of results was consistent regardless of whether an additional HV+LV term was included in the GLM, as in *Chau et al., 2014*; a significant (HV-LV)(D-HV) effect was found in Experiments 1–3 when an additional HV+LV term was included in the GLM (GLM1b; *Figure 3—figure supplement 1*).

It is clear that data collected under the same conditions in 105 participants at all three sites are very similar and that a positive distractor effect consistently recurs when decisions are difficult. Next, we aggregated the data collected from the three sites and repeated the same GLM to confirm that the (HV-LV)(D-HV) interaction ($\beta$ = −0.101, $t_{104}$ = −4.366, p<10$^{-4}$), D-HV ($\beta$ = 0.223, $t_{104}$ = 6.400, p<10$^{-8}$) and HV-LV ($\beta$ = 0.529, $t_{104}$ = 20.775, p<10$^{-38}$) effects were all collectively significant. Additional control analyses suggest that these effects were unlikely due to any statistical artefact (see 'Distractor effects are not driven by statistical artefact' for details).

As in the empirical data from human participants, in the dual route model it is particularly obvious that there is a negative (HV-LV)(D-HV) interaction effect on the simulated choices, which is contributed by a combination of positive D-HV effect on hard trials and negative D-HV effect on easy trials. In contrast, in the mutual inhibition and divisive normalisation models only one of the two D-HV effects is present (*Figure 1*). To ascertain which model best describes the behaviour of the participants, each model was fitted to the empirical data, that is to each participant's choices and reaction time (RT) data. An additional *null model* that assumes that no distractor is present was also included in the model comparison. After fitting the models, a Bayesian model selection was performed to compare the goodness-of-fit of the models. Interestingly, the dual route model provided the best account of the participants' behaviour when *Experiments 1–3* were considered as a whole (estimated frequency Ef = 0.898; exceedance probability Xp = 1.000, *Figure 3g–h*) and when individual experiments were considered separately (*Experiment 1*: Ef = 0.843, Xp = 1.000; *Experiment 2*: Ef = 0.924, Xp = 1.000; *Experiment 3*: Ef = 0.864, Xp = 1.000). Furthermore, the fitted parameters were applied back to each model to predict participants' behaviour (*Figure 3—figure supplement 2*). The results show that the dual route model is better than the mutual inhibition, divisive normalisation, and null models in predicting both choice accuracy and reaction time.

Additional models were run to confirm that the dual route model is a better model. The above models involve assigning fixed values for the non-decision time *Tnd* (at 0.3 s) and inhibition level *f*. In one set of analysis the *f* is fitted as a free parameter (*Figure 3—figure supplement 3b*) and in another set of analysis both *Tnd* and *f* are fitted as free parameters (*Figure 3—figure supplement 3c*). In both cases, as in the models with fixed *Tnd* and *f*, the dual route model is a better fit compared to the other three alternative models (Ef = 0.641, Xp = 1.000 and Ef = 0.587, Xp = 1.000 respectively). Finally, a comparison of all twelve models (four models × three versions of free parameter set) shows that the dual route model with fixed *Tnd* and *f* is the best fit (Ef = 0.413, Xp = 1.000; *Figure 3—figure supplement 3d*).

The next step is to examine whether the (HV-LV)(D-HV) interaction effect from GLM1a and 1b arises because of the presence of a divisive normalisation effect (i.e. negative D-HV effect) on easy trials, a positive distractor effect on hard trials, or both effects. In other words, we need to establish which component of the interaction (or in other words, which main effect) is driving the interaction. To establish which is the case, the data were median split as a function of difficulty, defined as HV-LV, so that it is possible to easily visualise the separate predictions of the divisive normalisation and positive distractor accounts (*Figure 4a*; a similar approach was also used by Chau and colleagues in their Supplementary Figure SI.4). Then, to analyse each half of the choice accuracy data we applied GLM2a in a stepwise manner. *Step one* included the regressor HV-LV to partial out any choice variance shared between this term and the relative distractor term D-HV. Another regressor HV+LV was also included in the same step to completely partial out any remaining choice variance shared between the HV/LV options and D-HV. *Step two* then only included the regressor D-HV and was fitted on the residual choice variance of step one to determine the unique impact of the distractor. In the simulated choices of the dual route model, a positive distractor effect is found on hard trials; in

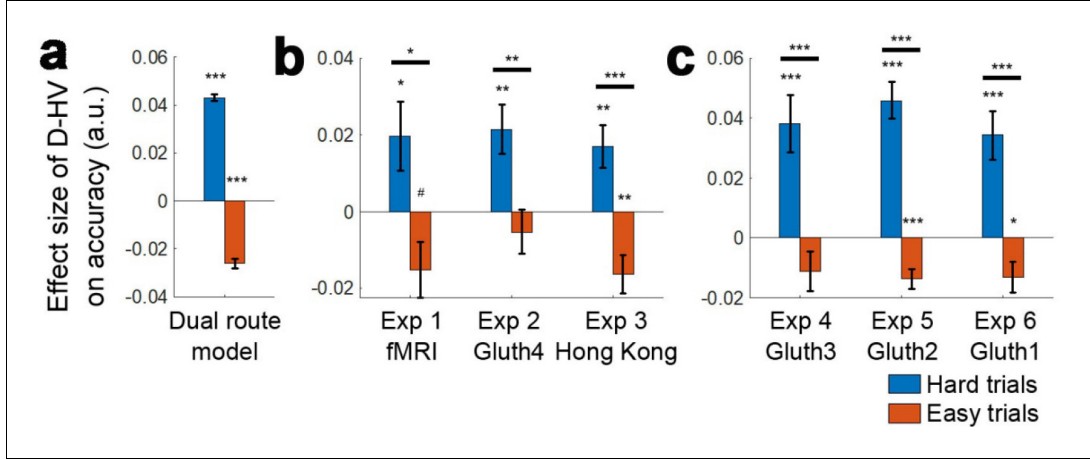

**Figure 4.** Distractors had opposite effects on decision accuracy as a function of difficulty in all experiments. The main effect of the distractor was different depending on decision difficulty. (**a**) In accordance with the predictions of the dual route model, high value distractors (D-HV is high) facilitated decision making when the decision was hard (blue bars), whereas there was a tendency for high value distractors to impair decision making when the decision was easy (red bars). Data are shown for (**b**) Experiment 1 fMRI2014, Experiment 2 Gluth4, Experiment 3 Hong Kong. (**c**) The same is true when data from the other experiments, Experiments 4–6 (i.e. Gluth1-3), are examined in a similar way. However, participants made decisions in these experiments in a different manner: they were less likely to integrate probability and magnitude features of the options in the optimal manner when making decisions and instead were more likely to choose on the basis of a weighted sum of the probability and magnitude components of the options. Thus, in Experiments 4–6 (i.e. Gluth1-3), the difficulty of a trial can be better described by the weighted sum of the magnitude and probability components associated with each option rather than the true objective value difference HV-LV. This may be because these experiments included additional 'decoy' trials that were particularly difficult and on which it was especially important to consider the individual attributes of the options rather than just their integrated expected value. Whatever the reason for the difference in behaviour, once an appropriate difficulty metric is constructed for these participants, the pattern of results is the same as in panel a. # p<0.1, *p<0.05, **p<0.01, ***p<0.001. Error bars indicate standard error.

The online version of this article includes the following figure supplement(s) for figure 4:

**Figure supplement 1.** Similar difficulty-dependent distractor effects were observed when accuracy is explained using the value of D, rather than the relative value of D in comparison to HV (i.e. D–HV).

**Figure supplement 2.** The dual-route model best describes participant behaviour in Experiments 1–6.

other words, high D-HV value is associated with significantly greater accuracy on hard trials (*Figure 4a*, blue bar; $\beta = 0.008$, $t_{104} = 32.173$, p<$10^{-55}$). The same model also exhibited a negative distractor effect, which is predicted by the divisive normalisation account, on easy trials (*Figure 4a*, red bar; $\beta = -0.003$, $t_{104} = -17.105$, p<$10^{-31}$). A very similar pattern was also found in the empirical data across all three experiments. There was a positive distractor effect on hard trials (*Figure 4b*, blue bars – *Experiment 1 fMRI2014*: $\beta = 0.020$, $t_{20} = 2.173$, p=0.042; *Experiment 2 Gluth4*: $\beta = 0.021$, $t_{43} = 3.366$, p=0.002; *Experiment 3 Hong Kong*: $\beta = 0.017$, $t_{39} = 3.081$, p=0.004). Moreover, in *Experiment 1 fMRI2014* high value D-HV was marginally associated with less accuracy on easy trials ($\beta = -0.015$, $t_{20} = -2.080$, p=0.051; *Figure 4b* red bars) and in *Experiment 3 Hong Kong* it was significantly associated with less accuracy on easy trials ($\beta = -0.016$, $t_{39} = -3.339$, p=0.002). Although the effect was not significant in *Experiment 2 Gluth4*, a similar negative trend was found ($\beta = -0.005$, $t_{43} = -0.914$, p=0.366), which at least supports the notion that the D-HV effect became less positive from hard to easy trials in this particular experiment. The D-HV effects were significantly different between hard and easy trials across all three experiments (*Experiment 1 fMRI2014*: $t_{20}$=2.706, p=0.014; *Experiment 2 Gluth4*: $t_{43}$=3.001, p=0.005; *Experiment 3 Hong Kong*: $t_{39}$=4.847, p<$10^{-4}$). Similar results were found if we tested the effect of the absolute value of D instead of the relative D-HV effect (GLM2b; *Figure 4—figure supplement 1a*). In summary, a positive distractor effect is present when decisions are difficult in all three data sets. Divisive normalisation is apparent on easier trials, at least in two of the three data sets.

It might be asked why the presence of distractor effects in their data was not noted by *Gluth et al., 2018*. The answer is likely to be complex. A fundamental consideration is that it is important to examine the possibility that both distractor effects exist rather than just the possibility that one or other effect exists. This means that it is necessary to consider not just the main effect of D-HV but also D-HV interaction effects. Gluth and colleagues, however, focus on the main effect of D-HV in most sections of their paper, apart from their table S2. Careful scrutiny of their table S2 reveals that the (D-HV)(HV-LV) interaction is reliably significant in their data. A further consideration concerns the precise way in which the impact of the distractor D is indexed in the GLM particularly on control trials where no distractor is actually presented. *Gluth et al., 2018* recommend that a notional value of D is assigned to control trials which corresponds to the distractor's average value when it appears on distractor trials. In addition, they emphasise that HV-LV and D-HV should be normalised (i.e. demeaned and divided by the standard deviation) before calculating the (HV-LV)(D-HV) term. If we run an analysis of their data in this way then we obtain similar results to those described by Gluth and colleagues in their Table S2 (*Supplementary file 1A* here). Although a D-HV main effect was absent, the (HV-LV)(D-HV) interaction term was significant when data from all their experiments are considered together. While Gluth and colleagues omitted any analysis of the data from Experiment 1 fMRI, we have performed this analysis and once again a significant (HV-LV)(D-HV) effect is clear (*Supplementary file 1A*).

Another possible reason for the discrepancy in the interpretation concerns the other three experiments reported by Gluth and colleagues. We turn to these next.

## Examining distractor effects in further experiments

We can also examine the impact of distractors in three further experiments reported by Gluth and colleagues. Below, we show that essentially the same pattern of results emerges in these experiments (*Figure 4c*): *Experiment 4 Gluth3; Experiment 5 Gluth2; Experiment 6 Gluth1*. Before we examine the data in detail, however, it is worth noting some differences between the experiments. First, the way in which participants made decisions in these next experiments was different. In the first three experiments participants tended to combine the information that the choice stimuli provided about both reward probability and reward magnitude; their choices indicated that they tended to choose the option where the product of reward probability and reward magnitude was larger than that of the alternative. By contrast in the next three experiments participants were still attracted by large reward probability options and large reward magnitude options but they did not always integrate reward magnitude and reward probability information to choose the option with the larger overall value. This is apparent when two simple GLMs (GLM3a and GLM3b) were used to describe the accuracy of each participant's decision making. GLM3a involves two terms that relate to the expected values of the HV and LV option. The first term is, as previously, the HV-LV value difference term. The second term is an HV+LV term which captures the remaining variance associated with the HV and LV expected values. GLM3b, however, included four separate attribute-based terms. It involves two terms that describe the difference in reward magnitude and the difference in reward probability of the HV and LV options. It also involves two terms that describe the sum in reward magnitude and sum in probability of the two options, which capture the remaining variance of the attributes of the two options. The Bayesian Information Criterion (BIC) value was significantly smaller in the attribute-based GLM3b than the value-based GLM3a in all Gluth's experiments: that is *Experiments 2* ($t_{43}$ = 3.540, p<0.001), *4* ($t_{22}$ = 1.942, p=0.065; where the difference was marginal), *5* ($t_{48}$ = 5.616, p<$10^{-5}$) and *6* ($t_{30}$ = 3.635, p=0.004). This was not the case for the experiments conducted elsewhere: *Experiments 1* ($t_{20}$ = 1.067, p=0.299) and *3* ($t_{39}$ = 1.311, p=0.198).

In summary, in participants from *Experiments 2* and *4–6* (i.e. *Gluth1-4*), the difficulty of a trial could be better described by the weighted sum of the magnitude and probability components associated with each option rather than the true objective value difference HV-LV. It is not clear why Gluth and colleagues' participants performed the task in this way but we know that people and animals often make decisions in a similar manner in other experiments (*Farashahi et al., 2019*; *Scholl et al., 2014*; *Scholl et al., 2015*). Indeed, while the behaviour of the participants in experiments 1 and 3 is not explained better by an attribute-based model than by the more normative model employing values based on the integration of both magnitude and probability, the integrative model is not a significantly better one. This suggests that they may have a tendency to use attribute-based heuristics. One way of interpreting such behaviour is that participants are not acting as if they

estimate the HV and LV values from their magnitude and probability components in the optimal multiplicative manner but instead they are acting as if using an additive heuristic. Further consideration of the details of *Experiments 4–6* suggest possible reasons why participants might have been particularly prone to behave in this way in these experiments. The stimuli used in all six experiments were two dimensional in nature; stimulus color and orientation respectively indicated the expected magnitude and probability of the reward that participants would receive for taking a choice. This approach was taken by Chau and colleagues in their initial experiments because it was conjectured that positive distractor effects might be linked to vmPFC/mOFC and it is known that vmPFC/mOFC plays an important role when decisions are made between multi-attribute options (*Fellows, 2006*; *Hunt et al., 2012*; *Papageorgiou et al., 2017*). However, because *Gluth et al., 2018* were interested in the possibility of 'decoy' effects they included a number of 'novel' trials in *Experiments 4–6* in addition to those that were used in the initial experiment by Chau and colleagues. Some decoy effects occur when decisions are difficult (HV and LV are close together in value) but the value associated with one of the components of the distractor is close to the value of one of the components associated with either HV or LV. They therefore included additional trials that were particularly difficult and on which it was especially important to consider the individual attributes of the options rather than just their integrated expected value. These additional trials accounted for 27% of all trials in *Experiments 4–6* and it is possible that this caused the participants to use an attribute-based approach when making their choices.

When we median split the data from *Experiments 4–6* as a function of difficulty described by HV-LV, we were not able to find any distractor effects on both hard ($|t| < 1.393$, p>0.178) and easy trials ($|t| < 1.072$, p>0.295). However, it is not so surprising that such an analysis was unable to reveal any difficulty-dependent distractor effect because for these participants in these experiments, difficulty is a function of the individual attributes rather than the correctly integrated expected value of each option. Thus, we extracted the weighting of each individual attribute on choice accuracy estimated in GLM3b above and calculated the difference in weighting between HV and LV for each component attribute. We then estimated the sum of the weighted probability difference and weighted magnitude difference and used this to estimate the subjective difference in the values of HV and LV and used this as an index of difficulty.

Interestingly, when we median split the trials according to this attribute-based difficulty index and applied GLM2c, which only differs from GLM2a by how difficulty is defined, the results of *Experiments 4–6* were then highly comparable to those found in *Experiments 1–3*. On hard trials, a positive D-HV effect was found in *Experiments 4* ($\beta = 0.038$, $t_{22} = 3.958$, p<0.001), *5* ($\beta = 0.046$, $t_{48} = 7.403$, p<$10^{-8}$) and *6* ($\beta = 0.0343$, $t_{30} = 4.278$, p<0.001; *Figure 4c*). On easy trials, a negative D-HV effect was found in *Experiment 5* ($\beta = -0.014$, $t_{45} = -3.997$, p<0.001) and *Experiment 6* ($\beta = -0.013$, $t_{30} = -2.590$, p=0.015), while a similar trend was found in *Experiment 4* ($\beta = -0.011$, $t_{22} = -1.684$, p=0.106). We note that it was difficult to fit models to the data in three of the 149 participants (all were participants in *Experiment 5*) because their choices were all accurate on hard trials and so their data were omitted from the analysis. The D-HV effect was significantly different between hard and easy trials across all three experiments ($t > 4.428$, p<0.001). Again, a similar pattern of results emerged when the analyses employed the absolute value of D rather than the relative D-HV term (*Figure 4—figure supplement 1b*).

We realised that it is possible to observe similar difficulty-dependent distractor effects in all six experiments by analysing all data with one single approach. This is consistent with the fact that, even in *Experiments 1* and *3*, participants do not integrate probability and magnitude in the normatively optimal way but have some bias towards attribute-based heuristics even if it is not as strong as in *Experiments 2*, and *4–6*. We excluded all 'novel' trials in *Experiments 2, 4–6*, added by Gluth and colleagues for testing decoy effects and applied a simple regression GLM2c (see *Figure 4—figure supplement 1c* for details). We found that there was a significant difference in the D effect between hard and easy trials in each of the six experiments ($t > 2.220$, p<0.034). The same is also true when we combined the data from all six experiments ($t_{207} = 7.679$, p<$10^{-12}$). Finally, there was a positive distractor effect on hard trials ($\beta = 0.026$, $t_{207} = 6.080$, p<$10^{-7}$) and a negative distractor effect on easy trials ($\beta = -0.017$, $t_{207} = -4.732$, p<$10^{-5}$) when all six experiments were considered together.

Finally, the four models (dual-route, mutual inhibition, divisive normalisation and null) were applied to fit the data of *Experiments 4–6*. Again, the dual-route model provided the best account of participants' behaviour when individual experiments were considered separately (*Experiment 4*:

Ef = 0.806, Xp = 0.999; *Experiment 5*: Ef = 0.649, Xp = 1.000; *Experiment 6*: Ef = 0.946, Xp = 1.000; *Figure 4—figure supplement 2*) or when *Experiments 1–6* were considered as a whole (Ef = 0.846, Xp = 1.000).

## Distractor effects are not driven by statistical artefact

Several considerations suggest that the presence of the difficulty-dependent distractor effect is not due to some unusual statistical artefact. First, all of the interaction terms are calculated after their component parts, the difficulty and distractor terms, are *z*-scored (i.e. centered by the mean and then divided by the standard deviation). Second, the interaction effects were further confirmed by median splitting the data by difficulty level and testing the distractor effect on each half of the trials. The finding of opposite distractor effects on hard and easy trials when analysed separately is a key characteristic of true interaction effects. Additional simulations and control analyses also confirmed that the difficulty-dependent distractor effect was not due to any statistical artefact (Appendices 1 and 2).

## A more complete analysis of distractor effects

The dual route model predicts that the distractor effect varies to a degree as a function of difficulty (HV-LV; *Figure 1j,k*). Other factors also mean that different types of distractor effects should be seen in different parts of the decision space. Even, in isolation, the divisive normalisation model predicts that the distractor effect varies strongly as a function of another term, the value sum: HV+LV. Since the overall normalisation effect depends on the total integrated value of the options (HV+LV +D), variance in this term mainly reflects variance in D when HV+LV is small. Thus, the dual route model predicts that negative distractor effect driven by the divisive normalisation component should become weaker when the value sum HV+LV is large and positive distractor effect driven by the mutual inhibition component should become stronger. This is exactly what has been shown in further analyses of the simulated data of the dual route model and the empirical data of the actual participants (Appendix 3).

## Attentional capture distractor effects and further evidence for positive distractor effects and divisive normalisation

So far we have established that both positive and negative distractor effects on choice accuracy (i.e. choosing HV over LV) were each most robust in different parts of the decision space. Despite the fact that in all experiments participants were instructed not to choose the distractor, it was still chosen by mistake on some trials. These trials were excluded in all the analyses presented above. However, Gluth and colleagues emphasised that the distractor has an *attentional-capture* effect that impaired choice accuracy. As such, they found that when the distractor value was large, participants made more mistakes by choosing the distractor itself more often.

We consider that the *attentional-capture* effect and the difficulty-dependent distractor effects that we have described above (i.e. both the positive and negative distractor effects) should not be thought of as mutually exclusive. To illustrate this, we ran an analysis that also included trials where the distractor was chosen and then applied an analysis similar to GLM1a again. Unlike Gluth and colleagues who ran a binomial logistic regression to test whether HV or collectively LV and D were chosen, here in GLM1c we ran a multinomial logistic regression. In this analysis, we were able to test the effects of distractor value on the choice ratio between HV and LV and on the choice ratio between HV and D separately. This means that a single analysis has the power to reveal both attentional capture effects as well as other distractor effects. We also only included participants with at least three trials on which the distractor was chosen (n = 180). We removed another six participants that showed exceptionally large beta weights ($|\beta|>10$) for the constant term on the HV/D choice ratio (remaining participants: n = 174), although the removal of these participants did not change the pattern of the results. First, we again replicated the finding that there was a negative (HV-LV)(D-HV) effect ($\beta = -0.076$, $t_{173} = -4.044$, $p<10^{-4}$) and also a positive HV-LV effect ($\beta = -0.530$, $t_{173} = -23.953$, $p<10^{-56}$) on the choice ratio between HV and LV (i.e. choice accuracy; *Figure 5a*). In addition, as the attentional capture model suggested, there was also a positive HV-D effect on the choice ratio between HV and D ($\beta = 1.176$, $t_{173} = 18.996$, $p<10^{-43}$; *Figure 5b*) – the HV was chosen more often when its own value was large and the same was true for the distractor when D value was large.

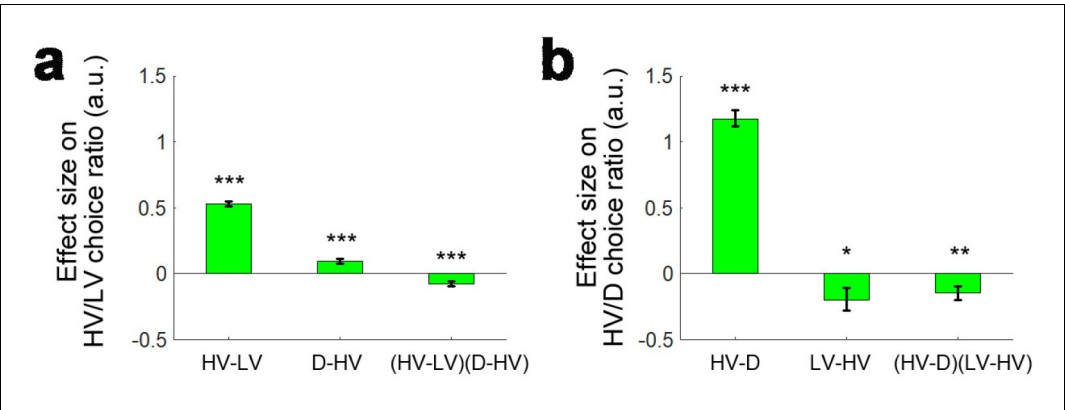

**Figure 5.** A further replication of distractor effects in 174 participants. On some occasions 'attentional capture' occurs and participants, contrary to their instructions choose the distractor itself. It is possible to analyse how participants distribute their choices between the choosable options HV and LV and between the high value choosable option HV and the distractor D using multinomial logistic regression. (a) Multinomial regression confirms that HV is chosen over LV if the HV-LV difference is large but also confirms that there is an interaction between this term and the difference in value between D and HV. (b) There are also, however, parallel effects when decisions between HV and D are now considered. In parallel to the main effect seen in decisions between HV and LV, decisions between HV and D are mainly driven by the difference in value between these two chooseable options – the HV-D difference. In parallel to the distractor effect seen when decisions are made between HV and LV, there is also an effect of the third option, now LV, when decisions between HV and D are considered; there is an (HV-D)(LV-HV) interaction. *p<0.05, **p<0.01, ***p<0.001. Error bars indicate standard error.

Interestingly, in the same analysis we found additional evidence of both difficulty-dependent distractor effects and the attentional capture distractor effect. This is because in each of the six experiments we have considered so far there is effectively a second data set that co-exists somewhat like a palimpsest alongside the main one. Normally, the D option is regarded as a distractor when the HV/LV choice ratio (i.e. choice accuracy) is considered. However, analogously in the multinomial analysis the LV option can be regarded as a 'distractor' when we consider the HV/D choice ratio. Hence, we can also test the difficulty-dependent 'distractor' effect of LV on the HV/D choice ratio using the (HV-LV)(D-HV) term. It may be helpful to point out that the analogous term for testing the LV distractor effect on the HV/D choice ratio is perhaps most obviously (HV-D)(LV-HV) but this is mathematically identical to (HV-LV)(D-HV). Critically, we found a negative (HV-D)(LV-HV) effect on the HV/D choice ratio ($\beta$ = −0.142, $t_{173}$ = −2.674, p=0.008; *Figure 5b*), suggesting that on hard trials (HV-D is the relevant metric of difficulty when considering the HV/D choice ratio; a small HV-D difference means that it is difficult to select the HV option over the D option) more HV choices were chosen over D when the value of the 'distracting' LV option was large. Ideally, a follow-up analysis of the negative (HV-D)(LV-HV) effect should also be run to examine how the distractor LV-HV effect varied from hard (small HV-D) to easy (large HV-D) trials. Since choices of the D option were rare, splitting the trials further according to the median HV-D index is obviously likely to result in even smaller numbers of D choices in each half of the data, especially in the half with large HV-D. This, in turn, is likely to lead to unreliable estimates of effect sizes. However, we still attempted to perform this analysis (which is analogous to those performed in *Figure 4*). In particular, we applied a GLM that involved regressors HV-D, HV+D and LV-HV and focused on the LV-HV distractor effect in each half of the data. The results showed that on hard (small HV-D) trials there was a positive distractor LV-HV effect on the HV/D choice ratio ($\beta$ = 0.170, $t_{171}$ = −2.517, p=0.013). Another two participants were excluded in this analysis due to exceptionally large beta weights (|$\beta$|>10). Next, we repeated the same analysis on easy (large HV-D) trials. As expected, a large number of participants had to be excluded in this analysis – 107 participants did not choose the D option on these trials and 22 participants showed exceptionally large beta weights (|$\beta$|>10). Nevertheless, on easy (large HV-D) trials in the remaining 45 participants there was a lack of distractor LV-HV effect ($\beta$ = 0.017, $t_{44}$ = 1.407, p=0.936). Although it is expected that the distractor LV-HV effect should be negative on easy trials,

it is possible that this is due to the scarcity of D choices. Finally, there was a significant difference in LV-HV effect between the hard and easy trials in these participants ($t_{44}$ = 6.126, p<$10^{-6}$).

Taken together, the multinomial logistic regression provided evidence supporting both difficulty-dependent distractor effects and attentional capture distractor effects both when we consider the HV/LV choice ratio that has been the focus of previous studies but also when we consider the HV/D choice ratio that is often simultaneously present in these studies due to erroneous choices of the distractor.

## Experiment 7: Loss experiment

The attentional capture model raises the question of whether any distractor effect on choice accuracy is due to the value or the salience of the distractor. This is difficult to test in most reward-based decision making experiments because the value and salience of an option are often collinear – more rewarding options are both larger in value and more salient – and it is not possible to determine which factor drives behaviour. One way of breaking the collinearity between value and salience is to introduce options that lead to loss (*Kahnt et al., 2014*). As such, the smallest value options that lead to great loss are very salient (*Figure 6a*, bottom), the medium value options that lead to small gains or losses are not salient and the largest value options that lead to great gain are again very salient. Having a combination of gain and loss scenarios in an experiment enables the investigation of whether the positive and negative distractor effects, related to mutual inhibition and divisive normalisation respectively, are driven by the distractor's value, salience or both. *Figure 6a and b* show four hypothetical cases of how the distractor may influence accuracy. Hypothesis one suggests that larger distractor *values* (*Figure 6a*, first row, left-to-right), which correspond to fewer losses or more gains, are related to greater accuracies (brighter colors). This is also predicted by the mutual inhibition component of the dual route model (*Figure 1*) and can be described as a positive D effect (*Figure 6b*). Hypothesis two suggests larger distractor saliences (*Figure 6a*, second row, center-to-sides) are related to greater accuracies (brighter colors). This can be described as a positive |D| effect (*Figure 6b*). Under this hypothesis the mutual inhibition decision making component receives salience, rather than value, as an input. Hypotheses 3 and 4 are the opposites of Hypotheses 1 and 2, and predict negative distractor effects as a result of the divisive normalisation component depending on whether the input involves value or salience. Hypothesis three predicts a value-based effect in which larger distractor values (*Figure 6a*, third row, left-to-right) are related to poorer accuracies (darker colors). Hypothesis four predicts a salience-based effect in which larger distractor saliences (*Figure 6a*, fourth row, center-to-sides) are related to poorer accuracies (darker colors). It is important to note that these four hypotheses are not necessarily mutually exclusive. The earlier sections have demonstrated that positive and negative distractor effects can co-exist and predominate in different parts of decision space. Value-based and salience-based distractor effects can also be teased apart with a combination of gain and loss scenarios.

To test these hypotheses, we adopted this approach in an additional experiment performed at the same time as *Experiment 3 Hong Kong*, in which half of the trials included options that were all positive in value (gain trials) and the other half of the trials included options that were all negative in value (loss trials; the loss trials were not analysed in the previous sections). We therefore refer to these additional trials as belonging to *Experiment 7 Loss Experiment* (n = 40 as in *Experiment 3 Hong Kong*). The effect of signed D reflects the value of the distractor while the effect of the unsigned, absolute size of D (i.e. |D|) reflects the salience of the distractor. The correlation between these two parameters was low (*r* = 0.005), such that it was possible to isolate the impact that they each had on behaviour.

As in other experiments, we first plotted the accuracy as a function of difficulty (HV-LV) and relative distractor value (D-HV). For ease of comparison, *Figure 3e* that illustrates the accuracy data for the gain trials in *Experiments three* is shown again in the right panel of *Figure 6c*. As described before, when the decisions were hard (bottom rows) larger distractor values were associated with greater accuracies (left-to-right: the colors change from dark to bright; also see *Figure 4b*) and when the decisions were easy, larger distractor values were associated with poorer accuracies (left-to-right: the colors change from bright to dark; also see *Figure 4b*). In a similar manner, the left panel of *Figure 6c* shows the accuracy data of the loss trials in *Experiment 7*. Strikingly, on both hard and easy trials (top and bottom rows), larger distractor values were associated with poorer accuracies (left-to-right: the colours changes from bright to dark).

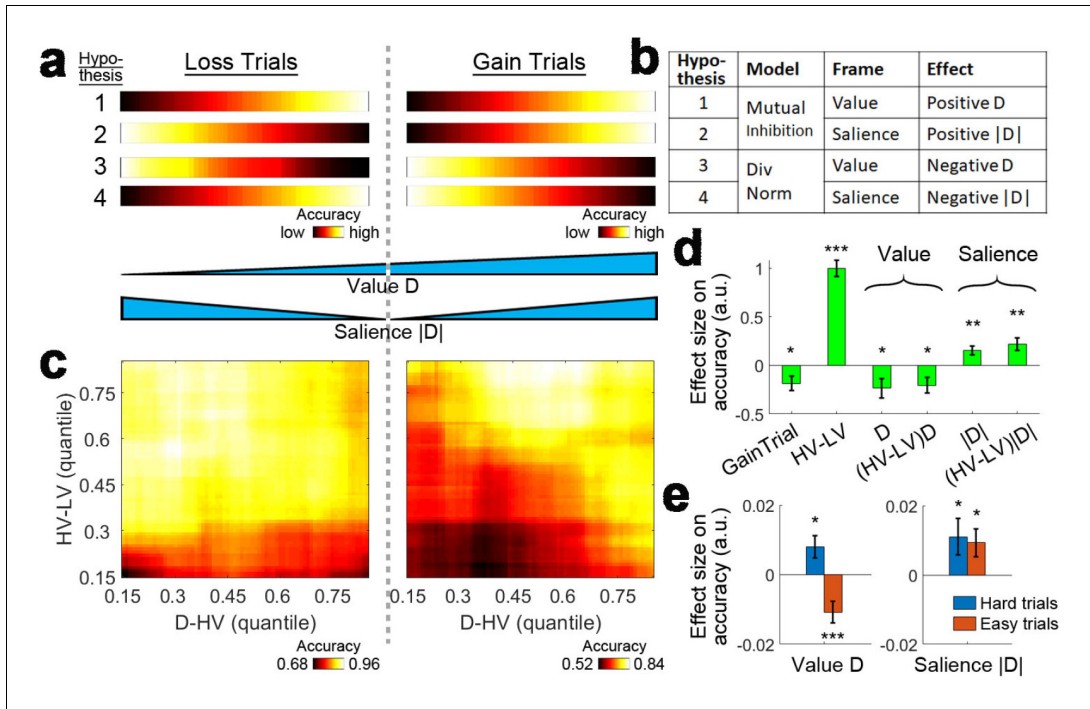

**Figure 6.** Loss Experiment. (a, bottom) Value and salience are collinear when values are only related to rewards (right half) but it is possible to determine whether each factor has an independent effect by also looking at choices made to avoid losses (both left and right halves). (a, top and b) Four hypothetical effects of distractor on accuracy. The first and second hypotheses suggest that the distractor effect is positive, which is predicted by the mutual inhibition model and is related to the distractor's value and salience respectively. The third and fourth hypotheses suggest that the distractor effect is negative, which is predicted by the divisive normalisation model, and is related to its value and salience respectively. All four hypothesis are not mutually exclusive – value and salience are orthogonal factors and positive/negative distractor effects can predominate different parts of decision space. (c, right) A plot identical to that in *Figure 3e* that shows the data from the gain trials of Experiment 3 Hong Kong. Accuracy (light-yellow indicates high accuracy, dark-red indicates low accuracy) is plotted across the decision space defined by decision difficulty (HV-LV) and relative distractor value (D–HV). (c, left) A similar plot using the data from the loss trials of Experiment 7 Loss Experiment is shown. (d) GLM analysis indicates the distractor value D had a negative effect, suggesting that accuracy was more impaired on trials with distractors that were associated with fewer losses or more gains. In contrast, the distractor salience |D| had a positive effect, suggesting that accuracy was more facilitated on trials with more salient distractors (i.e. those related to larger gains or losses). (e, left) The negative value D effect was significant on easy trials (orange) and reversed and became positive on hard trials (blue). (e, right) In contrast, the positive salience |D| effect was significant on both hard and easy trials. In the dual route model, there are two components that guide decision making in parallel. This pattern suggests that the positive distractor effect of the mutual inhibition component is related to the salience and value of the distractor whereas the negative distractor effect of the divisive normalisation component is most closely related to the value of the distractor. *p<0.05, **p<0.01, ***p<0.001. Error bars indicate standard error.

To isolate the value-based and salience-based effects of D, we performed GLM6a (Materials and methods) to analyse both the gain and loss trials in *Experiments 3* and *7* at the same time. GLM6 includes the signed and unsigned value of D (i.e. D and |D| respectively). We also included a binary term, GainTrial, to describe whether the trial presented gain options or loss options and, as in GLM1a, we included the HV-LV term and its interaction with D but now also with |D| [i.e. (HV-LV)D and (HV-LV)|D| respectively]. The results showed a negative effect of value D ($\beta = -0.236$, $t_{39} = -2.382$, p=0.022; *Figure 6d*) and a negative effect of (HV-LV)D interaction ($\beta = -0.205$, $t_{39} = -2.512$, p=0.016). In addition, there was a positive effect of salience |D| ($\beta = 0.152$, $t_{39} = 3.253$, p=0.002) and a positive effect of (HV-LV)|D| ($\beta = 0.219$, $t_{39} = 3.448$, p=0.001). Next, we examined closely the value-based and salience-based effect in different parts of decision space.

As in the analysis for Experiments 1–6 in *Figure 6e*, we split the data (which included both gain and loss trials) according to the median HV-LV, such that the distractor effects can be examined on hard and easy trials separately. We applied GLM6b that first partialled out the effects of HV-LV, HV+LV and GainTrial from the accuracy data and then tested the overall effect of value D across the gain and loss trials. Similar to Experiments 1–6, a positive value D effect was identified on hard trials ($\beta$ = 0.008, $t_{39}$ = 2.463, p=0.017; *Figure 6e*, left) and a negative value D effect was identified on easy trials ($\beta$ = −0.011, $t_{38}$ = −3.807, p<$10^{-3}$; note that one participant was excluded due to the lack of variance in the accuracy data). Then we applied GLM6c which was similar to GLM6b but the value D term was replaced by the salience |D| term. The results showed that there were positive salience |D| effects on both hard ($\beta$ = 0.011, $t_{39}$ = 2.119, p=0.041; *Figure 6e*, right) and easy trials ($\beta$ = 0.009, $t_{38}$ = 2.338, p=0.025).

Taken together, in Experiments 1–6 a positive distractor effect predicted by the mutual inhibition model and a negative distractor effect predicted by the divisive normalisation model were found on hard and easy trials respectively. The results of Experiments 3 and 7 suggest that these effects are value-based and that the effects are continuous across the gain and loss value space. In addition, however, there was also a positive distractor effect that was salience-based that appeared on both hard and easy trials, suggesting that the effects driven by the mutual inhibition decision making component can be both value-based and salience-based.

In the future it might be possible to extend the models outlined in *Figure 1* and S1 to provide a more quantitative descriptions of behaviour. While this topic is of great interest it will require modellers to agree on how loss trials might be modelled. For example, one possibility is to have a negative drift rate in the models that we have used. This implies that the decisions will then be about which option to avoid rather than which option to choose, because options that are more negative in value will reach the decision threshold more quickly. It is unclear, however, whether participants used such an avoidance approach in Experiment 7. Hence, we have refrained from modelling the results of Experiment 7.

## A relationship between attentional capture and positive distractor-salience effects

Although *Gluth et al., 2018* reported an absence of divisive normalisation and positive distractor effects, as we have noted they emphasised the attentional capture effect. In essence the attentional capture effect is the observation that sometimes participants attempted to choose the distractor itself rather that one or other of the choosable options (even though this ran contrary to the instructions they had been given). In a similar vein, when analysing the data from both gain and loss trials in *Experiments 3* and *7*, we also found that the positive effect of the distractor reflected its salience, in addition to value; as suggested by Gluth and colleagues, salient distractors capture attention. We therefore considered, in a final experiment, whether there is any relationship between these two findings – attentional capture and the positive distractor-salience effect.

We performed *Experiment 8 Eye Movement* (n = 35), using a new procedure (*Figure 7*), in which we probed the relationship between attentional capture and positive salience-distractor effects. First, as in other experiments, when GLM5 was applied we found a similar negative (HV-LV)D effect ($\beta$ = −0.258, $t_{33}$ = −2.593, p=0.014) and a positive (HV+LV)D effect ($\beta$ = 0.743, $t_{33}$ = 5.417, p<$10^{-5}$) on choice accuracy (*Figure 7—figure supplement 1*) suggesting the presence of both positive distractor effects and divisive normalisation effects respectively. Note that in this analysis, data from one outlier participant was removed since the (HV-LV)D effect was larger than the mean by 4.1 times the standard deviation.

Next, we analysed the eye movement data collected from a subset of participants (n = 21) and tested the relationships between option value and fixation frequency. We found evidence for attentional capture when D had a high value; participants fixated D more frequently when D's value was high (*Figure 8a*; GLM7; $\beta$ = 0.058, $t_{20}$ = 4.719, p<0.001). The attractor network model (*Chau et al., 2014*; *Wang, 2002*; *Wang, 2008*) or a mutual inhibition model such as the one outlined in *Figure 1a* also predicts that high value distractors should become the focus for behaviour. For example, in the cortical attractor network model, this is due to the pulse of activity in the inhibitory interneuron pool that follows from the high-valued D input. This slows the selection of all options, which subsequently allows more time for evidence in favor of HV rather than LV to be accumulated. However, such models do more than simply predict the unavailable distractor option, D should be the

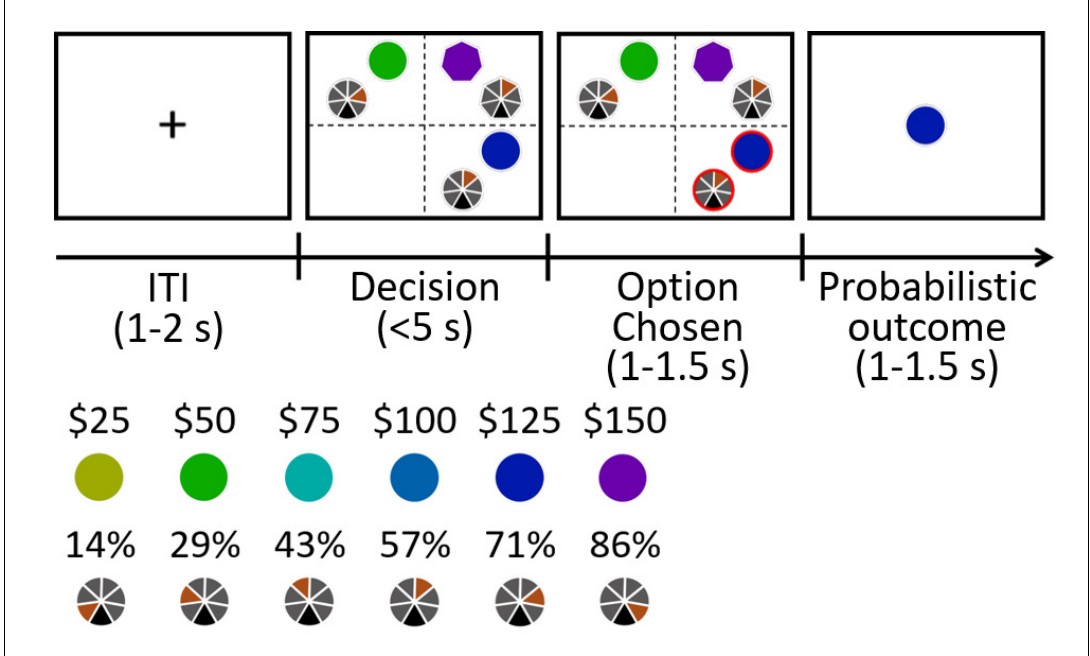

**Figure 7.** Experiment 8: eye tracking experiment. The behavioural paradigm was adapted from that used in Experiments 1–7. (**a**) Each choosable option was represented by a pair of circles. Each distractor option was represented by a pair of heptagons. The screen was divided into four quadrants and each quadrant had two positions for presenting a pair of stimuli associated with an option, making up a total of eight positions. The eight positions were all 291 pixels from the center of the screen and equally separated. Options and distractors appeared in different quadrants on each trial. (**b**) The reward magnitude of an option/distractor was represented by the color of one component stimulus whereas the angle of the tick on the other component stimulus indicated reward probability.

The online version of this article includes the following figure supplement(s) for figure 7:

**Figure supplement 1.** As in Experiments 1–6, in Experiment eight the distractor effect was modulated as a function of difficulty and the total value of the choosable options.

focus of behaviour. In addition they argue that participants should go on to shift focus to the HV option as they realise the distractor option D is unavailable, and so no longer attend to it, so that the distractor boosts rather than lessens accuracy.

Consistent with this prediction in GLM8 we found that as the value of D increased so did gaze shifts between D and HV (*Figure 8b*; $\beta = 0.022$, $t_{20} = 2.937$, p=0.008), while gaze shifts between D and LV decreased ($\beta = -0.031$, $t_{20} = -3.365$, p=0.003). These effects could not merely be due to more fixations at HV or LV per se because we partialled out the effects of fixations at HV, LV and D on the gaze shifts before we tested the relationships between D value and gaze shifts. In addition, the effect was directionally specific. We applied the same GLM8 to predict the proportion of gaze shift from D to HV, D to LV, HV to D, and LV to D. The results showed that larger D values were associated with more D-to-HV shifts (*Figure 8c*; $\beta = 0.028$, $t_{20} = 4.589$, p<0.001) and with fewer D-to-LV shifts ($\beta = -0.027$, $t_{20} = -4.001$, p<0.001). In contrast, the value of D was unrelated to the frequencies of the opposite HV-to-D shift ($\beta = -0.005$, $t_{20} = -0.890$, p=0.384) and LV-to-D shift ($\beta = -0.004$, $t_{20} = -0.633$, p=0.533). A two-way ANOVA confirmed that there was a Direction (from/to D)×Option (HV/LV) interaction effect ($F_{1,20}=11.360$, p=003). Finally, more D-to-HV shifts (*Figure 8d*; GLM9; $\beta = 0.028$, $t_{20} = 2.782$, p=0.012) and fewer D-to-LV shifts ($\beta = -0.043$, $t_{20} = -5.360$, p<10$^{-4}$) were related to higher choice accuracies.

In summary, both Gluth and colleagues (*Gluth et al., 2018*) and we think attention is captured by D (*Figure 8a*). However, our data also suggest D fixations guided by large D values were followed by D-to-HV gaze shifts (*Figure 8b,c*). This was associated with more accurate HV choices because there was a higher chance that the HV option was last fixated (*Figure 8d*).

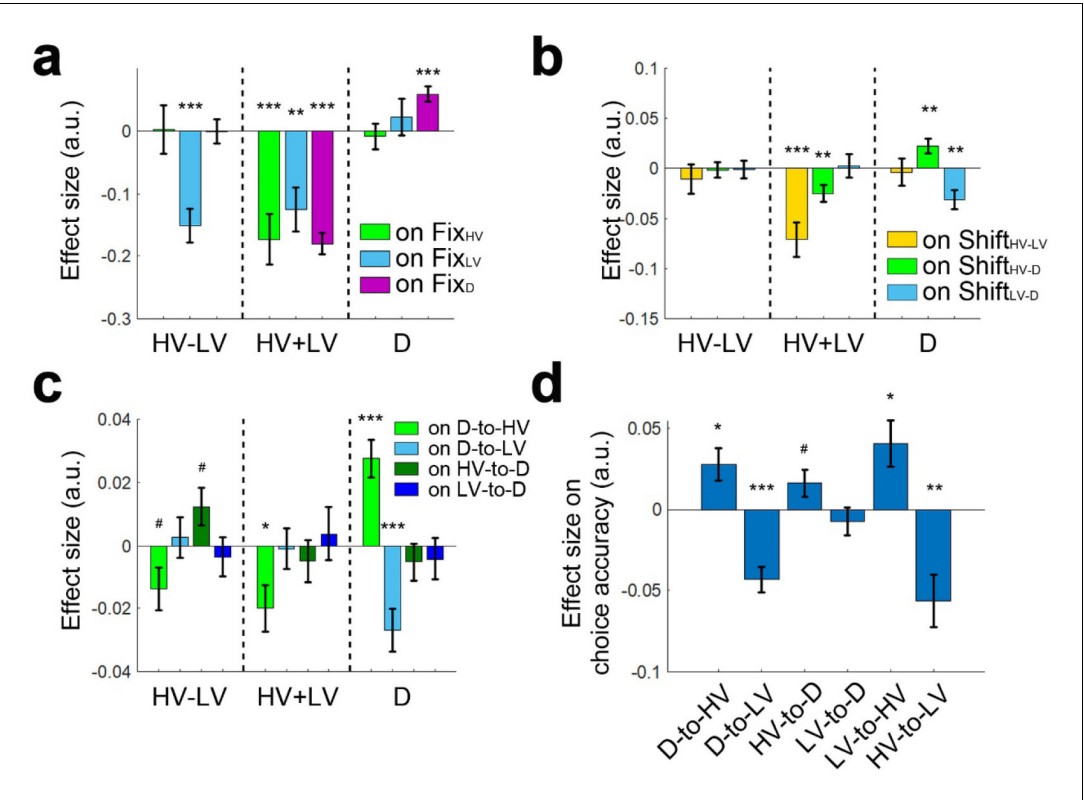

**Figure 8.** Larger D values were associated with more gaze shifts from D to HV and more accurate decisions. (**a**) A multivariate regression analysis showed that larger D values were associated with attentional capture effects as indexed by more fixations at D (right, purple bar). In addition, larger HV-LV difference was associated with fewer fixations at LV (left, blue bar) and larger total HV+LV values were associated with fewer fixations in general (middle). (**b**) As D value increased so did gaze shifts between D and HV (right, green bar), while gaze shifts between D and LV decreased (right, blue bar). These effects could not merely be due to more fixations at HV or LV per se because the effects of fixations at HV, LV and D on the gaze shifts were partialled out before testing the relationships between D value and gaze shifts. (**c**) The effect was directionally specific; larger D values were associated with more D-to-HV shifts and fewer D-to-LV shifts but not the opposite (right, green and blue bars; HV-to-D or LV-to-D shifts). (**d**) In turn, more D-to-HV shifts and fewer D-to-LV shifts predicted greater decision accuracy. Fix$_{HV}$ fixation at HV; Fix$_{LV}$ fixation at LV; Fix$_D$ fixation at D; Shift$_{HV-LV}$ gaze shift between HV and LV; Shift$_{HV-D}$ gaze shift between HV and D; Shift$_{LV-D}$ gaze shift between LV and D. # $p<0.1$, *$p<0.05$, **$p<0.01$, ***$p<0.001$. Error bars indicate standard error.

The online version of this article includes the following figure supplement(s) for figure 8:

**Figure supplement 1.** The same analysis performed in *Figure 8* was performed by including one additional participant and the results remained similar.

## Discussion

There has been considerable interest in the mechanisms mediating decision making. The majority of studies have focused on binary choice scenarios in which participants decide between two options (*Glimcher, 2002*; *Shadlen and Kiani, 2013*) but it is increasingly clear that the presence of an additional distractor may have an impact on which decisions are made between two choosable options. The nature of this impact varies as a function of the decision's position within a decision space defined by the values of the choosable options (HV, LV) and the distractor (*Figures 1*, *3* and *4*). When the values of choosable options and distractor are relatively low and high respectively then divisive normalisation means that decision accuracy is robustly and consistently impaired in data sets from a total of 243 participants from three sites (*Experiments 1–6* and *8*) including data sets from which such effects have previously been claimed to be absent (*Figures 4–6*, *Appendix 3—figure 1*).

When decisions are difficult (HV-LV is small) then high value as opposed to low value distractors robustly and consistently lead to an increase in accuracy in the same data sets.

A further experiment demonstrated a key difference in the nature of the positive distractor effect and the divisive normalisation effect. While the divisive normalisation effect is related to the distractor value on easy trials and the positive distractor effect is related to the distractor value on hard trials, an additional positive distractor effect is also related to the salience of distractor on both hard and easy trials. When participants make decisions between options associated with losses, the presence of more appealing distractor options (i.e. those associated with less loss) are also less salient. On easy trials (*Figure 6c*, left panel, top rows) the distractors exert a combination of a value-based divisive normalization effect and a salience-based distractor effect (i.e. the reverse of the 'positive distractor effect') that results in a particularly strong negative effect on accuracy. In contrast, on hard trials (*Figure 6c*, left panel, bottom rows) the distractors exert a combination of a value-based positive distractor effect and a salience-based effect (i.e. reversed 'positive distractor effect') that result in a particularly weak effect on accuracy.

Positive distractor effects may be a consequence of interactions not just between the representations of choosable options in neural networks but interactions between these representations and the representation of the distractor. *Chau et al., 2014* argued that one way to think about these interactions is in terms of the cortical attractor model of decision making (*Wang, 2002*; *Wang, 2008*; *Wong and Wang, 2006*). In such models separate pools of recurrently connected excitatory interneurons represent each possible choosable option and the distractor. Each pool receives an excitatory input proportional to its value. A common pool of inhibitory interneurons mediates competition or comparison between the pools of excitatory neurons representing the choosable options and distractor. As each option pool becomes more active it increases the activity in the inhibitory pool and this in turn leads to inhibition of the other option pools. Ultimately this means that the network ends up in one of a limited number of attractor states in which the pool representing the chosen option is in a high firing state but the pools representing the other options are in low firing states. If the option that is left in the high firing state is the highest value option (HV) then the decision taken is accurate. This should be the outcome of the competition because this pool received the highest input. However, the presence of some degree of stochasticity in the neural activity levels means that this is not always the case and when decisions are difficult (HV-LV is low) then sometimes the wrong choice is taken. Such effects are less likely to occur when the inhibitory interneurons mediating the decision process are more active, so that the evidence accumulation process is extended and decisions are less influenced by the noise. Sometimes the inhibitory interneuron pool is more active simply because of an individual difference; individuals with higher levels of inhibitory neurotransmitters such as gamma-aminobutyric acid (GABA) in vmPFC/mOFC are more accurate (*Jocham et al., 2012*). The inhibitory interneuron pool will also be more active when the distractor has a high value (*Chau et al., 2014*). In other situations, allowing participants more time to consider a decision, by giving them a later opportunity to revise their initial decision, also leads to greater accuracy (*Resulaj et al., 2009*; *van den Berg et al., 2016*).

Such predictions are not, however, limited to the cortical attractor model. A similar prediction might be made by other models, such as the mutual inhibition model used here, which is essentially a diffusion model but which posits interactions between the diffusion process comparing the values of the choosable options and the processes involved in selecting (and subsequently inhibiting selection) of the distractor (*Figure 1a,c*).

Decision making has been linked to neural processes in a number of brain circuits and it is increasingly clear that more than a single neural mechanism for decision making exists and that they operate in parallel (*Hunt and Hayden, 2017*; *Rushworth et al., 2012*). The mechanisms are not completely redundant with one another. Different mechanisms operate on different time scales (*Meder et al., 2017*; *Wittmann et al., 2016*), are in receipt of different types of information (*Hunt et al., 2018*; *Kennerley et al., 2009*), and are anatomically placed to exert different types of influence on behaviour (*Hunt and Hayden, 2017*; *Rushworth et al., 2012*). There has been particular interest in two cortical regions in humans and other primates such as macaques: vmPFC/mOFC and IPS (*Boorman et al., 2009*; *Chau et al., 2014*; *Hunt and Hayden, 2017*; *Hunt et al., 2012*; *Papageorgiou et al., 2017*; *Philiastides et al., 2010*; *Shadlen and Kiani, 2013*). The positive distractor effect may be particularly associated with vmPFC/mOFC. Activity in vmPFC/mOFC reflects the key decision variable – the difference in value between choosable options – during decision

making (*Basten et al., 2010*; *Boorman et al., 2009*; *Chau et al., 2014*; *De Martino et al., 2013*; *Fouragnan et al., 2019*; *Hunt et al., 2012*; *Lim et al., 2011*; *Lopez-Persem et al., 2016*; *Papageorgiou et al., 2017*; *Philiastides et al., 2010*; *Strait et al., 2014*; *Wunderlich et al., 2012*). However, the size of the vmPFC/mOFC signal reflecting the difference in value between choosable options increases as D increases (*Chau et al., 2014*; *Figure 4*). A similar phenomenon has also been noted in vmPFC/mOFC in macaques (*Fouragnan et al., 2019*). Moreover, individual variation in the size of the neural effect is related to individual variation in the size of the positive distractor effect in behaviour (Figure 5 in *Chau et al., 2014*). In addition, it is noteworthy that lesions in vmPFC/mOFC disrupt the balance between positive and negative distractor effects. VmPFC/mOFC lesions leave both macaques and human patients more vulnerable to the disruptive effects of distractors that are predicted by divisive normalisation (*Noonan et al., 2017*; *Noonan et al., 2010*).

Both lesion and neuroimaging experiments suggest that vmPFC/mOFC is especially important in novel as opposed to over-trained decisions and in decisions involving multi-attribute options (*Fellows, 2006*; *Hunt et al., 2012*; *Papageorgiou et al., 2017*). It is therefore possible that the positive distractor effects seen in the current experiment and elsewhere (*Fouragnan et al., 2019*) may be most prominent when decisions are being made between multi-attribute options or option values that have been newly learned or which are changing. A valuable distractor slows down the decision-making process and reduces choice stochasticity.

By contrast, divisive normalisation may be particularly associated with IPS. Divisive normalisation is a useful feature for a neural network model to have if it is to adapt to contexts with different overall levels of input (*Carandini and Heeger, 2012*). Chau and colleagues (Figure 7 in *Chau et al., 2014*) reported a divisive normalisation-like decrement in performance that was associated with a decrease in the decision variable signal in IPS. Moreover, individual variation in the IPS-signal change was correlated with individual variation in the behavioural effect. Divisive normalisation has also been reported in the value signals recorded from individual neurons in IPS (*Louie et al., 2011*; *Louie et al., 2014*). IPS neurons frequently have sensorimotor fields and given the prominence of divisive normalisation effects throughout sensory brain circuits it is perhaps not surprising that normalisation is also found in value-driven signals in IPS (*Louie et al., 2015*). In contrast to vmPFC/mOFC, decision-related signals in IPS become more prominent rather than less prominent with practice and training (*Grol et al., 2006*). If divisive normalisation is indeed linked to IPS then it may become more prominent with familiarity. Other types of distractor effects may be mediated by activity changes in yet other circuits (*Li et al., 2018*).

## Materials and methods

### Summary of approach

We first conducted a series of computational modelling analyses to make predictions about the effect of distractors on choice accuracy. Then, we re-analysed empirical data from a series of experiments reported by *Chau et al., 2014* and *Gluth et al., 2018* and report data from three new experiments. In the first experiment, we re-analysed a data set that comes from a group of participants who performed an fMRI study. We refer to this first experiment as *Experiment 1 fMRI2014*. The second data set comes from Gluth and colleagues' experiment 4 (*Experiment 2 Gluth4*). We focus on this data set next because it employs an identical schedule to the one used by Chau and colleagues. The third experiment again uses the same schedule but is previously unpublished. It was collected at a third site – Hong Kong – and so we refer to it as *Experiment 3 Hong Kong*. The fourth, fifth, and sixth experiments are re-analyses of Gluth and colleagues experiments 1, 2, and 3 and are therefore referred to as *Experiment 4 Gluth1, Experiment 5 Gluth2, Experiment 6 Gluth3* respectively. An additional analysis was based on additional trials also collected at the same time as those collected for *Experiment three* and which again have not previously been published. The procedure used in these trials was similar to that used in all previous experiments and the schedule is very similar to that used in *Experiments 1, 2,* and *3*. The difference between these trials and those in all earlier experiments is that participants' choices lead them to lose rather than win money. This experiment was conducted to establish whether positive distractor effects and divisive normalisation effects were related to the value or the salience of the distractor. It is therefore referred to as *Experiment 7 loss experiment*. The final experiment is also previously unpublished and used a new experimental

procedure that made it possible to examine eye movements. It is therefore referred to as *Experiment 8 Eye Movement*. Finally, we note that although we have previously shared our data with Gluth and colleagues, the arrangement was not reciprocal, and so this is the first time we have been able to analyse the data from their experiments.

## Computational modelling

The *mutual inhibition model* is a simplified form of a biophysically plausible model that is reported elsewhere (*Chau et al., 2014*). It involves three pools of excitatory neurons $P_i$, each receives noisy input $E_i$ from an option $i$ (i.e. HV, LV or D option) at time $t$:

$$E_{i,t} \sim N\left(dV_i, \sigma^2\right)$$

where $d$ is the drift rate, $V_i$ is the value of option $i$ (HV, LV or D) and $\sigma$ is the standard deviation of the Gaussian noise. The noisy input of all options (HV, LV and D) are all provided simultaneously.

All excitatory neurons excite a common pool of neurons $P_{Inh}$ that exerts inhibition $I_t$ to the excitatory neurons to the same extent:

$$I_t = f \frac{\sum_i^k E_{i,t}}{k}$$

where $k$ is the number of excitatory neuron pools (i.e. k=3 in this model) and $f$ is the level of inhibition (set at 0.5).

The evidence $y_{i,t+1}$ of choosing option $i$ at time $t+one$ follows:

$$y_{i,t+1} = y_{i,t} + \left(E_{i,t} - I_t\right)$$

The HV or LV option, but not the D option, is chosen when its cumulative evidence exceeds a decision threshold of 1. The reaction time is defined as the duration of the evidence accumulation before the decision threshold is reached added by a fixed non-decision time (Tnd) of 300 ms to account for lower-order cognitive processes before choice information reaches the excitatory neurons (*Grasman et al., 2009*; *Ratcliff et al., 1999*; *Tuerlinckx, 2004*). If no option is selected within 6 s, the trial is considered as indecisive and is omitted from the analysis.

The *divisive normalisation model* follows the same architecture, except that there are only two pools of excitatory neurons, and each receives *normalised* input from the HV or LV option. The D only participates in this model by normalising the input from the HV and LV options. The normalised input of the HV or LV option follows the following equation:

$$E_{i,t} \sim N\left(d\frac{V_i}{V_{HV} + V_{LV} + V_D}, \sigma^2\right)$$

where $d$ is the drift rate, $V_i$ is the value of option $i$ (HV or LV) and $\sigma$ is the standard deviation of the Gaussian noise. The inhibition $I_t$ and evidence $y_{i,t+1}$ follow the same equations as the mutual inhibition model.

The *dual route model* involves a mutual inhibition component route and a divisive normalisation component route that run in parallel and independently. A decision is made as soon as one of the component routes contains evidence that exceeds the decision threshold.

Finally, the *null model* is similar to the mutual inhibition model, except that it lacks a pool of excitatory neurons that receive input from the D.

In *Figure 1*, the value sum HV+LV is set at [3.0, 3.2, 3.4 . . . 5]. The value difference HV-LV is set at [0.6, 0.8, 1.0 . . . 2]. The relative distractor value (D-HV) is set at [-1.2, -0.8, -0.4 . . . 4.8]. The simulation of each model was run for all combinations of value sum, value difference and relative distractor value and for 5000 iterations for each combination. Simulated data from each iteration were then randomly assigned to 105 "simulated participants"; this ensured that the number of simulated participants matched the total number of participants actually tested in Experiments 1-3. The levels of d and σ of each model (i.e. the mutual inhibition, divisive normalisation or dual route model) were selected in order to produce an overall choice accuracy of 0.85. For the mutual inhibition model, d, and f were set at 1.3 s−1, 1 s−1 and 0.5 respectively. For the divisive normalisation model, d, and f were set at 5.5 s−1, 0.6 s−1 and 0.5 respectively. For the dual route model, d, and f of the mutual

inhibition component were set at 0.8 s−1, 0.6 s−1 and 0.5 respectively; d, and f of the divisive normalisation component were set at 1.0 s−1, 0.6 s−1 and 0.5 respectively. The proportions of indecision trials (no decision made in 6 s) are less than 0.2% in all models in *Figure 1*. In the models that apply fitted parameters (*Figure 3—figure supplement 2*) the proportions of indecision trials are 3.9% in the mutual inhibition model and less than 0.1% in the dual route, divisive normalisation, and null models.

## Model fitting and comparison

The d and σ parameters of each model were fitted separately at the individual level to the choices and RTs of each participant in *Experiments 1–6*, using the VBA-toolbox (http://mbb-team.github.io/VBA-toolbox/). In the dual route model, the mutual inhibition and divisive normalisation components involved separate d and σ parameters during the fitting. The other model parameters (i.e. f, Tnd, $V_i$ and k) were fixed at the values mentioned above, except for some models reported in *Figure 3—figure supplement 3*. The parameter estimation procedure involved a two-stage procedure that employed a Variational Bayesian analysis under the Laplace approximation (*Daunizeau et al., 2014*). First, an initial search was performed over a grid of fixed priors (i.e. with zero variance) with d at [0.01, 0.1, 1, 10] and σ at [0.1, 1]. Second, the set of priors that shows the best fit (highest log-evidence) was selected, assigned with a variance of 1, and fitted to participants' behaviour.

To compare the goodness-of-fit between the models, the log-model evidence of all models and all participants were then tested using a group-level random-effect Bayesian model selection (BMS) procedure (*Penny et al., 2010*). BMS estimates the exceedance probability (xp) that indicates how likely a model is the best fit model compared to other competing models in the population from which participants were drawn. The same model comparison procedures were repeated by including participants from the same experiment only.

## Human participants

A total of 243 participants participated in the experiments including 21 (9 female; age range 19–34 years) in *Experiment 1*, 44 (36 female; age range 18–46) in *Experiment 2*, 40 (20 female; age range 18–27) in *Experiment 3*, 23 (14 female; age range 18–54) in *Experiment 4*, 49 (24 female; age range 19–46) in *Experiment 5*, 31 (21 female; age range 20–47) in *Experiment 6*, 40 in *Experiment 7*, who were the same participants as in *Experiment 3*, and 35 (20 female; age range 19–42) in *Experiment 8*. Eye movement data were collected from 25 of these 35 participants in *Experiment 8*. Experiments 3, 7 and 8 were approved by ethics committee of The Hong Kong Polytechnic University and Experiment one was approved by that of University of Oxford.

We conducted a posteriori power analysis to confirm that the sample sizes of the experiments were adequate. All analyses were conducted using criteria of power = 80% and alpha = 0.05 two tailed. Experiments 2–6 were replicates of Experiment one in which they all tested whether there was a difficulty-dependent distractor effect via the (HV-LV)(D-HV) term in GLM1a. In Experiment 1, the effect size in Cohen's d for the (HV-LV)(D-HV) effect was d = 0.790. The required sample size was calculated at 13 participants and Experiments 1–6 all involved larger samples. Experiment seven was conducted simultaneously with Experiment three in which it tested the salience-based effect of the distractor via the |D| term in GLM6a. The effect size in Cohen's d for the |D| effect was d = 0.514. The required sample size was calculated at 30 participants and the sample of Experiment seven exceeded this size. Experiment eight examined the impact of attentional capture on decision making. One key analysis was to examine whether more gaze shifts from D to HV were related to greater accuracies (GLM9), in which the effect size in Cohen's d was 0.607. The required sample size was calculated at 22 participants. The sample size of Experiment 8, after excluding four participants using the inclusion criterion of data validity >85% (see *Eye tracking experiment procedures*), was one participant less than this number estimated in a posteriori power analysis. However, when we relaxed the inclusion criterion to >70% data validity, a total of 22 participants were included and the results remained similar (*Figure 8* and *Figure 8—figure supplement 1*).

## Multi-attribute decision-making task (Experiments 1–7)

The experimental task used in *Experiment one* has previously been described by *Chau et al., 2014* as follows (*Figure 1*). Participants chose repeatedly between stimuli associated with different reward

magnitudes (£2, £4, £6, £8, £10, £12) and probabilities (12.5%, 25%. 37.5%, 50%, 62.5%, 75%, 87.5%), represented by colors (red to blue) and orientations (0˚ to 90˚) of rectangular bars. However, associations between visual features and decision variables were counterbalanced across subjects. Participants were presented with 150 trials of two-option trials (in which no distractor was presented) and 150 distractor trials (300 trials in total) randomly interleaved. All the option value configurations in the two-option trials were matched with the *available* options in the distractor trials.

Each trial began with a central fixation cross indicating an inter-trial interval (3–6 s) followed by an *initial phase* in which two (two-option trials) or three (distractor trials) stimuli were presented in randomly selected screen quadrants. The initial phase was brief – only 0.1 s. Then, in the *decision phase*, orange boxes were presented around two stimuli indicating those options were available for choice. In distractor trials, a purple box was also presented around the third stimulus to indicate a distractor. Subjects were instructed to select one of the available options within 1.5 s. Subjects were warned they were 'too slow' if no response was made within 1.5 s and a new trial began. After an option was chosen, the box surrounding it turned red in the *interval phase* (1–3 s). Then the edge of each stimulus turned either yellow or grey in the *outcome phase* to indicate, respectively, whether the choice had been rewarded or not (1–3 s). The final reward allocated to the subject on leaving the experiment was calculated by averaging the outcome of all trials. Subjects learned the task and visual feature associations and experienced Distractor trials in a practice session before scanning. At the end of practice, all subjects chose HV on >70% of two-option trials when it was associated with both higher reward magnitude and probability.

We provided incentives for subjects to attend to the visual features of every stimulus by interleaving 'catch' trials between decision phase and interval phase in 15% of all the trials. In this way we ensured that it was unlikely that subjects would ignore the distractor values. In a 'catch' trial, the word 'MATCH' was presented once subjects selected an option in the decision phase (1 s). Then, an exemplar stimulus was presented at the center of the screen and subjects had to indicate, within 2 s, the position of the same stimulus presented before and during the decision phase. Feedback was then given to indicate whether the response was correct or not (1.5 s). The trial then continued with the resumption of the interval phase, followed by the outcome phase. Each correct response in the 'catch' trial added an extra 10 pence to the final reward.

A very similar procedure was used in *Experiment three* with the exception that the reward magnitudes were presented in Hong Kong dollar ($25, $50, $75, $100, $125, $150). A very similar approach was used in *Experiment 7*. Participants performed trials that were identical to those in *Experiment three* with the exception that now participants ran the risk of losing rather than gaining money of the same amount. The same participants were involved in both *Experiments 3* and *7* in randomised order. The final reward was calculated by the average gain in *Experiment three* deducted by the average loss in *Experiment 7*.

Similar approaches were used in *Experiments 2, 4, 5,* and six reported by *Gluth et al., 2018* with the exception that the reward magnitudes were presented in Swiss Francs (CHF2, CHF4, CHF6, CHF8, CHF10, CHF12). In *Experiments 4–6*, 56 additional 'novel' distractor trials were introduced to test the presence of any decoy effects in their study. The same number of two-option trials with identical HV and LV options but lacking the distractor were also included.

## Eye tracking experiment procedures

The behavioural paradigm used in the eye tracking experiment (*Experiment 8 Eye Movement*) was adapted from that of our 2014 study (*Chau et al., 2014*). In this experiment, the choosable and distractor options were represented by circles and heptagons respectively (*Figure 7a*). Each choosable option was composed of a combination of two circles and the distractor consisted of two heptagons. The reward magnitude of an option was indicated by the color of one of the component shapes, whereas the angle of the tick on the other component shape indicated the option's reward probability (*Figure 7b*). The screen was divided into four quadrants and each quadrant had two positions for presenting a pair of stimuli associated with an option, making up a total of eight positions. The eight positions were all 291 pixels from the center of the screen and equally separated.

The experiment consisted of 150 Distractor Trials (two options available for choice and one distractor, unavailable for choice, were presented) and 150 Two-Option Trials (two choosable options in the absence of a distractor were presented) in randomised order. On each Distractor Trial, a fixation cross was presented for 1 to 2 s. Then two available options and one distractor option were

presented on three of the four quadrants of the screen and a choice had to be made within 5 s by pressing one of the four buttons corresponding to the four quadrants. The frame of the circles related to the chosen option then turned red in color for 1 to 1.5 s. Finally, a circle was presented at the center of the screen for 1 to 1.5 s, indicating whether the chosen option had led to a reward (the color of the circle matched with that of the chosen option) or not (a grey circle).

Eye movement data were recorded using a Tobii TX300 eye tracker with a sampling rate of 300 Hz. The screen attached to the eye tracker was 23 inch in size, 16:9 aspect ratio and $1920 \times 1080$ pixels resolution. Each circle/heptagon stimulus associated with an option was $256 \times 256$ pixel in size. Eye movement data were recorded from both eyes. The dominant eye of each subject was determined using the Miles test and only data from the dominant eye was analysed. Validity of each sample was determined by the Tobii Pro SDK software package. Eye movement data of a subject were excluded when the average validity across all samples was below 85% and so eye movement data from four of the initial 25 subjects tested were not analysed further (although the behavioural data from all 25 subjects were still analysed together with the behavioural data from 10 participants used to pilot the behavioural paradigm). Only data recorded during the decision phase were analysed. Trials where average validity within a trial was below 85% was excluded from analysis. On each trial, an area-of-interest (AOIs) of $300 \times 300$ pixel were drawn surrounding each option stimulus. The size of an AOI was slightly larger than the size of a stimulus ($256 \times 256$ pixel) to account for any minor calibration error. A fixation was defined when the gaze stayed within an AOI for more than 50 ms.

## Analysis procedures

Behaviour was analysed using a series of GLMs containing the following regressors:

GLM1a: $\text{logit(accuracy)} = \beta_0 + \beta_1\, z_{(HV-LV)} + \beta_2\, z_{(D-HV)} + \beta_3\, z_{(HV-LV)}\, z_{(D-HV)} + \varepsilon$
GLM1b: $\text{logit(accuracy)} = \beta_0 + \beta_1\, z_{(HV-LV)} + \beta_2\, z_{(HV+LV)} + \beta_3\, z_{(D-HV)} + \beta_4\, z_{(HV-LV)}\, z_{(D-HV)} + \varepsilon$
GLM1c: $\ln(P_{HV}/P_j) = \beta_{j,0} + \beta_{j,1}\, z_{(HV-LV)} + \beta_{j,2}\, z_{(D-HV)} + \beta_{j,3}\, z_{(HV-LV)}\, z_{(D-HV)} + \varepsilon_j$

where HV, LV, and D refer to the values of the higher value choosable option, the lower value choosable option, and the distractor respectively. $P_{HV}$ and $P_j$ refer to the probability of choosing the HV option or option j. j = [LV, D]. $\varepsilon$ refers to the unexplained error. GLM1a and GLM1c only differed by that the regressions were binomial and multinomial respectively. $z(x)$ refers to $z$-scoring of term $x$, which is applied to all terms in all GLMs in this study. In addition, all interaction terms in all GLMs are calculated after the component terms are $z$-scored.

GLMs2a-c involved a stepwise procedure to partial out the effects of HV and LV from the choice accuracy data before the distractor effects were tested on the residual $\varepsilon_0$ in the second step:

GLM2a:
Step 1, $\text{logit(accuracy)} = \beta_0 + \beta_1\, z_{(HV-LV)} + \beta_2\, z_{(HV+LV)} + \varepsilon_1$
Step 2, $\varepsilon_1 = \beta_3 + \beta_4\, z_{(D-HV)} + \varepsilon_2$
GLM2b:
Step 1, $\text{logit(accuracy)} = \beta_0 + \beta_1\, z_{(HV-LV)} + \beta_2\, z_{(HV+LV)} + \varepsilon_1$
Step 2, $\varepsilon_1 = \beta_3 + \beta_4\, z_{(D)} + \varepsilon_2$
GLM2c:
Step 1, $\text{logit(accuracy)} = \beta_0 + \beta_1\, z_{(Difficulty)} + \varepsilon_1$
Step 2, $\varepsilon_1 = \beta_2 + \beta_3\, z_{(D-HV)} + \varepsilon_2$
GLM2d:
Step 1, $\text{logit(accuracy)} = \beta_0 + \beta_1\, z_{(Difficulty)} + \varepsilon_1$
Step 2, $\varepsilon_1 = \beta_2 + \beta_3\, z_{(D)} + \varepsilon_2$

Where in GLM2c,d Difficulty $= w_1\, z_{[Mag(HV)-Mag(LV))} + w_2\, z_{[Mag(HV)+Mag(LV))} + w_3\, z_{[Prob(HV)-Prob(LV)]} + w_4\, z_{[Prob(HV)+Prob(LV)]}$ (HV)/Prob(HV) and Mag(LV)/Prob(LV) are the reward magnitude/probability of the HV and LV options respectively. Difficulty is the weight sum of the differences and sums between the attributes of HV and LV option. The weights $w_1$, $w_2$, $w_3$, $w_4$ are extracted from GLM 3b (i.e. $\beta_1$, $\beta_2$, $\beta_3$, and $\beta_4$ in GLM3b respectively).

GLM3a: $\text{logit(accuracy)} = \beta_0 + \beta_1\, z_{(HV-LV)} + \beta_2\, z_{(HV+LV)} + \varepsilon$
GLM3b: $\text{logit(accuracy)} = \beta_0 + \beta_1\, z_{[Mag(HV)-Mag(LV))} + \beta_2\, z_{[Mag(HV)+Mag(LV)]} + \beta_3\, z_{[Prob(HV)-Prob(LV)]} + \beta_4\, z_{[Prob(HV)+Prob(LV)]} + \varepsilon$
GLM4: $\text{logit(accuracy)} = \beta_0 + \beta_1\, z_{(SubjDiff)} + \beta_2\, z_{(Congruence)} + \beta_3\, z_{(D-HV)} + \beta_4\, z_{(SubjDiff)}\, z_{(D-HV)} + \varepsilon$

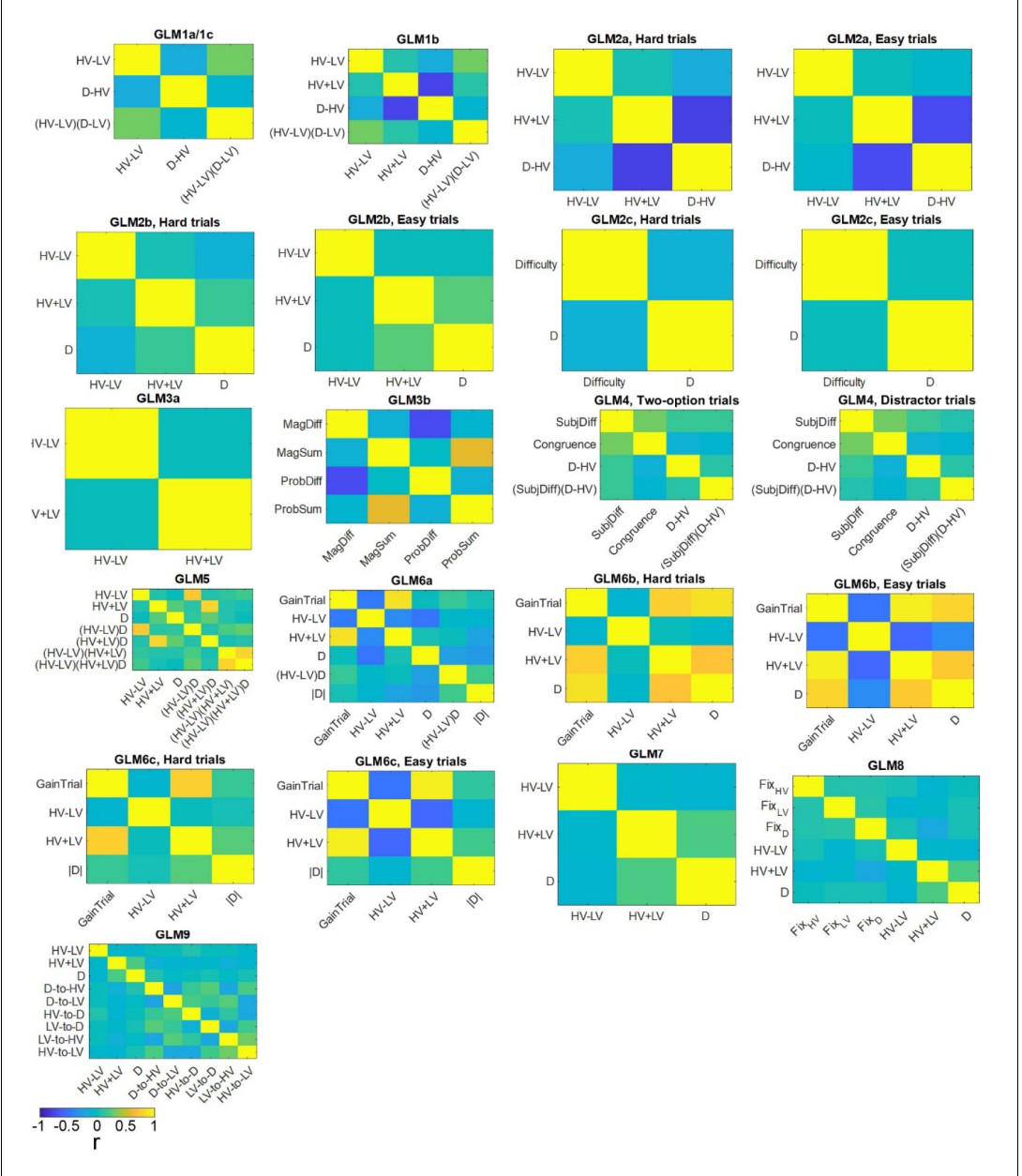

**Figure 9.** Correlations between regressors. Each plot shows the levels of correlation between regressors in the general linear models used. Correlation is presented as Pearson's r averaged across participants.

The online version of this article includes the following figure supplement(s) for figure 9:

**Figure supplement 1.** Variance inflation factor of each regressor.

SubjDiff refers to the subjective difficulty estimate best explaining behaviour in each experiment. SubjDiff is defined simply as HV-LV in *Experiment 1 fMRI2014* and *Experiment 3 Hong Kong* and as the weighted sum of probability and magnitude differences in *Experiment two* and *Experiments 4, 5, 6 (Gluth1-4)*. Congruence is a binary variable indexing whether or not both probability and magnitude components of one option are better than the probability and magnitude components of the other options. In contrast to congruent trials, the probability of one option is higher and the magnitude of the other option is higher on incongruent trials.

GLM5: logit(accuracy) = $\beta_0 + \beta_1\,z_{(HV-LV)} + \beta_2\,z_{(HV+LV)} + \beta_3\,z_{(D)} + \beta_4\,z_{(HV-LV)}\,z_{(D)} + \beta_5\,z_{(HV+LV)}\,z_{(D)} + \beta_6\,z_{(HV-LV)}\,z_{(HV+LV)} + \beta_7\,z_{(HV-LV)}\,z_{(HV+LV)}\,z_{(D)} + \varepsilon$

GLM6a: logit(accuracy) = $\beta_0 + \beta_1\,z_{(GainTrial)} + \beta_2\,z_{(HV-LV)} + \beta_3\,z_{(D)} + \beta_4\,z_{(HV-LV)}\,z_{(D)} + \beta_5\,z_{(|D|)} + \beta_5\,z_{(HV-LV)}\,z_{(|D|)} + \varepsilon$

GLM6b: Step 1, logit(accuracy) = $\beta_0 + \beta_1\,z_{(GainTrial)} + \beta_2\,z_{(HV-LV)} + \beta_3\,z_{(HV+LV)} + \varepsilon_1$

Step 2, $\varepsilon_1 = \beta_4 + \beta_5\,z_{(D)} + \varepsilon_2$

GLM6c: Step 1, logit(accuracy) = $\beta_0 + \beta_1\,z_{(GainTrial)} + \beta_2\,z_{(HV-LV)} + \beta_3\,z_{(HV+LV)} + \varepsilon_1$

Step 2, $\varepsilon_1 = \beta_4 + \beta_5\,z_{(|D|)} + \varepsilon_2$

GainTrial is a binary variable indexing whether a given trial in the *Experiment 3/Experiment seven* data set was a trial on which participants could potentially win or lose money.

GLM7: $Fix_j = \beta_{j,0} + \beta_{j,1}\,z_{(HV-LV)} + \beta_{j,2}\,z_{(HV+LV)} + \beta_{j,3}\,z_{(D)} + \varepsilon_j$ where $Fix_j$ refers to the frequency of fixation on an option. j = [HV, LV, D].

GLM8: Step 1, $Shift_j = \beta_{j,0} + \beta_{j,1}\,z_{[Fix(HV)]} + \beta_{j,2}\,z_{[Fix(LV)]} + \beta_{j,3}\,z_{[Fix(D)]} + \varepsilon_{j,1}$

Step 2, $\varepsilon_{j,1} = \beta_{j,4} + \beta_{j,5}\,z_{(HV)} + \beta_{j,6}\,z_{(LV)} + \beta_{j,7}\,z_{(D)} + \varepsilon_{j,2}$

where $Shift_j$ refers to the frequency of bidirectional gaze shifts between two options. j=[HV-LV, HV-D, LV-D].

GLM9: Step 1, logit(accuracy) = $\beta_0 + \beta_1\,z_{(HV-LV)} + \beta_2\,z_{(HV+LV)} + \beta_3\,z_{(D)} + \varepsilon_1$

Step 2, $\varepsilon_1 = \beta_4 + \beta_5\,z_{[Shift(D-to-HV)]} + \beta_6\,z_{[Shift(D-to-LV)]} + \beta_7\,z_{[Shift(HV-to-D)]} + \beta_8\,z_{[Shift(LV-to-D)]} + \beta_9\,z_{[Shift(LV-to-HV)]} + \beta_{10}\,z_{[Shift(HV-to-LV)]} + \varepsilon_2$

The autocorrelations between regressors and variance inflation factors of each regressor in each GLM is shown in *Figure 9* and *Figure 9—figure supplement 1* respectively. These analyses were run using the Statistics and Machine Learning toolbox of Matlab 2018a (The MathWorks, Inc, USA).

## Choice simulation procedures

We ran Simulations 1–3 to test whether any difficulty-dependent distractor effect emerged as a result of statistical artefact. In Stimulations 1 and 2, simulated choices were generated by incorporating the effect of HV-LV only. Then GLM1a was applied to analyse the accuracy data of the simulated choices to test whether the D-HV and (HV-LV)(D-HV) effects could be found when these effects were supposed to be absent in the simulated choice. The simulations involved the same number of hypothetical participants as in *Experiments 1–6* (i.e. n = 208) and the simulation of each hypothetical participant was iterated for 1000 times. In Simulation 1, the effect of HV-LV that was used to generate the simulated choice in each hypothetical participant was estimated from the choices of one actual participant using GLM1a, where D-HV and (HV-LV)(D-HV) were also present in the model. In Simulation 2, the effect of HV-LV was estimated from another GLM where only the HV-LV term was present.

Simulation three was run as a positive control analysis to confirm that the distractor effects should be present in the data when they are introduced during the generation of the simulated choices. Similar procedures to those used in Simulations 1 and 2 were employed to generate the simulated choices, however, now all the effects of the terms present in GLM1a, i.e. HV-LV, D-HV and (HV-LV)(D-HV), were incorporated.

## Data availability

The codes for running the mutual inhibition model, divisive normalisation model, dual route model and null model can be found at: https://doi.org/10.5061/dryad.k6djh9w3c. Behavioural data of Experiments 1, 3 and 7 can be found at: https://datadryad.org/stash/dataset/doi:10.5061/dryad.040h9t7. Behavioural data of Experiments 2, 4–6 can be found from a link provided by *Gluth et al., 2018* at: https://osf.io/8r4fh/. Behavioural and eye tracking data of Experiment eight can be found at: https://doi.org/10.5061/dryad.k6djh9w3c.

## Acknowledgements

This work was supported by the Hong Kong Research Grants Council (25610316), Wellcome Trust (WT100973AIA; 203139/Z/16/Z) and Medical Research Council (MR/P024955/1).

# Additional information

### Funding

| Funder | Grant reference number | Author |
|---|---|---|
| Research Grants Council, University Grants Committee | 25610316 | Bolton K H Chau |
| Wellcome | WT100973AIA | Matthew F S Rushworth |
| Wellcome | 203139/Z/16/Z | Matthew F S Rushworth |
| Medical Research Council | MR/P024955/1 | Matthew F S Rushworth |

The funders had no role in study design, data collection and interpretation, or the decision to submit the work for publication.

### Author contributions

Bolton KH Chau, Conceptualization, Resources, Data curation, Software, Formal analysis, Supervision, Funding acquisition, Validation, Investigation, Visualization, Methodology, Writing - original draft, Project administration, Writing - review and editing; Chun-Kit Law, Conceptualization, Data curation, Formal analysis, Visualization, Methodology, Writing - review and editing; Alizée Lopez-Persem, Software, Formal analysis, Validation, Investigation, Visualization, Methodology, Writing - review and editing; Miriam C Klein-Flügge, Conceptualization, Formal analysis, Visualization, Methodology, Writing - review and editing; Matthew FS Rushworth, Conceptualization, Formal analysis, Supervision, Funding acquisition, Investigation, Writing - original draft, Project administration, Writing - review and editing

### Author ORCIDs

Bolton KH Chau (ID) https://orcid.org/0000-0002-6854-5176
Chun-Kit Law (ID) https://orcid.org/0000-0002-1185-1308
Alizée Lopez-Persem (ID) http://orcid.org/0000-0002-7566-5715
Miriam C Klein-Flügge (ID) http://orcid.org/0000-0002-5156-9833

### Ethics

Human subjects: Experiments 3, 7 and 8 were approved by ethics committee of The Hong Kong Polytechnic University and Experiment 1 was approved by that of University of Oxford.

### Decision letter and Author response

Decision letter https://doi.org/10.7554/eLife.53850.sa1
Author response https://doi.org/10.7554/eLife.53850.sa2

# Additional files

### Supplementary files

• Supplementary file 1. Supplementary tables. (**A**) Regression coefficients estimated using the analysis procedures suggested by *Gluth et al., 2018*. (**B**) Simulation 1: testing the probabilities of Type I and II errors when distractor effects were assumed to be absent in the simulated choice data. (**C**) Simulation 2: an alternative approach for testing the probabilities of Type I and II errors when distractor effects were assumed to be absent in the simulated choice data. (**D**) Simulation 3: testing the probabilities of Type I and II errors when distractor effects were assumed to be present in the simulated choice data.

• Transparent reporting form

## Data availability

The codes for running the mutual inhibition model, divisive normalization model, dual route model and null model can be found at: https://doi.org/10.5061/dryad.k6djh9w3c Behavioural data of Experiments 1, 3 and 7 can be found at: https://datadryad.org/stash/dataset/doi:10.5061/dryad.040h9t7 Behavioural data of Experiments 2, 4-6 can be found from a link provided by Gluth and colleagues (2018) at: https://osf.io/8r4fh/ Behavioural and eye tracking data of Experiment 8 can be found at: https://doi.org/10.5061/dryad.k6djh9w3c.

The following dataset was generated:

| Author(s) | Year | Dataset title | Dataset URL | Database and Identifier |
|---|---|---|---|---|
| Chau BKH, Law C, Lopez-Persem A, Klein-Flügge MC, Rushworth MFS | 2019 | Consistent patterns of distractor effects during decision making | https://doi.org/10.5061/dryad.k6djh9w3c | Dryad Digital Repository, 10.5061/dryad.k6djh9w3c |

The following previously published datasets were used:

| Author(s) | Year | Dataset title | Dataset URL | Database and Identifier |
|---|---|---|---|---|
| Chau BKH, Kolling N | 2018 | Data from: A neural mechanism underlying failure of optimal choice with multiple alternatives | https://doi.org/10.5061/dryad.040h9t7 | Dryad Digital Repository, 10.5061/dryad.040h9t7 |
| Spektor MS, Gluth S, Rieskamp J | 2018 | Value-based attentional capture affects multi-alternative decision making | https://osf.io/8r4fh/ | Open Science Framework, 8r4fh |

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

## Appendix 1

# Simulation suggests that the difficulty-dependent distractor effect was not due to any statistical artefact

To test whether the (HV-LV)(D-HV) effect, for example in GLM1a, could emerge as a statistical artefact if it were absent in a hypothetical situation, we simulated three sets of choice data based on these GLMs. We ran Simulations 1 and 2 where simulated choices were generated using a GLM including only the empirically measured HV-LV effect size as a coefficient estimate. The simulations employed the same set of trials as those performed by the participants. Simulations 1 and 2 differ in whether the effect size of HV-LV is estimated by GLM1a or by a similar GLM in which HV-LV is the only regressor. In both cases, the simulated choices should only contain the HV-LV effect and any distractor effect observed in these simulated choices should be considered a statistical artefact. When we applied GLM1a to analyse the simulated choices, we found a significant HV-LV effect (Simulation 1: $\beta = 0.534$, $t_{207} = 20.970$, p<$10^{-52}$; Simulation 2: $\beta = 0.577$, $t_{207} = 22.337$, p<$10^{-56}$). Importantly, we found no D-HV effect (Simulation 1: $\beta = -0.009$, $t_{207} = -0.583$, p=0.561; Simulation 2: $\beta = -0.004$, $t_{207} = -0.216$, p=0.830) and (HV-LV)(D-HV) effect (Simulation 1: $\beta = 0.010$, $t_{207} = 0.537$, p=0.592; Simulation 2: $\beta = 0.007$, $t_{207} = 0.410$, p=0.682).

In addition, a complementary test was performed to confirm that distractor effects should be detectable with the GLMs that were used if the distractor effect is introduced into the simulated choice data. In order to do this, we ran Simulation 3 that incorporated effects of all terms in GLM1a with effect sizes corresponding to those that had been empirically observed, that is HV-LV, D-HV and (HV-LV)(D-HV), while the simulated choices were generated. When re-fitting the choices produced by this simulation, as expected, we found a positive HV-LV effect ($\beta = 0.545$, $t_{207} = 18.432$, p<$10^{-44}$), positive D-HV effect ($\beta = 0.149$, $t_{207} = 5.564$, p<$10^{-7}$) and negative (HV-LV)(D-HV) effect ($\beta = -0.108$, $t_{207} = -4.129$, p<$10^{-4}$). Thus, these simulations confirm that the difficulty-dependent distractor effect was not due to any statistical artefact (*Palminteri et al., 2017*).

Moreover, we conducted additional simulations to test whether Type I error is inflated in GLM1a. We repeated Simulation 1 (which assumes only an HV-LV, but no D-HV and (HV-LV)(D-HV), effect in the accuracy data) for 1000 iterations and estimated the p value of each regressor in each iteration. *Appendix 1—figure 1a* shows the cumulative distribution of p values of each regressor for all 1000 iterations. Since the statistical threshold is set at p=0.05, to determine the likelihood of Type I errors of D-HV and (HV-LV)(D-HV) it is critical to examine the proportion of simulations that show p<0.05 (at the dotted line in *Appendix 1—figure 1a*). The result show that the Type I error of D-HV and (HV-LV)(D-HV) are 5.6% and 5.1% respectively. Similarly, the Type II error of the HV-LV term can be determined by examining the proportion of iterations with p>0.05 (i.e. 1-cumulative probability at Simulated p=0.05; *Appendix 1—figure 1a*). The results show that Type II error of the HV-LV term is 0%. These results are summarized in *Supplementary file 1B*, GLM1a.

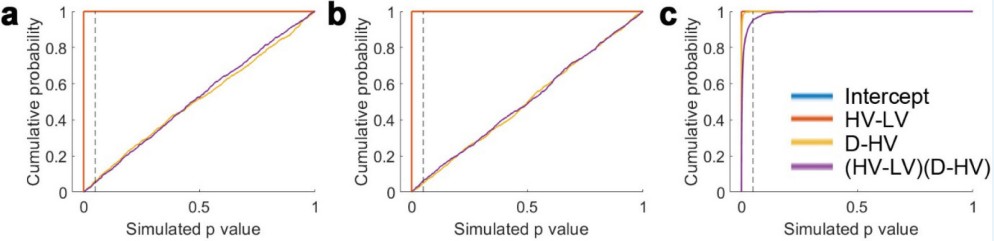

**Appendix 1—figure 1.** No clear indication of inflated statistical artefact in GLM1a. GLM1a includes regressors HV-LV, D-HV and (HV-LV)(D–HV) to predict choice accuracy. (**a**) In Simulation 1, choice accuracy data was simulated using the effect size of HV-LV estimated by

GLM1a from actual participants and assuming that the D-HV and (HV-LV)(D–HV) effects are absent. Then GLM1a was applied again to analyse the simulated choice accuracy data. In 1000 iterations, the chance of obtaining a significant effect (p<0.05; at the dashed line) from the HV-LV term is 100% (i.e. 0% Type II error; orange). For the D-HV and (HV-LV)(D–HV) terms, for which no effect is assumed, the chances of obtaining Type I errors are 5.6% and 5.1% respectively (yellow and purple respectively). (**b**) Simulation 2 is similar to Simulation 1, except that the effect size of HV-LV was estimated from the empirical data using a GLM that only includes the HV-LV term. (**c**) Simulation 3 assumes that HV-LV, D-HV and (HV-LV)(D–HV) effects are all present in the simulated choice accuracy data. The chances of obtaining a significant effect are 0%, 0.1% and 4.7% respectively. Note that in (**a**) and (**b**) the blue and orange lines overlap with each other; in (**c**) the blue, orange and yellow lines overlap with each other.

In a similar manner, we performed these additional simulations for Simulation 2. As before, these two simulations only differ in how the effect of HV-LV is introduced into the simulated accuracy data. In Simulation 1 it was estimated by applying GLM1a to participants' empirical data. In Simulation 2 it was estimated by applying a GLM with only the HV-LV term to participants' empirical data. The results show that Type I errors of D-HV and (HV-LV)(D-HV) are 4.1% and 5.4% respectively and the Type II error of HV-LV is 0% (*Appendix 1—figure 1b*; also see *Supplementary file 1C*, GLM1a).

Finally, the additional simulations for Simulation 3 assume that effects of HV-LV, D-HV and (HV-LV)(D-HV) all exist in the accuracy data. The probabilities of successfully detecting a significant effect of HV-LV, D-HV and (HV-LV)(D-HV) are 100%, 99.8% and 97.1% respectively (*Appendix 1—figure 1c*; also see *Supplementary file 1D*, GLM1a). In other words, Type II errors of these terms are 0%, 0.2% and 2.9% respectively.

Similar simulation procedures are performed with GLMs 1b and 5 and 6 and there is no clear indication that the distractor effects (i.e. the effects of the D-HV and (HV-LV)(D-HV) terms) are driven by statistical artefact. The results are summarized in *Supplementary file 1B–D*.

## Appendix 2

# Control analyses suggested that the difficulty-dependent distractor effect was not due to any statistical artefact

In another set of control analyses we focused on control trials where only two options, but no distractors, were presented. In two sets of experiments, fMRI2014 and Gluth1-4, an equal number of precisely matched trials were collected in which HV and LV options with exactly the same probabilities and magnitudes (and thus values) were employed but no distractor was presented. Thus the same participants chose between the same HV and LV options and a 'notional distractor' of the same value – D – was assigned to the trial but not actually presented. Unlike in the previous analysis, if there is no collinearity between the interaction term and the main effect of difficulty (HV-LV), then there should now be a stronger (Difficulty) (D-HV) interaction effect on distractor trials than the matched 'two-choice' trials. This is what *Chau et al., 2014* examined in their analysis SI.3 and thus we applied a very similar analysis (GLM4) to test the data from fMRI2014 and Gluth1-4. When difficulty was defined as HV-LV in fMRI2014, the distractor-difficulty interaction effect (Difficulty)(D-HV) was stronger on distractor trials than two-choice trials ($t_{109}$ = 2.381, p=0.0279). Similarly, when difficulty was defined as the weighted sum of probability and magnitude differences in Gluth1-4 the (Difficulty)(D-HV) effect was also stronger on distractor trials than two-choice trials ($t_{109}$ = 2.314, p=0.023). Note that there were no 'two-choice' trials in *Experiment 3 Hong Kong* and so it is not possible to run this analysis with those data.

## Appendix 3

### A more complete analysis of distractor effects

In order to examine distractor effects as a function of both difficulty (HV-LV) and the integrated value of the options (HV+LV+D), we plotted the effect of the distractor on accuracy (rather than accuracy itself as in *Figure 3*) as a function of both HV-LV and HV+LV in the dual route model and *Experiments 1 to 3* (*Appendix 3—figure 1*). Again, in both model's simulated choices and participants' actual choice a positive distractor effect (yellow area) is most prominent on the hard trials at the bottom of *Appendix 3—figure 1a-d* while the negative distractor effect (blue area) is most prominent on the easy trials at the top of these figures. As predicted by divisive normalization models the negative distractor effect is most prominent at the left of these figures when the total sum of HV and LV is small.

Hence, rather than just looking at the (HV-LV)D interaction, a more complete three-way GLM (GLM5, Materials and methods) can be performed which also includes the (HV+LV)D term. For completeness we include the other possible interactions terms such as (HV-LV)(HV+LV) and (HV-LV)(HV+LV)D as well. Now the opposite effects exerted by the distractor as a function of difficulty (HV-LV) manifest as a negative (HV-LV)D interaction term; the distractor has a positive effect on decision accuracy when decisions are difficult but a negative effect when decisions are easy just as was seen in previous analyses (*Figures 3* and *4* and *Figure 4—figure supplement 1*). The divisive normalization model also predicts that the negative impact of D is reduced when HV+LV becomes larger and hence the positive distractor effect predicted by a cortical attractor or diffusion model should become predominant. This manifests as a positive (HV+LV)D interaction term. These interaction effects are clearly present in the data of all six experiments that we have surveyed above; there is a negative effect of the (HV-LV)D interaction term ($\beta = -0.116$, $t_{207} = -4.262$, p<$10^{-4}$; *Appendix 3—figure 2a*) and a positive effect of the (HV+LV)D interaction term ($\beta = 0.414$, $t_{207} = 13.066$, p<$10^{-28}$). All experiments showed the same trend in these key interaction terms. A one-way ANOVA showed that they were comparable in size across experiments [*Appendix 3—figure 2b*; (HV-LV)D: $F_{5,202}=1.524$, p=0.184; (HV+LV)D: $F_{5,202}=0.625$, p=0.681]. Finally, a follow-up analysis was run to confirm how the D effect varied as a function of HV-LV or HV+LV. These questions related to the (HV-LV)D and (HV+LV)D interaction effects respectively. We applied a mean split by HV-LV and then estimated the effect of D using GLM2b. On hard trials (small HV-LV), larger D values were related to *greater* choice accuracies ($\beta = 0.056$, $t_{207} = 4.113$, p<$10^{-4}$); whereas on easy trials (large HV-LV), larger D values were related to *poorer* choice accuracies ($\beta = -0.007$, $t_{206} = -2.049$, p=0.042; note that one participant was excluded from this analysis because of the lack of behavioral variance – there was only one inaccurate choice). On trials with a small HV+LV sum, larger D values were associated with poorer choice accuracies ($\beta = -0.009$, $t_{207} = -2.530$, p=0.012); whereas on trial with large HV+LV sums, larger D values were only marginally associated with greater choice accuracies ($\beta = 0.006$, $t_{207} = 1.810$, p=0.072).

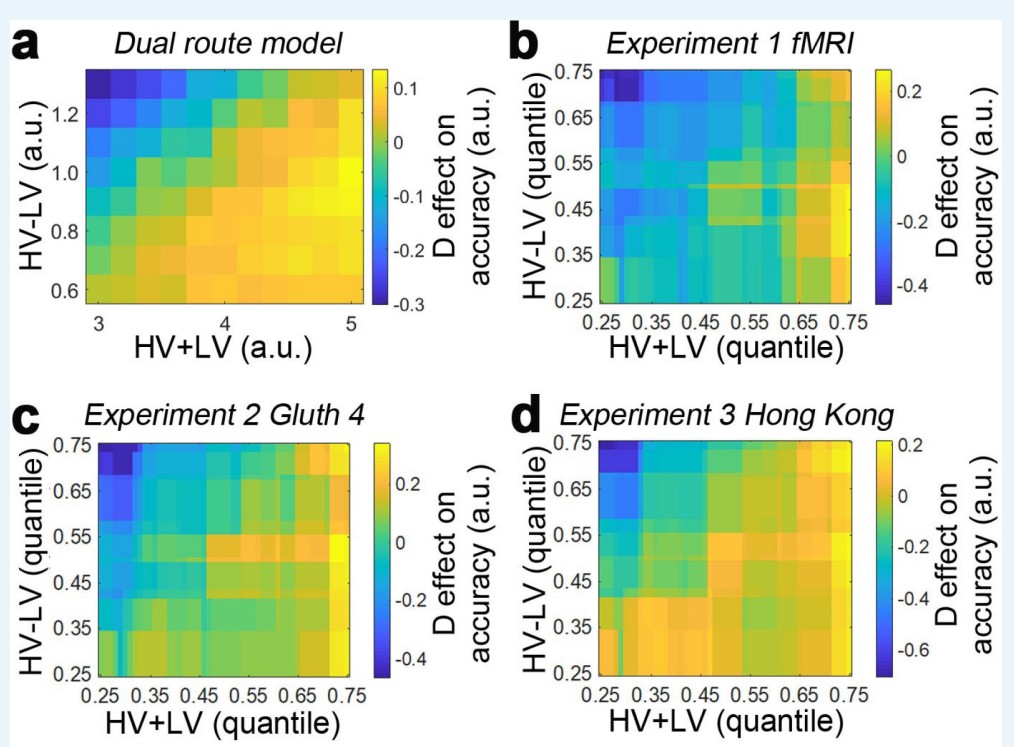

**Appendix 3—figure 1.** Distractor effects measured across the decision space. As in *Figure 1*, the distractor effect (yellow = positive, blue = negative) is plotted across the decision space defined by difficulty (HV-LV) and the total value of the choosable options (HV+LV). Results are shown for (**a**) the dual route model, (**b**) Experiment 1 fMRI2014, (**c**) Experiment 2 Gluth4, (**d**) Experiment 3 Hong Kong.

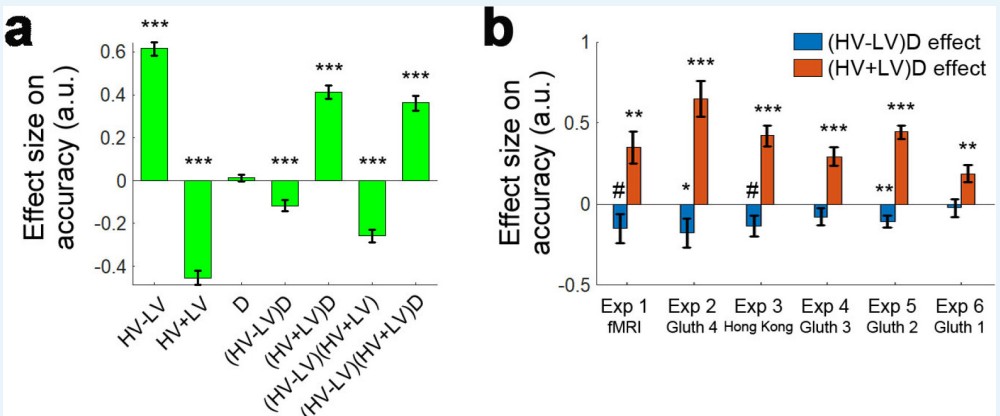

**Appendix 3—figure 2.** The distractor effect was modulated as a function of difficulty and the total value of the chooseable options. (**a**) Here the data have been analysed with a more complete GLM (GLM5) including all main effects and two- and three-way interactions. Across all experiments the (HV-LV)D interaction is negative – the distractor has a positive effect on decision accuracy when decisions are difficult but a negative effect when decisions are easy. This resembles the effects illustrated in *Figures 3–5*. Moreover, across all experiments the (HV+LV)D interaction is positive – when the total value of the choosable options changed from large to small the distractor effect also changed from positive to negative. The divisive normalization model predicts that the negative impact of D is greatest when HV+LV is small (*Figure 1*) but its impact is reduced when HV+LV is large and so the positive distractor effect predicted by the interacting diffusion process model predominates when HV+LV is large. (**b**) The same two key interaction terms, (HV-LV)D and (HV+LV)D, indexing the positive distractor

and divisive normalization effects are illustrated for each component experiment that was included in panel a. ***p<0.001. Error bars indicate standard error.

Finally, we found that there was a significant (HV-LV)(HV+LV)D effect in GLM5 ($\beta = 0.362$, $t_{207} = 10.417$, p<$10^{-19}$; **Appendix 3—figure 2a**). Next we examined how the (HV+LV)D effect or (HV-LV)D effect varied as a function of the third variable (HV-LV). One way to examine this is to look at simple effects in sub-sections of the data but because we are now considering a three way interaction, the necessary subsection may be only a quarter in size of the original data, which could produce unreliable results. Hence, we took an alternative approach by using the beta weights estimated in GLM5 and investigating the three-way interaction effect. In particular, small and large HV-LV (or HV+LV) was defined as the 25th percentile ($z$ score = $-0.675$) and 75th percentile ($z$ score = 0.675) respectively. Then we tested the two-way (HV+LV)D effect at small and large HV-LV:

GLM5: logit(accuracy) = $\beta_0 + \beta_1\,z_{(HV-LV)} + \beta_2\,z_{(HV+LV)} + \beta_3\,z_{(D)} + \beta_4\,z_{(HV-LV)}\,z_{(D)} + \beta_5\,z_{(HV+LV)}\,z_{(D)} + \beta_6\,z_{(HV-LV)}\,z_{(HV+LV)} + \beta_7\,z_{(HV-LV)}\,z_{(HV+LV)}\,z_{(D)} + \varepsilon$
GLM5 at small HV-LV: logit(accuracy) = $\beta_0 + \beta_1\,z_{(HV-LV)} + \beta_2\,(-0.675) + \beta_3\,z_{(D)} + \beta_4\,(-0.675)\,z_{(D)} + \beta_5\,z_{(HV+LV)}\,z_{(D)} + \beta_6\,(-0.675)\,z_{(HV+LV)} + \beta_7\,(-0.675)\,z_{(HV+LV)}\,z_{(D)} + \varepsilon$
GLM5 at large HV-LV: logit(accuracy) = $\beta_0 + \beta_1\,z_{(HV-LV)} + \beta_2\,(0.675) + \beta_3\,z_{(D)} + \beta_4\,(0.675)\,z_{(D)} + \beta_5\,z_{(HV+LV)}\,z_{(D)} + \beta_6\,(0.675)\,z_{(HV+LV)} + \beta_7\,(0.675)\,z_{(HV+LV)}\,z_{(D)} + \varepsilon$

As such, the effect of (HV+LV)D at small HV-LV is ($\beta_5$ - 0.675$\beta_7$) and that at large HV-LV is ($\beta_5$ + 0.675$\beta_7$). The analysis showed that at small HV-LV there was a positive (HV+LV)D effect ($\beta = 0.169$, $t_{207} = 5.656$, p<$10^{-7}$) and at large HV-LV the (HV+LV)D effect was even more positive ($\beta = 0.659$, $t_{207} = 13.999$, p<$10^{-31}$). A similar procedure was applied to test the (HV-LV)D effect at small or large HV+LV. The results showed that at small HV+LV there was a negative (HV-LV)D effect ($\beta = -0.361$, $t_{207} = -9.208$, p<$10^{-16}$), whereas at large HV+LV there was a positive (HV-LV)D effect ($\beta = 0.128$, $t_{207} = 3.953$, p<$10^{-3}$).

We tested the effect of D at different levels of HV-LV and HV+LV. When both HV-LV and HV+LV was small, there was a lack of D effect ($\beta = -0.023$, $t_{207} = -0.778$, p=0.437). When HV-LV was small and HV+LV was large, there was a positive D effect ($\beta = 0.206$, $t_{207} = 7.119$, p<$10^{-10}$). When HV-LV was large and HV+LV was small, there was a negative D effect ($\beta = -0.510$, $t_{207} = -11.414$, p<$10^{-22}$). When HV-LV was large and HV+LV was large, there was a positive D effect ($\beta = 0.379$, $t_{207} = 9.606$, p<$10^{-17}$).

