## [Decision Letter]

**Acceptance summary:**

This manuscript presents a re-analysis of previous data as well as new data on the effects of distractors on value-based choices. It is an ongoing debate in the field whether such effects exist and what form they take. The current results convincingly show that across multiple experiments, distractors both improve and impair choice accuracy, depending on the difficulty of the decision. Moreover, these effects are reproduced by a dual-route model that combines divisive normalization and mutual inhibition.

**Decision letter after peer review:**

Thank you for submitting your article "Consistent patterns of distractor effects during decision making" for consideration by *eLife*. Your article has been reviewed by three peer reviewers, one of whom is a member of our Board of Reviewing Editors, and the evaluation has been overseen by Joshua Gold as the Senior Editor. The following individual involved in review of your submission has agreed to reveal their identity: Rani Moran (Reviewer #2).

The reviewers have discussed the reviews with one another and the Reviewing Editor has drafted this decision to help you prepare a revised submission.

Summary:

In the current study, the authors show robust distractor effects both improving and impairing accuracy in different "parts" of the decision space, despite previous reports (Gluth et al.) that these effects do not exist in some of the datasets (analyzed here). The authors present a "dual-route" model that explains the co-existence of these effects and they relate the facilitation and impairment effects to distractor salience and value respectively.

All reviewers agreed that this research is important, informative and of interest for a broad readership. Reviewers also identified a number of issues which need to be addressed before the paper can be accepted for publication. Most importantly, it would be critical to determine why exactly the authors come to such vastly different conclusions than Gluth et al., when analyzing the same data.

Essential revisions:

1) The paper is extremely long and detailed. The reviewers agreed that it would be good to edit it by delegating non-essential material to the supplement. For example, the sections "distractor effect are not driven by artefacts" and "A more complete analysis of distractor effects" could simply be referenced from the main text but detailed in Supplementary Information. Additionally, the manuscript contains substantial redundancy which could be streamlined.

2) The current paper directly contrasts with the results of Gluth, 2018, whose datasets comprise a portion of the results presented. Critically, both Gluth and the current authors examine identical datasets (Chau, 2014 and Gluth, 2018 Experiment 4) but come to different conclusions about the existence of distracter effects. While the authors in places do offer suggestions about why the conclusions are different, they provide no definite answers. It would be important to do more to address this directly. A simple explanation might be that two opposing effects exist in the data, that these on average can cancel out (given specific HV-LV conditions) – therefore the focus on the interaction term between (HV-LV)(D-HV). However, in the same dataset (Gluth, 2018 Experiment 4), Gluth sees no positive distractor effect (in fact he finds the opposite) and no interaction effect; the authors see both. Barring error, this means the authors are running different analyses and the critical question is what exactly is different?

2.1) Gluth et al., make a very specific technical point about the importance of centering or standardizing HV-LV and D-HV before computing the interaction term; without doing so, the interaction term can be highly correlated with one or both of the main predictor variables. Are predictors centered here prior to calculating interaction terms? Subsection “Distractor effects are not driven by statistical artefact” suggests so, but it is unclear what "normalization" means. Despite the statement in subsection “Distractor effects are not driven by statistical artefact”, there is no mention in the Materials and methods section of (1) whether there is centering before interaction terms are calculated, or (2) if so, in which GLMs. The authors should be explicit here.

2.2) The original analyses in Chau, 2014 and Gluth, 2018 included LV+HV as a covariate in the main analyses, which is not the case here for GLM1 which documents the main finding. Was this excluded for a specific reason, and what are the results if it is included? HV+LV *is* included in the stepwise regression in GLM2, but that is not a straightforward comparison.

2.3) The authors suggest (subsection “Both divisive normalization of value and positive distractor effects co-exist in data sets from three sites”) that including two-choice trials (with a nominal D value of zero) may have biased previous results. This sounds plausible but is speculative. It would help if the authors re-ran their analyses with these trials included. A different result would not only back up their assertion but would provide a more definite explanation for the reported differences in findings.

3) Given the reliance on regression measures throughout the paper, reviewers were concerned about whether there are potential multicollinearity issues, particularly because the predictor variables HV-LV and D-HV may be related (due to task design), and due to interaction terms. Illustrations in Figure 9 suggest that some of the GLMs feature strong correlations.

3.1) Please state whether or not the task design orthogonalized HV, LV, and D.

3.2) Please report multicollinearity measures (e.g. variance inflation factors) for the different regression models. This is a concern for all the models, but in particular GLM5 which has many regressors with related terms.

4) In analyzing Experiment 7, it would be important to investigate interactions with D or |D| (e.g., D*(HV-LV), |D|*(HV-LV)) as such interactions play a critical role in studying distractor effects in the rest of the paper. Additionally, it would be highly informative to present panels as in Figure 1 for this experiment and separately for the reward/loss conditions. Do the patterns look different for gains and losses? And can the dual route model account for separate effects of value and salience? Relatedly, how do the authors think negative values are handled in the normalization model?

5) There is now some history between the authors and Gluth et al. This shows in multiple places in the paper. For the sake of de-escalation, the authors are encouraged to tone down their language. Specific examples include (but are not limited to) the subsection “Both divisive normalization of value and positive distractor effects co-exist in data sets from three sites”.

6) In many places, statistical interactions are not interpreted using "simple effects". When an interaction (e.g., X*Y) is significant it is unclear whether the main effects (e.g. of X) is meaningful or whether the simple effects change sign depending on the other variable (e.g., Y). It would be important to conduct follow-up simple effect analyses. Some of the analyses even contain triple interactions. If these are not interpreted it is difficult to understand what the patterns of results mean.

7) The dual route model is attractive as a simple conceptual mechanism for a combination of effects, but there were some questions about the precise implementation, model comparison, and whether the models can account for RT data:

7.1) As reported in Chau, 2014, distracter input to the mutual inhibition only occurs for a brief period of time (before it is indicated as unchoosable); is the same format used for the divisive normalization model?

7.2) How were relevant model parameters (d and σ) determined in the dual model? It appears that for individual mutual inhibition and normalization models, they were chosen to give 85% correct choices. Is the same thing true for all 4 parameters in the dual model?

7.3) It would be more informative to show model predictions based on parameters that were derived from model fits vis a vis empirical data (and show qualitative aspects that the dual route model fits better than the other models) as these parameters are more relevant.

7.4) The authors suggest that the ability of the model to generate both effects is due to the relative speed of each model component in different value conditions. While intuitive, it would be helpful if the authors actually showed this to be the case in the simulation data. Since the two processes act entirely in parallel, it would be simple to perform the simulations for the individual component models (using the dual model parameters) and report average RTs (in the [HV-LV, D-HV] space). In other words, rather than showing solely predictions for accuracy, it would be important to show also predictions for RT. Additionally, in models for RT it is essential to include a "residual, non-decision, time". This doesn't seem to be the case here but should be.

[Editors' note: further revisions were suggested prior to acceptance, as described below.]

Thank you for submitting your article "Consistent patterns of distractor effects during decision making" for consideration by *eLife*. Your article has been reviewed by three peer reviewers, one of whom is a member of our Board of Reviewing Editors, and the evaluation has been overseen by Joshua Gold as the Senior Editor. The following individuals involved in review of your submission have agreed to reveal their identity: Kenway Louie (Reviewer #1); Rani Moran (Reviewer #2).

The reviewers have discussed the reviews with one another and the Reviewing Editor has drafted this decision to help you prepare a revised submission.

Summary:

All reviewers agreed that the authors have done a very thorough job of addressing the essential revisions and minor comments, and that the revised manuscript is much improved. In particular, the discussion of the significance of interaction effects in Gluth, 2018 is illuminating, as it supports the general conclusion in this manuscript. There are, however, a few remaining issues that the authors should address before publication.

Essential revisions:

1) In comment 6, the reviewers previously raised the issue of interpreting triple interactions (e.g., analyses pertaining to GLM5). The authors focus on interpretations of 2-way interactions (HV+LV)D and (HV-LV)D whose signs are of theoretical importance in their framework. The concern is that these interactions are qualified by a significant triple interaction (HV-LV)(HV+LV)D. This means that the sign of each of the 2-way interactions can change as a function of the third variable. Therefore, a simple effect analysis here should examine simple 2 way interactions as a function of the third variable. This analysis is critical for the interpreting the findings.

2) Some questions remain about the modelling.

2.1) In comment 7.4. Reviewers previously raised the importance of modeling residual time. In response the authors included RT but arbitrarily fixed it to 300ms rather that allowing it to vary freely. They argue in their response letter that "because non-decision time has no reason to be different across our models, it would not bring more evidence in favor of one or another model during model comparisons.". It may be impossible to determine this a-priori because in each model residual-RT might trade-off differently with the other parameters. It would be important to re-fit the model with free residual time parameters to see which model is best. It is important to rule out that the results of model-comparison are due to arbitrary assumptions about residual rt.

2.2) A very similar issues pertains to the parameter f (inhibition) which was also fixed to a constant value rather than being a free parameter. This could potentially affect model comparison results.

2.3) It is still unclear whether in fitting the dual-channel model, each channel had its own free parameters or whether they were identical for both channels.

2.4) Why are models comparisons reported only for Experiments 1-3 but not for the other experiments?

3) In comment 4 the reviewers previously raised questions about the loss trials. The revised version does not fully address these questions.

3.1) There are still questions pertaining to how to model loss trials. According to the current equations, it seems that drift rates can be negative for the mutual inhibition model, and in the normalization model, the drift will be strongest for the highest loss option. Clearly, if this is correct, the model will require adjustments to account for loss trials and these have to be explained.

3.2) There are also questions pertaining to whether and how the model can account for differences between gain and loss trials. These are important issues because the results seem quite different for gain and loss trials. It would be important to perform a model comparison for the loss trial to determine the best model for these trials. It is not clear if the dual route model, or one of the simpler models, is best for loss trials. Additionally- looking at Figure 6 and the results of the regression (panel d) it seems that when D is positive (i.e., D = abs(D)) corresponding regression effects for these two terms offset each other but when D is negative (D = -abs(D)) they compound. So this could simply mean that the distractor effects are stronger for losses than for gains. Is this true? This can be seen, by including in the regression, interactions terms with trial identity (what the authors call GainTrial) instead of terms with abs(D). Furthermore, if distractors effects are indeed stronger for losses then these stronger effects could presumably be caused by adjusting model parameters (e.g., inhibition strength or other parameters). It is important to examine this. In sum, the authors should consider fitting models to loss trials to see (1) which model provides the best account for loss trials, (2) what account do the mechanistic models provide loss trials and for differences between gain and loss trials. This will provide a much more informative understanding of the gain-loss issue as compared to the current reliance on the regression model. Currently the authors argue that there are 2 separate effects in play, one for distractor value and one for distractor saliency. But a more informative way to understand the data might be that the context (gain/loss) modulated the distractor value effect, and to query the mechanistic models to identify the locus of this modulations.

4) Reviewers were still unclear about the meaning of terms in the GLMs. This needs to be clarified so that the models are better understood and evaluated. Just for example consider GLM1:

logit(accuracy) = β0 + β1(HV-LV) + β2(D-HV) + β3(HV-LV)(D-HV) + ε

The authors state that "All interaction terms in all GLMs are calculated after the component terms are z-scored". Does this mean that the terms are z-scored only for the purpose of calculating the interaction, or are they z-scored for the main effects as well? Reviewers think they should be z-scored in all terms not just in interaction terms. Additionally, just to be sure- did the authors z-score HV and LV separately or z-score the difference (HV-LV)? A clearer way to write the model to avoid confusions could be:

logit(accuracy) = β0 + β1 z_(HV-LV) + β2z_(D-HV) + β3z_(HV-LV)*z_(D-HV) + ε (underscore indicates subscript).

---

## [Author Response]

Essential revisions:1) The paper is extremely long and detailed. The reviewers agreed that it would be good to edit it by delegating non-essential material to the supplement. For example, the sections "distractor effect are not driven by artefacts" and "A more complete analysis of distractor effects" could simply be referenced from the main text but detailed in Supplementary Information. Additionally, the manuscript contains substantial redundancy which could be streamlined.

We have moved the subsection "Distractor effects are not driven by artefacts" and subsection "A more complete analysis of distractor effects" to the appendices as requested. While we have moved these sections as requested and while we can see the advantages of moving subsection "Distractor effect are not driven by artefacts" to the appendices we would argue that subsection "A more complete analysis of distractor effects” should remain in the main manuscript. This subsection provides a rationale for when divisive normalization effects should and should not appear and then tests whether or not the predictions are fulfilled (and indeed they are). We think that this an important conceptual point that is not made elsewhere in the main manuscript. In addition, we have tried to streamline the rest of the manuscript while still including the additional analyses suggested by the reviewers in the points below.

2) The current paper directly contrasts with the results of Gluth, 2018, whose datasets comprise a portion of the results presented. Critically, both Gluth and the current authors examine identical datasets (Chau, 2014 and Gluth, 2018 Experiment 4) but come to different conclusions about the existence of distracter effects. While the authors in places do offer suggestions about why the conclusions are different, they provide no definite answers. It would be important to do more to address this directly. A simple explanation might be that two opposing effects exist in the data, that these on average can cancel out (given specific HV-LV conditions) – therefore the focus on the interaction term between (HV-LV)(D-HV). However, in the same dataset (Gluth, 2018 Experiment 4), Gluth sees no positive distractor effect (in fact he finds the opposite) and no interaction effect; the authors see both. Barring error, this means the authors are running different analyses and the critical question is what exactly is different?

In the revised manuscript we have tried to address these points. It is indeed the case that, in brief, our argument is, as the reviewer says “A simple explanation might be that two opposing effects exist in the data, that these on average can cancel out (given specific HV-LV conditions) – therefore the focus on the interaction term between (HV-LV)(D-HV)”.

The reviewer, however, points out that “in the same dataset (Gluth Experiment 4), Gluth sees no positive distractor effect (in fact he finds the opposite) and no interaction effect; the authors see both”. We understand why the reviewer makes this argument. Indeed we initially believed exactly the same; that Gluth et al., made both these argument. It is only after reading Gluth’s et al., paper many times that we realized that only the first part of this statement is true “Gluth sees no positive distractor effect”. The second part of the statement “Gluth sees … no interaction effect” is not true. Very close reading of the paper reveals that although Gluth et al., state that our results were “neither replicable nor reproducible” in both their abstract and main text, they do not provide any statistics regarding the interaction effect in their main paper.

We realize that statement maybe surprising to the reviewer. But Gluth et al., only briefly touch on the interaction term at the end of the Results section. At that point in the main text Gluth et al., note an “incorrect implementation of the interaction term (HV-LV)x(HV-D) in the performed regression analysis. After correcting this error, the effect of HV-D disappears”. On first reading this section we were left with exactly the same impression as the reviewer “Gluth sees … no interaction effect”. It is likely that most other readers will have received the same impression. However, if you read this sentence again you will see that what they are arguing is that the main effect of the distractor is not observed after Gluth et al., perform the analysis with the correct interaction term. Gluth et al., actually avoid saying anywhere in the main text of their paper whether the interaction term itself remains significant. It does; the interaction term is robust even when Gluth et al., analyze the data in the way they suggest. The key statistics are not referred to or presented in their main manuscript but instead appear only in Supplementary file 2 (and some are omitted even from Supplementary file 2).

Careful reading of this table shows that Gluth et al.,, in fact, found a negative (HV-LV)(D-HV) interaction effect on choice accuracy (Gluth et al., 2018) just as we do. In Supplementary file 2 of their paper, they presented the distractor effects on choice accuracy across 5 experiments. We reproduce the table below. Since Gluth’s analysis involved the terms HV-D and (HV-LV)(HV-D), we edited the Table slightly by flipping these terms so that they become D-HV and (HV-LV)(D-HV) respectively and are therefore consistent with the other analyses in our manuscript and the terminology used in the reviewer’s comments and elsewhere in our reply:

Critically, in four out of five experiments, Gluth et al., observed negative (HV-LV)(D-HV) effects. These effects were significant in two experiments (Experiment 3 and Experiment 2 LP) and were marginally significant in another two experiments (Experiment 1 and Experiment 4). They also present an additional analysis that combined data from four experiments (All _(exc. 2LP)_), albeit excluding one dataset in which the interaction effect was clear, and again there was a significant (HV-LV)(D-HV) interaction effect. Hence, we are very sympathetic to the fact that it is easy to read Gluth’s paper and come away with the impression “that they see no interaction term”. This is the impression that we had too and that we believe most readers will have. However, careful reading of the supplementary material suggests that, on the contrary, there is indeed a significant interaction effect – a distractor effect on accuracy (i.e. relative choice accuracy as in Gluth et al., 2018) regardless of which approach is taken to the data.

Careful reading reveals that Gluth et al., explain their suggestion for the specific procedures that should be followed while computing the (HV-LV)(D-HV) interaction term (see Gluth et al., 2018). A critical point concerns how the absence of distractors in the control trials are treated. Gluth et al., argue that they should be assigned a notional value that is the mean of the distractor values taken from the main experimental trials in which distractors were presented. We followed these procedures strictly and re-analyzed their data. We are able to generate virtually identical results to those in their Supplementary file 2. Importantly, there is an additional set of statistics that was omitted from Gluth’s et al.,’ Supplementary file 2; they omitted to show that when they take the approach they recommend with our original 2014 data (referred to as Experiment 1 fMRI in the current manuscript) using Gluth’s et al.,’ suggested analysis approach then yet again the interaction term is significant.

To the best of our knowledge, the only statistic from Gluth’s et al.,’ analysis, that is actually reported in their paper is the effect size of the HV-D term in Cohen’s d (approximately 0.35; see their Figure 5—figure supplement 3). In addition, when we calculated the Cohen’s d of the D-HV term we obtained a very similar effect size d=-0.352, suggesting that we had run the analysis in a very similar way to Gluth et al.,.

In summary, we concur with the reviewer that it is very easy to receive the impression that “Gluth sees no positive distractor effect (in fact he finds the opposite) and no interaction effect”. Moreover, we believe that most other readers will have had this impression too. However, careful scrutiny of the main text of Gluth’s paper reveals no mention of whether or not the interaction term is significant and careful scrutiny of Supplementary file 2 reveals that the interaction term is significant even when Gluth’s et al.,’ suggested analysis approach is used. We have, therefore tried to make this clear in the revised manuscript in a manner consistent with the reviewers’ and editor’s comment 5 as follows:

It might be asked why the presence of distractor effects in their data was not noted by Gluth et al., 2018. The answer is likely to be complex. A fundamental consideration is that it is important to examine the possibility that both distractor effects exist rather than just the possibility that one or other effect exists. This means that it is necessary to consider not just the main effect of D-HV but also D-HV interaction effects. Gluth et al., however, focus on main effects apart from in their table S2 where careful scrutiny reveals that the (D-HV)(HV-LV) interaction is reliably significant in their data. A further consideration concerns the precise way in which the impact of the distractor D is indexed in the GLM particularly on control trials where no distractor is actually presented. Gluth et al., 2018 recommend that a notional value of D is assigned to control trials which corresponds to the distractor’s average value when it appears on distractor trials. In addition, they emphasize that HV-LV and D-HV should be normalized (i.e. demeaned and divided by the standard deviation) before calculating the (HV-LV)(D-HV) term. If we run an analysis of their data in this way then we obtain similar results to those described by Gluth et al., in their Table S2 (Supplementary file 1 here). Although a D-HV main effect was absent, the (HV-LV)(D-HV) interaction term was significant when data from all their experiments are considered together. While Gluth et al., omitted any analysis of the data from Experiment 1 fMRI, we have performed this analysis and once again a significant (HV-LV)(D-HV) effect is clear (Supplementary file 1).

Another possible reason for the discrepancy in the interpretation concerns the other three experiments reported by Gluth et al. We turn to these next. (subsection “Both divisive normalization of value and positive distractor effects co-exist in data sets from three sites”)

2.1) Gluth et al., make a very specific technical point about the importance of centering or standardizing HV-LV and D-HV before computing the interaction term; without doing so, the interaction term can be highly correlated with one or both of the main predictor variables. Are predictors centered here prior to calculating interaction terms? Subsection “Distractor effects are not driven by statistical artefact” suggests so, but it is unclear what "normalization" means. Despite the statement in subsection “Distractor effects are not driven by statistical artefact”, there is no mention in the Materials and methods section of (1) whether there is centering before interaction terms are calculated, or (2) if so, in which GLMs. The authors should be explicit here.

We apologize for the confusion related to the term “normalization”. It actually means that predictors are *z*-scored before interaction terms are calculated. This involves all the component terms centered by their own mean and then divided by their own standard deviation. We applied the same procedures when calculating all interaction terms in every GLM. We have rephrased the sentence in subsection “Examining distractor effects in further experiments”:

“First, all of the interaction terms are calculated after their component parts, the difficulty and distractor terms, are z-scored (i.e. centered by the mean and then divided by the standard deviation).” (subsection “Examining distractor effects in further experiments”).

We have also added in the subsection “Analysis procedures*”* the following line:

“All interaction terms in all GLMs are calculated after the component terms are *z*-scored.”

2.2) The original analyses in Chau, 2014 and Gluth, 2018 included LV+HV as a covariate in the main analyses, which is not the case here for GLM1 which documents the main finding. Was this excluded for a specific reason, and what are the results if it is included? HV+LV *is* included in the stepwise regression in GLM2, but that is not a straightforward comparison.

The reviewer makes a good point about the degree to which analyses might be compared. It is true that GLM1 is different from a key analysis in Chau et al., 2014 and Gluth et al., 2018 by lacking the HV+LV term. However, GLM1 has the strength of being very simple and straightforward by itself – it involves only the HV-LV and D-HV main effect terms and the interaction between the two. GLM2 is a follow-up analysis of GLM1 to investigate the nature of the (HV-LV)(D-HV) interaction effect by testing trials with large and small HV-LVs separately. It involves an HV+LV term, in addition to the HV-LV, and a stepwise procedure in order to partial out completely the effects of HV and LV from the choice accuracy data (note that a combination of HV-LV and HV+LV terms is equivalent to a combination of HV and LV terms). This procedure has a strength of ruling out the possibility that any D-HV effect is driven by the variance of the HV.

It seems that a GLM (GLM1b) that involves both the (HV-LV)(D-HV) and HV+LV terms can fill the gap between GLMs 1 (now renamed as GLM1a) and 2. In other words, GLM1b includes the regressors HV-LV, HV+LV, D-HV and (HV-LV)(D-HV). We applied this to analyze the data in a similar way as that in Figure 3. Critically the results were highly comparable to those of GLM1a – Experiments 1-3 all showed a significant (HV-LV)(D-HV) effect. We have now added these results to our manuscript:

“In addition to being present in the data reported by Chau et al., 2014 and Gluth et al., 2018 the same effect emerged in a third previously unreported data set (Experiment 3 Hong Kong; n=40) employing the same schedule but collected at a third site (Hong Kong). The results were highly comparable not only when the choice accuracy data were visualized using the same approach (Figure 3E), but also when the same GLM was applied to analyze the choice accuracy data (Figure 3F). There was a significant (HV-LV)(D-HV) effect (*β*=-0.089, *t*_39_=-2.242, *p*=0.031). Again there was a positive D-HV effect (*β*=0.207, *t*_39_=5.980, *p*<10^-6^) and a positive HV-LV effect (*β*=0.485, *t*_39_=12.448, *p*<10^-14^). The pattern of results was consistent regardless of whether an additional HV+LV term was included in the GLM, as in Chau et al., 2014; a significant (HV-LV)(D-HV) effect was found in Experiments 1-3 when an additional HV+LV term was included in the GLM (GLM1b; Figure 3—figure supplement 1).” (Results section)

Behavior was analyzed using a series of GLMs containing the following regressors:

GLM1a: logit(accuracy) = β_0_ + β_1_(HV-LV) + β_2_(D-HV) + β_3_(HV-LV)(D-HV) + ε

GLM1b: logit(accuracy) = β_0_ + β_1_(HV-LV) + β_2_(HV+LV) + β_3_(D-HV) + β_4_(HV-LV)(D-HV) + ε” (Materials and methods section)

2.3) The authors suggest (subsection “Both divisive normalization of value and positive distractor effects co-exist in data sets from three sites”) that including two-choice trials (with a nominal D value of zero) may have biased previous results. This sounds plausible but is speculative. It would help if the authors re-ran their analyses with these trials included. A different result would not only back up their assertion but would provide a more definite explanation for the reported differences in findings.

As we have noted above, in the first part of the response to point 2, it is easy to be confused as to whether Gluth’s claims of non-significance refer to the main effects of HV-D or the HV-D interaction terms. We have explained that actually the interaction terms are significant but they are only reported in Gluth’s Supplementary Table S2. We have tried to avoid compounding confusion by not reporting any analysis of two-choice trials with a nominal D value of zero because neither Gluth et al., nor we advocate this approach. Instead, when comparing the results of our analyses, with those adopted by Gluth we have used the one analysis approach that is explained and advocated by Gluth et al., (see Gluth , 2018). A critical point does indeed concern how the absence of distractors in the control trials is treated. Gluth et al., argued that, in addition to a binary term that describes the presence/absence of a distractor, non-distractor trials should be assigned a notional distractor value that is the mean of the distractor values taken from the main experimental trials in which distractors were presented. We followed these procedures strictly and re-analyzed their data. We make clear in the revised manuscript that when we use this approach, we obtain virtually identical results and the interaction term is significant. Moreover we obtain the same results when we applied it to the original data from our 2014 paper (here referred to as Experimental 1 fMRI). These findings are now reported in subsection “Both divisive normalization of value and positive distractor effects co-exist in data sets from three sites” and Supplementary file 1 of our manuscript. As noted the revised text now reads as follows:

It might be asked why the presence of distractor effects in their data was not noted by Gluth et al., 2018. The answer is likely to be complex. A fundamental consideration is that it is important to examine the possibility that both distractor effects exist rather than just the possibility that one or other effect exists. This means that it is necessary to consider not just the main effect of D-HV but also D-HV interaction effects. Gluth et al., however, focus on main effects apart from in their Table S2 where careful scrutiny reveals that the (D-HV)(HV-LV) interaction is reliably significant in their data. A further consideration concerns the precise way in which the impact of the distractor D is indexed in the GLM particularly on control trials where no distractor is actually presented. Gluth et al., 2018 recommend that a notional value of D is assigned to control trials which corresponds to the distractor’s average value when it appears on distractor trials. In addition, they emphasize that HV-LV and D-HV should be normalized (i.e. demeaned and divided by the standard deviation) before calculating the (HV-LV)(D-HV) term. If we run an analysis of their data in this way then we obtain similar results to those described by Gluth et al., in their Table S2 (Supplementary file 1 here). Although a D-HV main effect was absent, the (HV-LV)(D-HV) interaction term was significant when data from all their experiments are considered together. While Gluth et al., omitted any analysis of the data from Experiment 1 fMRI we have performed this analysis and once again a significant (HV-LV)(D-HV) effect is clear (Supplementary file 1).

Another possible reason for the discrepancy in the interpretation concerns the other three experiments reported by Gluth et al., We turn to these next.

3) Given the reliance on regression measures throughout the paper, reviewers were concerned about whether there are potential multicollinearity issues, particularly because the predictor variables HV-LV and D-HV may be related (due to task design), and due to interaction terms. Illustrations in Figure 9 suggest that some of the GLMs feature strong correlations.

In Comment 3.2 it is suggested that the variance inflation factor (VIF) can be calculated to indicate multicollinearity of individual regressors. This is a helpful suggestion and so we have calculated the VIFs in all general linear models (GLMs). Since there is a concern about the multicollinearity issue of the HV-LV and D-HV terms, we checked carefully the VIFs of these terms, as well as the related terms D, (HV-LV)(D-HV), (HV-LV)D, HV+LV and (HV+LV)D. These results are now reported in Figure 9—figure supplement 1. All regressors show VIFs that are less than 10, which is sometimes used as a threshold for indicating high multicollinearity. In addition, the VIFs of all critical terms that are related to the distractor are less than 5.

3.1) Please state whether or not the task design orthogonalized HV, LV, and D.

There are two subtly, but importantly, different ways in which the question whether “the task design orthogonalized HV, LV, and D” might be interpreted. First, it might be interpreted as asking whether the variance in HV, in LV, and in D was unrelated. This interpretation of the question is consistent with comment 3.2 which asks for multicollinearity measures such as variance inflation factors. As described below, in answer to comment 3.2, the variance in HV, LV, and D was not collinear and this is made clear in Figure 1—figure supplement 1 and Figure 1—figure supplement 2 in the revised manuscript. An alternative interpretation, however, is that the question is asking whether one regressor was made to be orthogonal to another by partialling out the shared variance from one of the two regressors. We have made clear whenever we partial out the variance related one regressor by explicitly using the phrase “partial out” to avoid any confusion (for example, subsection “Both divisive normalization of value and positive distractor effects co-exist in data sets from three sites” and Subsection “Analysis procedures) but this was not done by default.

3.2) Please report multicollinearity measures (e.g. variance inflation factors) for the different regression models. This is a concern for all the models, but in particular GLM5 which has many regressors with related terms.

We would like to thank the reviewer for suggesting the calculation of variance inflation factor (VIF), which is a convenient indicator of multicollinearity of individual regressors. All general linear models (GLMs) include regressors with VIF less than 10. In GLM5, about which the reviewer is particularly concerned, the VIFs are all less than 3.106. We have now included the VIFs of all GLMs in Figure 9—figure supplement 1:

4) In analyzing Experiment 7, it would be important to investigate interactions with D or |D| (e.g., D*(HV-LV), |D|*(HV-LV)) as such interactions play a critical role in studying distractor effects in the rest of the paper. Additionally, it would be highly informative to present panels as in Figure 1 for this experiment and separately for the reward/loss conditions. Do the patterns look different for gains and losses? And can the dual route model account for separate effects of value and salience? Relatedly, how do the authors think negative values are handled in the normalization model?

The reviewer has made a very useful suggestion to better relate the effects in Experiment 7 to the earlier sections of the manuscript and to examine closely the interaction effects. To achieve that, we now present the results in Figure 6 in a similar fashion to those in Figure 1J and L, Figure 3E-F and Figure 4. In panels A and B, we first illustrate hypothetically how value-based and salience-based effects should manifest in the accuracy data. Then on the right side of panel C, it shows Figure 3E (which is in the same format as Figure 1B,F,Jj) once again to illustrate the changes in accuracy as a function of D-HV and HV-LV on gain trials. A similar plot that shows the data from loss trials is presented adjacent to it. In panel d, as in Figure 1D,H,L and Figure 3F, it shows the results of GLM6a. As suggested by the reviewer, GLM6a now includes the interaction terms (HV-LV)D and (HV-LV)|D|, in addition to the GainTrial, HV-LV, D and |D| terms. In panel e, as in Figure 4 and Figure 4—figure supplement 1, it shows the results of an analysis that tested the effects of value D and salience |D| on hard and easy trials separately. Critically, these further analyses revealed that the positive distractor effect was both value-based and salience-based, as opposed to our previous notion that it was only salience-based. We have now updated the subsection “Experiment 7: Loss experiment” substantially to describe the results of the additional analysis and how the results are related to the dual route model. The revised figure (Figure 6) and its accompanying legend are shown first below then we show the revised text that describes this part of the results.

The attentional capture model raises the question of whether any distractor effect on choice accuracy is due to the value or the salience of the distractor. This is difficult to test in most reward-based decision making experiments because the value and salience of an option are often collinear – more rewarding options are both larger in value and more salient – and it is not possible to determine which factor drives behavior. One way of breaking the collinearity between value and salience is to introduce options that lead to loss (Kahnt et al., 2014). As such, the smallest value options that lead to great loss are very salient (Figure 6A, bottom), the medium value options that lead to small gains or losses are not salient and the largest value options that lead to great gain are again very salient. Having a combination of gain and loss scenarios in an experiment enables the investigation of whether the positive and negative distractor effects, related to mutual inhibition and divisive normalization respectively, are driven by the distractor’s value, salience or both. Figures 6A and B show 4 hypothetical cases of how the distractor may influence accuracy. Hypothesis 1 suggests that larger distractor *values* (Figure 6A, first row, left-to-right), which correspond to fewer losses or more gains, are related to greater accuracies (brighter colors). This is also predicted by the mutual inhibition component of the dual route model (Figure 1) and can be described as a positive D effect (Figure 6B). Hypothesis 2 larger distractor saliences (Figure 6A, second row, center-to-sides) are related to greater accuracies (brighter colors). This can be described as a positive |D| effect (Figure 6B). Under this hypothesis the mutual inhibition decision making component receives salience, rather than value, as an input. Hypotheses 3 and 4 are the opposites of Hypotheses 1 and 2, and predict negative distractor effects as a result of the divisive normalization component depending on whether the input involves value or salience. Hypothesis 3 predicts a value-based effect in which larger distractor values (Figure 6A third row, left-to-right) are related to poorer accuracies (darker colors). Hypothesis 4 predicts a salience-based effect in which larger distractor saliences (Figure 6A, fourth row, center-to-sides) are related to poorer accuracies (darker colors). It is important to note that these four hypotheses are not necessarily mutually exclusive. The earlier sections have demonstrated that positive and negative distractor effects can co-exist and predominate in different parts of decision space. Value-based and salience-based distractor effects can also be teased apart with a combination of gain and loss scenarios.

To test these hypotheses, we adopted this approach in an additional experiment performed at the same time as Experiment 3 Hong Kong, in which half of the trials included options that were all positive in value (gain trials) and the other half of the trials included options that were all negative in value (loss trials; the loss trials were not analyzed in the previous sections). We therefore refer to these additional trials as belonging to Experiment 7 Loss Experiment (n=40 as in Experiment 3 Hong Kong). The effect of signed D reflects the value of the distractor while the effect of the unsigned, absolute size of D (i.e. |D|) reflects the salience of the distractor. The correlation between these two parameters was low (*r=*0.005), such that it was possible to isolate the impact that they each had on behavior.

As in other experiments, we first plotted the accuracy as a function of difficulty (HV-LV) and relative distractor value (D-HV). For ease of comparison, Figure 3E that illustrates the accuracy data for the gain trials in Experiments 3 is shown again in the right panel of Figure 6C. As described before, when the decisions were hard (bottom rows) larger distractor values were associated with greater accuracies (left-to-right: the colors change from dark to bright; also see Figure 4B) and when the decisions were easy larger distractor values were associated with poorer accuracies (left-to-right: the colors change from bright to dark; also see Figure 4B). In a similar manner, the left panel of Figure 6C shows the accuracy data of the loss trials in Experiment 7. Strikingly, on both hard and easy trials (top and bottom rows), larger distractor values were associated with poorer accuracies (left-to-right: the colours changes from bright to dark).

To isolate the value-based and salience-based effects of D, we performed GLM6a (Methods) to analyze both the gain and loss trials in Experiments 3 and 7 at the same time. GLM6 includes the signed and unsigned value of D (i.e. D and |D| respectively). We also included a binary term, GainTrial, to describe whether the trial presented gain options or loss options and, as in GLM1a, we included the HV-LV term and its interaction with D but now also with |D| [i.e. (HV-LV)D and (HV-LV)|D| respectively]. The results showed a negative effect of value D (*β*=-0.236, *t*_39_=-2.382, *p*=0.022; Figure 6d) and a negative effect of (HV-LV)D interaction (*β*=-0.205, *t*_39_=-2.512, *p*=0.016). In addition, there was a positive effect of salience |D| (*β*=0.152, *t*_39_=3.253, *p*=0.002) and a positive effect of (HV-LV)|D| (*β*=0.219, *t*_39_=3.448, *p*=0.001). Next, we examined closely the value-based and salience-based effect in different parts of decision space.

As in the analysis for Experiments 1-6 in Figure 6E, we split the data (which included both gain and loss trials) according to the median HV-LV, such that the distractor effects can be examined on hard and easy trials separately. We applied GLM6b that first partialled out the effects of HV-LV, HV+LV and GainTrial from the accuracy data and then tested the overall effect of value D across the gain and loss trials. Similar to Experiments 1-6, a positive value D effect was identified on hard trials (*β*=0.008, *t*_39_=2.463, *p*=0.017; Figure 6e, left) and a negative value D effect was identified on easy trials (*β*=-0.011, *t*_38_=-3.807, *p*<10^-3^; note that one participant was excluded due to the lack of variance in the accuracy data). Then we applied GLM6c which was similar to GLM6b but the value D term was replaced by the salience |D| term. The results showed that there were positive salience |D| effects on both hard (*β*=0.011, *t*_39_=2.119, *p*=0.041; Figure 6E, right) and easy trials (*β*=0.009, *t*_38_=2.338, *p*=0.025).

Taken together, in Experiments 1-6 a positive distractor effect predicted by the mutual inhibition model and a negative distractor effect predicted by the divisive normalization model were found on hard and easy trials respectively. The results of Experiments 3 and 7 suggest that these effects are value-based and that the effects are continuous across the gain and loss value space. In addition, however, there was also a positive distractor effect that was salience-based that appeared on both hard and easy trials, suggesting that the effects driven by the mutual inhibition decision making component can be both value-based and salience-based.

5) There is now some history between the authors and Gluth et al. This shows in multiple places in the paper. For the sake of de-escalation, the authors are encouraged to tone down their language. Specific examples include (but are not limited to) subsection “Both divisive normalization of value and positive distractor effects co-exist in data sets from three sites”.

We have attempted to maintain a neutral tone throughout the manuscript. We have attempted to do this while acting in accord with the request made in comment 2 for clarity about the difference between the two different analysis approaches. We have paid particular attention to the paragraphs highlighted by the reviewer but are happy to change other paragraphs too. For example, the paragraphs highlighted by the reviewer have been changed as follows:

Figure 3A and C show the data from the fMRI experiment (Experiment 1 *fMRI2014; n=21*) reported by Chau et al., 2014 and Gluth et al., 2018 experiment 4 (Experiment 2 Gluth4; n=44) respectively. It is important to consider these two experiments first because they employ an identical schedule. Specifically, Chau et al., reported both divisive normalization effects and positive distractor effects, while Gluth et al., claimed they were unable to replicate these effects in their own data and when they analyzed this data set from Chau et al.,. Here we found that both data sets show a positive D-HV distractor effect. In both data sets, when decisions are difficult (HV-LV is small) then high value D-HV is associated with higher relative accuracy in choices between HV and LV; for example, the bottom rows of Figure 3A and c turn from black/dark red to yellow moving from left to right, indicating decisions are more accurate. However, when decisions were easy (HV-LV is large) then the effect is much less prominent or even reverses as would be predicted if divisive normalization becomes more important in this part of the decision space. As in the predictions of the dual route model (Figure 1J,K), on easy trials although there was an overall decreasing trend in accuracy as a function of D-HV, there was an increasing trend at very low D-HV levels. Overall, a combination of positive and negative D-HV effects on hard and easy trials respectively suggests that there should be a negative (HV-LV)(D-HV) interaction effect on choice accuracy. (Subsection “Both divisive normalization of value and positive distractor effects co-exist in data sets from three sites”)

Another possible reason for the discrepancy in the interpretation concerns the other three experiments reported by Gluth et al. We turn to these next. (Subsection “Both divisive normalization of value and positive distractor effects co-exist in data sets from three sites”)

We have tried to make related changes elsewhere. For example the revised manuscript is changed as follows:

“It is clear thatdata collected under the same conditions in 105 participants at all three sites are very similar and that a positive distractor effect consistently recurs when decisions are difficult. Next, we aggregated the data collected from the three sites and repeated the same GLM to confirm that the (HV-LV)(D-HV) interaction (*β*=-0.101, *t*_104_=-4.366, *p*<10^-4^), D-HV (*β*=0.223, *t*_104_=6.400, *p*<10^-8^) and HV-LV (*β*=0.529, *t*_104_=20.775, *p*<10^-38^) effects were all collectively significant. Additional control analyses suggest that these effects were unlikely due to any statistical artefact (see subsection “Distractor effects are not driven by statistical artefact” for details).” (subsection “Both divisive normalization of value and positive distractor effects co-exist in data sets from three sites”).

6) In many places, statistical interactions are not interpreted using "simple effects". When an interaction (e.g., X*Y) is significant it is unclear whether the main effects (e.g. of X) is meaningful or whether the simple effects change sign depending on the other variable (e.g., Y). It would be important to conduct follow-up simple effect analyses. Some of the analyses even contain triple interactions. If these are not interpreted it is difficult to understand what the patterns of results mean.

The reviewer has made a very important point about how to clearly illustrate the pattern of interaction effects. We agree that testing the simple effects is critical and indeed we followed this procedure in the initial submission of our manuscript. For example, GLM1a (formerly GLM1; see below for the GLM) mainly tested any presence of (HV-LV)(D-HV) interaction and that was followed by GLM2a that split the trials into small HV-LV (i.e. hard trials) and large HV-LV (i.e. easy trials), thereby testing for simple main effects, and tested the effects of D-HV (or D) separately (Figure 4 and Figure 4—figure supplement 1). In retrospect, however, we realize that we did not categorically state that in conducting this analysis we were examining the simple main effects. In the revised manuscript we make this point clear. In Figure 3, we also plotted how the accuracy varied as a function of D-HV at different levels of HV-LV as an illustration of the interaction effect. Again, in the revised manuscript, we make clear that this figure illustrates the simple main effects. It is also possible that the link between GLM1a and GLM2a was not very clear, since GLM2a has an additional HV+LV term (this is also suggested in comment 2.2). We have now added GLM1b that included also the HV+LV term to fill this gap and this is described in Subsection “Analysis procedures”:

GLM1a: logit(accuracy) = β_0_ + β_1_(HV-LV) + β_2_(D-HV) + β_3_(HV-LV)(D-HV) + ε

GLM1b: logit(accuracy) = β_0_ + β_1_(HV-LV) + β_2_(HV+LV) + β_3_(D-HV) + β_4_(HV-LV)(D-HV) + ε

GLM2a: Step 1, logit(accuracy) = β_0_ + β_1_(HV-LV) + β_2_(HV+LV) + ε_1_

Step 2, ε_1_ = β_3_ + β_4_(D-HV) + ε_2_

Further, we have made the following revision to improve the link between GLM1a,b and GLM2a (subsection “Both divisive normalization of value and positive distractor effects co-exist in data sets from three sites”):

“The next step is to examine whether the (HV-LV)(D-HV) interaction effect from GLM1a and 1b arises because of the presence of a divisive normalization effect (i.e. negative D-HV effect) on easy trials, a positive distractor effect on hard trials, or both effects. In other words, we need to establish which component of the interaction (or in other words, which main effect) is driving the interaction. To establish which is the case, the data were median split as a function of difficulty, defined as HV-LV, so that it is possible to easily visualize the separate predictions of the divisive normalization and positive distractor accounts (Figure 4A; a similar approach was also used by Chau et al., in their supplementary Figure SI.4). Then, to analyze each half of the choice accuracy data we applied GLM2a in a stepwise manner.”

The legend for Figure 4 now reads as follows:

Figure 4. Distractors had opposite effects on decision accuracy as a function of difficulty in all experiments. The main effect of the distractor was different depending on decision difficulty. (a) In accordance with the predictions of the dual route model, high value distractors (D-HV is high) facilitated decision making when the decision was hard (blue bars), whereas there was a tendency for high value distractors to impair decision making when the decision was easy (red bars). Data are shown for (b) Experiment 1 fMRI2014, Experiment 2 Gluth4, Experiment 3 Hong Kong. (c) The same is true when data from the other experiments, Experiments 4-6 (i.e. Gluth1-3), are examined in a similar way. However, participants made decisions in these experiments in a different manner: they were less likely to integrate probability and magnitude features of the options in the optimal manner when making decisions and instead were more likely to choose on the basis of a weighted sum of the probability and magnitude components of the options. Thus, in Experiments 4-6 (i.e. Gluth1-3), the difficulty of a trial can be better described by the weighted sum of the magnitude and probability components associated with each option rather than the true objective value difference HV-LV. This may be because these experiments included additional “decoy” trials that were particularly difficult and on which it was especially important to consider the individual attributes of the options rather than just their integrated expected value. Whatever the reason for the difference in behavior, once an appropriate difficulty metric is constructed for these participants, the pattern of results is the same as in panel a. # *p*<0.1, * *p*<0.05, ** *p*<0.01, *** *p*<0.001. Error bars indicate standard error.

Perhaps the simple effect analysis for GLM5 is less clear in our original manuscript. GLM5 involves:

GLM5: logit(accuracy) = β_0_ + β_1_(HV-LV) + β_2_(HV+LV) + β_3_D + β_4_(HV-LV)D + β_5_(HV+LV)D + β_6_(HV-LV)(HV+LV) + β_7_(HV-LV)(HV+LV)D + ε

(Subsection “Analysis procedures”)

The critical interaction terms were (HV-LV)D and (HV+LV)D. The terms (HV-LV)(HV+LV) and (HV-LV)(HV+LV)D were regressors of no interest, but were added for the sake of completing a full three-way interaction model. To better describe the pattern of the interaction effects, we have now added the following simple effect analysis in Appendix 3:

“Finally, a follow-up analysis was run to confirm how the D effect varied as a function of HV-LV or HV+LV. These questions related to the (HV-LV)D and (HV+LV)D interaction effects respectively. We applied a mean split by HV-LV and then estimated the effect of D using GLM2b. On hard trials (small HV-LV), larger D values were related to *greater* choice accuracies (*β*=0.056, *t*_207_=4.113, *p*<10^-4^); whereas on easy trials (large HV-LV), larger D values were related to *poorer* choice accuracies (*β*=-0.007, *t*_206_=-2.049, *p*=0.042; note that one participant was excluded from this analysis because of the lack of behavioral variance – there was only one inaccurate choice). On trials with a small HV+LV sum, larger D values were associated with poorer choice accuracies (*β*=-0.009, *t*_207_=-2.530, *p*=0.012); whereas on trial with large HV+LV sums, larger D values were only marginally associated with greater choice accuracies (*β*=0.006, *t*_207_=1.810, *p*=0.072). These results are consistent with our predictions that the mutual inhibition model, associated with a positive D effect, was better at predicting behavior on hard trials and when the HV+LV sum was large. The divisive normalization model, associated with a negative D effect, was better at predicting behavior on easy trials and when the HV+LV sum was small.”

7) The dual route model is attractive as a simple conceptual mechanism for a combination of effects, but there were some questions about the precise implementation, model comparison, and whether the models can account for RT data:

Please refer to our point-by-point replies below.

7.1) As reported in Chau, 2014, distracter input to the mutual inhibition only occurs for a brief period of time (before it is indicated as unchoosable); is the same format used for the divisive normalization model?

We aimed to keep the mutual inhibition model as simple as possible, while only keeping some critical features (e.g. a pooled inhibition) of the biophysical model reported in Chau, 2014. As such, the mutual inhibition model involves distractor input for an equal amount of time to that of the HV and LV options. We have revised subsection “Summary of approach” to clarify this:

“The *mutual inhibition model* is a simplified form of a biophysically plausible model that is reported elsewhere (Chau et al., 2014). It involves three pools of excitatory neurons *P_i_*, each receives noisy input *E_i_* from an option *i* (i.e. HV, LV or D option) at time *t*:

Ei,t∼N(dVi,σ2) where *d* is the drift rate, *V_i_* is the value of option *i* (HV, LV or D) and σ is the standard deviation of the Gaussian noise. The noisy input of all options (HV, LV and D) are all provided simultaneously.”

The divisive normalization does not involve direct distractor input, but instead the distractor influences the evidence accumulation process by normalizing the input of the HV and LV options. We have revised subsection “Computational modelling” to clarify this:

“The *divisive normalization model* follows the same architecture, except that there are only two pools of excitatory neurons, and each receives *normalized* input from the HV or LV option. The D only participates in this model by normalizing the input from the HV and LV options. The normalized input of the HV or LV option follows the following equation: Ei,t∼N(dViVHV+VLV+VD,σ2) where *d* is the drift rate, *V_i_* is the value of option *i* (HV or LV) and σ is the standard deviation of the Gaussian noise. The inhibition *I_t_* and evidence *y_i,t+1_* follow the same equations as the mutual inhibition model.”

7.2) How were relevant model parameters (d and σ) determined in the dual model? It appears that for individual mutual inhibition and normalization models, they were chosen to give 85% correct choices. Is the same thing true for all 4 parameters in the dual model?

It is indeed the case that in the dual route model the four parameters (two ds and two sigmas) were selected in order to give 85% choice accuracy. We have now revised subsection “Computational modelling” such that this is more explicit:

“The levels of d and σ of each model (i.e. the mutual inhibition, divisive normalization or dual route model) were selected in order to produce an overall choice accuracy of 0.85. For the mutual inhibition model, d and σ were set at 1.3 s^-1^ and 1 s^-1^ respectively.”

7.3) It would be more informative to show model predictions based on parameters that were derived from model fits vis a vis empirical data (and show qualitative aspects that the dual route model fits better than the other models) as these parameters are more relevant.

Thank you for the suggestion. One clear way of showing how well each model fits with the empirical data is to plot how much the models’ predictions deviate from the empirical data across the decision space of HV-LV and D-HV. We have now shown in Figure 3—figure supplement 2A,C the empirical data in a format identical to Figure 3A,C,E. We then show in Figure 3—figure supplement 2B,D the degree in which the predictions of each model deviate from the empirical data. The results show that the dual route model is better than the mutual inhibition, divisive normalization, and null models in predicting participants’ accuracy and reaction time. These results are now reported in the Results section and Figure 3—figure supplement 2.

“Interestingly, the dual route model provided the best account of the participants’ behaviour when Experiments 1-3 were considered as a whole (estimated frequency Ef=0.898; exceedance probability Xp=1.000, Figure 3G-H) and when individual experiments were considered separately (Experiment 1: Ef=0.843, Xp=1.000; Experiment 2: Ef=0.924, Xp=1.000; Experiment 3: Ef=0.864, Xp=1.000). Furthermore, the fitted parameters were applied back to each model to predict participants’ behavior (Figure 3—figure supplement 2). The results show that the dual route model is better than the mutual inhibition, divisive normalization, and null models in predicting both choice accuracy and reaction time.” (Subsection “Both divisive normalization of value and positive distractor effects co-exist in data sets from three sites”)

7.4) The authors suggest that the ability of the model to generate both effects is due to the relative speed of each model component in different value conditions. While intuitive, it would be helpful if the authors actually showed this to be the case in the simulation data. Since the two processes act entirely in parallel, it would be simple to perform the simulations for the individual component models (using the dual model parameters) and report average RTs (in the [HV-LV, D-HV] space). In other words, rather than showing solely predictions for accuracy, it would be important to show also predictions for RT. Additionally, in models for RT it is essential to include a "residual, non-decision, time". This doesn't seem to be the case here but should be.

The reviewer has provided very helpful suggestions for better understanding the predictions made by the dual route model. We have now plotted the reaction time of the model when the drift rate and noise level are set at zero for one of the two components at a time. These plots are now shown in Figure 1—figure supplement 2. However, it might not be straightforward to see how the distractor effects of the dual route model are linked to the reaction time of each component because it is necessary to consider not just which model component is likely to produce a response but also how likely it is to be correct. Hence, we suggest that the next analyses, described below, do achieve what we think the reviewer ultimately wants: an intuitive sense of which types of response, both correct and incorrect, that each model causes.

We analyzed the choices predicted by the dual route model when both components are assigned the non-zero parameters. We examined on each trial whether a choice is made by the mutual inhibition or divisive normalization component. Choices made by the mutual inhibition component show greater accuracy as a function of relative distractor value and choices made by the divisive normalization component show smaller accuracy as a function of relative distractor value (Figure 1—figure supplement 1C). These results are broadly similar to those presented in Figure 1A-H in which either the mutual inhibition model or divisive normalization model is run alone. Since the accuracy of the two components are plotted together on the same scale, it is clear, when comparing the slopes, that on hard trials the positive effect of the mutual inhibition component outweighs the negative effect of the divisive normalization component. On easy trials the negative effect of the divisive normalization component outweighs the positive effect of the mutual inhibition component. One key reason for these phenomena is because on easy trials, errors made by the mutual inhibition component are rare and the positive distractor effect on reducing these errors is therefore weaker (Figure 1—figure supplement 1D). We have now presented these analyses in our manuscript:

“In the dual route model positive and negative distractor effects predominate in different parts of the decision space. It is possible to understand the underlying reasons by analyzing the choices made by the mutual inhibition and divisive normalization components separately (Figure 1—figure supplement 1). On hard trials, when the distractor value becomes larger, the errors made by the mutual inhibition component decrease more rapidly in frequency than the increase in errors made by the divisive normalization component, resulting in a net positive distractor effect. In contrast, on easy trials when the distractor value becomes larger the decrease in errors made by the mutual inhibition model is much less than the increase in errors made by the divisive normalization model. Figure 1—figure supplement 2 shows the reaction time of choices made by each component when the other component is switched off.” (Subsection “Divisive normalization of value and positive distractor effects should predominate in different parts of the decision space”)

In addition, we agree with the reviewer that a non-decision time is indeed a very important parameter in the framework of diffusion models. However, because non-decision time has no reason to be different across our models, it would not bring more evidence in favor of one or another model during model comparisons. Nevertheless, we still included a non-decision time [fixed at 300ms Grasman et al., 2009Ratcliff et al., 1999Tuerlinckx, 2004(; ; )] in all the models. This additional procedure is now described in subsection “Computational modelling”.

“The reaction time is defined as the duration of the evidence accumulation before the decision threshold is reached added by a fixed non-decision time of 300 ms to account for lower-order cognitive processes before choice information reaches the excitatory neurons Grasman et al., 2009Ratcliff et al., 1999Tuerlinckx, 2004(; ; ).”

The results of the model comparison remain similar – the dual route model best describes participants behaviour.

“Interestingly, the dual route model provided the best account of the participants’ behaviour when Experiments 1-3 were considered as a whole (estimated frequency Ef=0.898; exceedance probability Xp=1.000, Figure 3G-H) and when individual experiments were considered separately (Experiment 1: Ef=0.843, Xp=1.000; Experiment 2: Ef=0.924, Xp=1.000; Experiment 3: Ef=0.864, Xp=1.000).” (subsection “Both divisive normalization of value and positive distractor effects co-exist in data sets from three sites”).

[Editors' note: further revisions were suggested prior to acceptance, as described below.]

Essential revisions:1) In comment 6, the reviewers previously raised the issue of interpreting triple interactions (e.g., analyses pertaining to GLM5). The authors focus on interpretations of 2-way interactions (HV+LV)D and (HV-LV)D whose signs are of theoretical importance in their framework. The concern is that these interactions are qualified by a significant triple interaction (HV-LV)(HV+LV)D. This means that the sign of each of the 2-way interactions can change as a function of the third variable. Therefore, a simple effect analysis here should examine simple 2 way interactions as a function of the third variable. This analysis is critical for the interpreting the findings.

Thank you for the suggestion. We have now followed these suggestions of testing the three-way (HV-LV)(HV+LV)D by examining how the (HV+LV)D effect or (HV-LV)D effect varied as a function of the third variable. Note, however, that while it is not uncommon to check the interpretation of two-way interactions by examining the simple effects it is less common to see this approach used to examine three way interaction effects. The reason is simple. Examining the simple effects associated with a two way interaction means looking at approximately half of the original data set. Examining the simple effects associated with a three way interaction means looking at approximately a quarter of the original data set which could produce unreliable results. Hence, we took an alternative approach that is recommended in such situations by using the β weights estimated in GLM5 and investigating the three-way interaction effect. In particular, small and large HV-LV (or HV+LV) was defined as the twenty-fifth percentile (*z* score=-0.675) and seventy-fifth percentile (*z* score=0.675) respectively. Then we tested the two-way (HV+LV)D effect at small and large HV-LV:

GLM5:

logit(accuracy) = β_0_ + β_1_ z_(HV-LV)_ + β_2_ z_(HV+LV)_ + β_3_ z_(D)_ + β_4_ z_(HV-LV)_ z_(D)_ + β_5_ z_(HV+LV)_ z_(D)_ + β_6_ z_(HV-LV)_ z_(HV+LV)_ + β_7_ z_(HV-LV)_ z_(HV+LV)_ z_(D)_ + ε

GLM5 at small HV-LV:

logit(accuracy) = β_0_ + β_1_ z_(HV-LV)_ + β_2_ (-0.675) + β_3_ z_(D)_ + β_4_ (-0.675) z_(D)_ + β_5_ z_(HV+LV)_ z_(D)_ + β_6_ (-0.675) z_(HV+LV)_ + β_7_ (-0.675) z_(HV+LV)_ z_(D)_ + ε

GLM5 at large HV-LV:

logit(accuracy) = β_0_ + β_1_ z_(HV-LV)_ + β_2_ (0.675) + β_3_ z_(D)_ + β_4_ (0.675) z_(D)_ + β_5_ z_(HV+LV)_ z_(D)_ + β_6_ (0.675) z_(HV+LV)_ + β_7_ (0.675) z_(HV+LV)_ z_(D)_ + ε

As such, the effect of (HV+LV)D at small HV-LV is (β_5 –_ 0.675β_7_) and that at large HV-LV is (β_5_ + 0.675β_7_). The analysis showed that at small HV-LV there was a positive (HV+LV)D effect (*β*=0.169, *t*_207_=5.656, *p*<10^-7^) and at large HV-LV the (HV+LV)D effect was even more positive (*β*=0.659, *t*_207_=13.999, *p*<10^-31^). A similar procedure was applied to test the (HV-LV)D effect at small or large HV+LV. The results showed that at small HV+LV there was a negative (HV-LV)D effect (*β*=-0.361, *t*_207_=-9.208, *p*<10^-16^), whereas at large HV+LV there was a positive (HV-LV)D effect (*β*=0.128, *t*_207_ = 3.953, *p*<10^-3^).

Finally, we tested the effect of D at different levels of HV-LV and HV+LV. When both HV-LV and HV+LV was small, there was a lack of D effect (*β*=-0.023, *t*_207_=-0.778, *p*=0.437). When HV-LV was small and HV+LV was large, there was a positive D effect (*β*=0.206, *t*_207_=7.119, *p*<10^-10^). When HV-LV was large and HV+LV was small, there was a negative D effect (*β*=-0.510, *t*_207_=-11.414, *p*<10^-22^). When HV-LV was large and HV+LV was large, there was a positive D effect (*β*=0.379, *t*_207_=9.606, *p*<10^-17^).

We have now reported these results in Appendix 3.

Finally, we found that there was a significant (HV-LV)(HV+LV)D effect in GLM5 (*β*=0.362, *t*_207_=10.417, *p*<10^-19^; Appendix 3-figure 2A). Next we examined how the (HV+LV)D effect or (HV-LV)D effect varied as a function of the third variable (HV-LV). One way to examine this is to look at simple effects in sub-sections of the data but because we are now considering a three way interaction, the necessary subsection may be only a quarter in size of the original data, which could produce unreliable results. Hence, we took an alternative approach by using the β weights estimated in GLM5 and investigating the three-way interaction effect. In particular, small and large HV-LV (or HV+LV) was defined as the twenty-fifth percentile (*z* score=-0.675) and 75^th^ percentile (*z* score=0.675) respectively. Then we tested the two-way (HV+LV)D effect at small and large HV-LV:

GLM5:

logit(accuracy) = β_0_ + β_1_ z_(HV-LV)_ + β_2_ z_(HV+LV)_ + β_3_ z_(D)_ + β_4_ z_(HV-LV)_ z_(D)_ + β_5_ z_(HV+LV)_ z_(D)_ + β_6_ z_(HV-LV)_ z_(HV+LV)_ + β_7_ z_(HV-LV)_ z_(HV+LV)_ z_(D)_ + ε

GLM5 at small HV-LV:

logit(accuracy) = β_0_ + β_1_ z_(HV-LV)_ + β_2_ (-0.675) + β_3_ z_(D)_ + β_4_ (-0.675) z_(D)_ + β_5_ z_(HV+LV)_ z_(D)_ + β_6_ (-0.675) z_(HV+LV)_ + β_7_ (-0.675) z_(HV+LV)_ z_(D)_ + ε

GLM5 at large HV-LV:

logit(accuracy) = β_0_ + β_1_ z_(HV-LV)_ + β_2_ (0.675) + β_3_ z_(D)_ + β_4_ (0.675) z_(D)_ + β_5_ z_(HV+LV)_ z_(D)_ + β_6_ (0.675) z_(HV+LV)_ + β_7_ (0.675) z_(HV+LV)_ z_(D)_ + ε

As such, the effect of (HV+LV)D at small HV-LV is (β_5 –_ 0.675β_7_) and that at large HV-LV is (β_5_ + 0.675β_7_). The analysis showed that at small HV-LV there was a positive (HV+LV)D effect (*β*=0.169, *t*_207_=5.656, *p*<10^-7^) and at large HV-LV the (HV+LV)D effect was even more positive (*β*=0.659, *t*_207_=13.999, *p*<10^-31^). A similar procedure was applied to test the (HV-LV)D effect at small or large HV+LV. The results showed that at small HV+LV there was a negative (HV-LV)D effect (*β*=-0.361, *t*_207_=-9.208, *p*<10^-16^), whereas at large HV+LV there was a positive (HV-LV)D effect (*β*=0.128, *t*_207_ = 3.953, *p*<10^-3^).

We tested the effect of D at different levels of HV-LV and HV+LV. When both HV-LV and HV+LV was small, there was a lack of D effect (*β*=-0.023, *t*_207_=-0.778, *p*=0.437). When HV-LV was small and HV+LV was large, there was a positive D effect (*β*=0.206, *t*_207_=7.119, *p*<10^-10^). When HV-LV was large and HV+LV was small, there was a negative D effect (*β*=-0.510, *t*_207_=-11.414, *p*<10^-22^). When HV-LV was large and HV+LV was large, there was a positive D effect (*β*=0.379, *t*_207_=9.606, *p*<10^-17^).

2) Some questions remain about the modelling.2.1) In comment 7.4. Reviewers previously raised the importance of modeling residual time. In response the authors included RT but arbitrarily fixed it to 300ms rather that allowing it to vary freely. They argue in their response letter that "because non-decision time has no reason to be different across our models, it would not bring more evidence in favor of one or another model during model comparisons.". It may be impossible to determine this a-priori because in each model residual-RT might trade-off differently with the other parameters. It would be important to re-fit the model with free residual time parameters to see which model is best. It is important to rule out that the results of model-comparison are due to arbitrary assumptions about residual rt.

Thank you for the suggestion. We have now allowed the non-decision time (Tnd) and inhibition level f (in response to comment 2.2) to vary freely. We have also applied these models to fit the data of Experiments 4-6, in addition to those of Experiments 1-3 that were fitted before (in response to comment 2.4). Critically, the results of the model comparison remain largely similar – in all six experiments the dual route model was a better fit compared to the other three alternative models. In addition, we have run another comparison between all models that we performed – the four new models (with free Tnd and free inhibition f level), the four old mdels (with Tnd fixed at 0.3 sec and f fixed at 0.5) and another four models (with Tnd fixed at 0.3 sec and free f level). The results show that the dual route model with fixed Tnd and f provide the best fit. To summarize, among the four models (dual route, mutual inhibition, divisive normalization, and null) the dual route model provides the best account of participants’ behaviour and it is especially the case when the Tnd and f parameters are fixed.

We have now included these additional models in the Results section:

Additional models were run to confirm that the dual route model is a better model. The above models involve assigning fixed values for the non-decision time *Tnd* (at 0.3 s) and inhibition level *f*. In one set of analysis the *f* is fitted as a free parameter (Figure 3—figure supplement 3B) and in another set of analysis both *Tnd* and *f* are fitted as free parameters (Figure 3—figure supplement 3C). In both cases, as in the models with fixed *Tnd* and *f*, the dual route model is a better fit compared to the other three alternative models (Ef=0.641, Xp=1.000 and Ef=0.587, Xp=1.000 respectively). Finally, a comparison of all twelve models (four models × three versions of free parameter set) shows that the dual route model with fixed *Tnd* and *f* is the best fit (Ef=0.413, Xp=1.000; Figure 3—figure supplement 3D).

2.2) A very similar issues pertains to the parameter f (inhibition) which was also fixed to a constant value rather than being a free parameter. This could potentially affect model comparison results.

We have now followed these useful suggestions. Please refer to the responses to comment 2.1 for details.

2.3) It is still unclear whether in fitting the dual-channel model, each channel had its own free parameters or whether they were identical for both channels.

Thank you for the comment. We agree that it was unclear whether the two components of the dual route model involved separate sets of free parameters or not and in fact they are separated. We have now clarified this in subsection “Model fitting and comparison”:

***“***The d and σ parameters of each model were fitted separately at the individual level to the choices and RTs of each participant in Experiments 1-6, using the VBA-toolbox (http://mbb-team.github.io/VBA-toolbox/). In the dual route model, the mutual inhibition and divisive normalization components involved separate d and σ parameters during the fitting. The other model parameters (i.e. f, Tnd, V_i_ and k) were fixed at the values mentioned above, except for some models reported in Figure 3—figure supplement 3.”

2.4) Why are models comparisons reported only for Experiments 1-3 but not for the other experiments?

We have presented model comparisons for Experiments 1-3 because the data from these experiments are relatively straightforward and therefore there is little debate about how they might be modelled. It is difficult to apply these models to the data of Experiments 7-8 because of differences in tasks were designed (see responses to Comments 3.1 and 3.2). It is, however, also possible to apply these models to fit the data of Experiments 4-6 because they involve tasks that are very similar to those in Experiments 1-3. The results showed that among the four models, as in Experiments 1-3, the dual route model provided the best account of participants’ behaviour in Experiments 4-6. These results are now included in the Results section and Figure 4—figure supplement 2:

“Finally, the four models (dual-route, mutual inhibition, divisive normalization and null) were applied to fit the data of Experiments 4-6. Again, the dual-route model provided the best account of participants’ behaviour when individual experiments were considered separately (Experiment 4: Ef=0.806, Xp=0.999; Experiment 5: Ef=0.649, Xp=1.000; Experiment 6: Ef=0.946, Xp=1.000; Figure 4—figure supplement 2) or when Experiments 1-6 were considered as a whole (Ef=0.846, Xp=1.000). ” (subsection “Examining distractor effects in further experiments”).

3) In comment 4 the reviewers previously raised questions about the loss trials. The revised version does not fully address these questions.3.1) There are still questions pertaining to how to model loss trials. According to the current equations, it seems that drift rates can be negative for the mutual inhibition model, and in the normalization model, the drift will be strongest for the highest loss option. Clearly, if this is correct, the model will require adjustments to account for loss trials and these have to be explained.

It is indeed an important to ask about how best to model loss trials in Experiment 7. We agree with the reviewer that one possibility is to have a negative drift rate in the models that are applied to Experiments 1-3. This also implies that the decisions should then be about which option to avoid rather than which option to choose, because options that are more negative in value will reach the decision threshold more quickly. It is unclear whether participants used an avoidance approach in Experiment 7. Hence, we consider that in Experiment 7 (loss experiment) the focus should be on participants’ empirical behavior and the fact that the experiments constitutes a test of whether the distractor affected choices by its value or salience. We think that more detailed modelling is an interesting question but because it requires a new series of arguments about whether it is appropriate to think of avoiding or “ruling out” options we think that it would detract from the discussion of all the other points, which is comprehensive. If, however, we were to add a section relating to whether model loss trials with negative drift rates, which is a whole topic unto itself then we would inevitably deal with this issue in a way that some readers might consider partial. We have, however, added some text to the revised manuscript that notes that it might be possible in the future to extend the model to look at this loss trials but that we have refrained from a detailed modelling of loss trials because of the number of other issues that need to be debated whenever a model of this type is constructed. We have therefore added the following paragraph to the end of subsection “Experiment 7: Loss experiment” reporting the results of Experiment 7:

In the future it might be possible to extend the models outlined in Figure 1 and Figure 1—figure supplement 1 to provide a more quantitative descriptions of behavior. While this topic is of great interest it will require modelers to agree on how loss trials might be modelled. For example, one possibility is to have a negative drift rate in the models that we have used. This implies that the decisions will then be about which option to avoid rather than which option to choose, because options that are more negative in value will reach the decision threshold more quickly. It is unclear, however, whether participants used such an avoidance approach in Experiment 7. Hence, we have refrained from modelling the results of Experiment 7.

3.2) There are also questions pertaining to whether and how the model can account for differences between gain and loss trials. These are important issues because the results seem quite different for gain and loss trials. It would be important to perform a model comparison for the loss trial to determine the best model for these trials. It is not clear if the dual route model, or one of the simpler models, is best for loss trials. Additionally- looking at Figure 6 and the results of the regression (panel d) it seems that when D is positive (i.e., D = abs(D)) corresponding regression effects for these two terms offset each other but when D is negative (D = -abs(D)) they compound. So this could simply mean that the distractor effects are stronger for losses than for gains. Is this true? This can be seen, by including in the regression, interactions terms with trial identity (what the authors call GainTrial) instead of terms with abs(D). Furthermore, if distractors effects are indeed stronger for losses then these stronger effects could presumably be caused by adjusting model parameters (e.g., inhibition strength or other parameters). It is important to examine this. In sum, the authors should consider fitting models to loss trials to see (1) which model provides the best account for loss trials, (2) what account do the mechanistic models provide loss trials and for differences between gain and loss trials. This will provide a much more informative understanding of the gain-loss issue as compared to the current reliance on the regression model. Currently the authors argue that there are 2 separate effects in play, one for distractor value and one for distractor saliency. But a more informative way to understand the data might be that the context (gain/loss) modulated the distractor value effect, and to query the mechanistic models to identify the locus of this modulations.

Again we thank the reviewer for this interesting point but once again we feel that detailed modelling of the loss trials is beyond the scope of the current manuscript because of the number of other issues that need to be dealt with in order to consider how loss trials might be modelled with precision. We have dealt with the major points that were raised by the reviewers in the first round of review and we have moved large parts of the manuscript to the supplementary materials to make way for the many new sections that the reviewers have requested but detailed modelling of the loss trials really is beyond the scope of the current manuscript because it would require us to address several other, distinct areas of research. We feel that this is likely to detract from the main message our manuscript is conveying.

4) Reviewers were still unclear about the meaning of terms in the GLMs. This needs to be clarified so that the models are better understood and evaluated. Just for example consider GLM1:logit(accuracy) = β0 + β1(HV-LV) + β2(D-HV) + β3(HV-LV)(D-HV) + εThe authors state that "All interaction terms in all GLMs are calculated after the component terms are z-scored". Does this mean that the terms are z-scored only for the purpose of calculating the interaction, or are they z-scored for the main effects as well? Reviewers think they should be z-scored in all terms not just in interaction terms. Additionally, just to be sure- did the authors z-score HV and LV separately or z-score the difference (HV-LV)? A clearer way to write the model to avoid confusions could be:logit(accuracy) = β0 + β1 z_(HV-LV) + β2z_(D-HV) + β3z_(HV-LV)*z_(D-HV) + ε (underscore indicates subscript).

We agree that there is still some potential ambiguity of how the GLMs are described. In this study, all terms are z-scored before entering into a GLM, which may not be clear in the previous version of the manuscript. We have now followed the format suggested by the reviewer and updated our Materials and methods section (only the lines showing the GLMs are shown here).

GLM1a: logit(accuracy) = β_0_ + β_1_ z_(HV-LV)_ + β_2_ z_(D-HV)_ + β_3_ z_(HV-LV)_ z_(D-HV)_ + ε

GLM1b: logit(accuracy) = β_0_ + β_1_ z_(HV-LV)_ + β_2_ z_(HV+LV)_ + β_3_ z_(D-HV)_ + β_4_ z_(HV-LV)_ z_(D-HV)_ + ε

GLM1c: ln(P_HV_/P_j_) = β_j,0_ + β_j,1_ z_(HV-LV)_ + β_j,2_ z_(D-HV)_ + β_j,3_ z_(HV-LV)_ z_(D-HV)_ + ε_j_

GLM2a: Step 1, logit(accuracy) = β_0_ + β_1_ z_(HV-LV)_ + β_2_ z_(HV+LV)_ + ε_1_

Step 2, ε_1_ = β_3_ + β_4_ z_(D-HV)_ + ε_2_

GLM2b: Step 1, logit(accuracy) = β_0_ + β_1_ z_(HV-LV)_ + β_2_ z_(HV+LV)_ + ε_1_

Step 2, ε_1_ = β_3_ + β_4_ z_(D)_ + ε_2_

GLM2c: Step 1, logit(accuracy) = β_0_ + β_1_ z_(Difficulty)_ + ε_1_

Step 2, ε_1_ = β_2_ + β_3_ z_(D-HV)_ + ε_2_

GLM2d: Step 1, logit(accuracy) = β_0_ + β_1_ z_(Difficulty)_ + ε_1_

Step 2, ε_1_ = β_2_ + β_3_ z_(D)_ + ε_2_

Where in GLM2c,d Difficulty = w_1_ z_[Mag(HV)-Mag(LV))_ + w_2_ z_[Mag(HV)+Mag(LV))_ +

w_3_ z_[Prob(HV)-Prob(LV)]_ + w_4_ z_[Prob(HV)+Prob(LV)]_

GLM3a: logit(accuracy) = β_0_ + β_1_ z_(HV-LV)_ + β_2_ z_(HV+LV)_ + ε

GLM3b: logit(accuracy) = β_0_ + β_1_ z_[Mag(HV)-Mag(LV))_ + β _2_ z_[Mag(HV)+Mag(LV)]_ + β _3_ z_[Prob(HV)-Prob(LV)]_ +

β _4_ z_[Prob(HV)+Prob(LV)]_ + ε

GLM4: logit(accuracy) = β_0_ + β_1_ z_(SubjDiff)_ + β_2_ z_(Congruence)_ + β_3_ z_(D-HV)_ +

β_4_ z_(SubjDiff)_ z_(D-HV)_ + ε

GLM5: logit(accuracy) = β_0_ + β_1_ z_(HV-LV)_ + β_2_ z_(HV+LV)_ + β_3_ z_(D)_ + β_4_ z_(HV-LV)_ z_(D)_ + β_5_ z_(HV+LV)_ z_(D)_ + β_6_ z_(HV-LV)_ z_(HV+LV)_ + β_7_ z_(HV-LV)_ z_(HV+LV)_ z_(D)_ + ε

GLM6a: logit(accuracy) = β_0_ + β_1_ z_(GainTrial)_ + β_2_ z_(HV-LV)_ + β_3_ z_(D)_ + β_4_ z_(HV-LV)_ z_(D)_ + β_5_ z_(|D|)_ +

β_5_ z_(HV-LV)_ z_(|D|)_ + ε

GLM6b: Step 1, logit(accuracy) = β_0_ + β_1_ z_(GainTrial)_ + β_2_ z_(HV-LV)_ + β_3_ z_(HV+LV)_ + ε_1_

Step 2, ε_1_ = β_4_ + β_5_ z_(D)_ + ε_2_

GLM6c: Step 1, logit(accuracy) = β_0_ + β_1_ z_(GainTrial)_ + β_2_ z_(HV-LV)_ + β_3_ z_(HV+LV)_ + ε_1_

Step 2, ε_1_ = β_4_ + β_5_ z_(|D|)_ + ε_2_

GLM7: Fix_j_ = β_j,0_ + β_j,1_ z_(HV-LV)_ + β_j,2_ z_(HV+LV)_ + β_j,3_ z_(D)_ + ε_j_

GLM8: Step 1, Shift_j_ = β_j,0_ + β_j,1_ z_[Fix(HV)]_ + β_j,2_ z_[Fix(LV)]_ + β_j,3_ z_[Fix(D)]_ + ε_j,1_

Step 2, ε_j,1_ = β_j,4_ + β_j,5_ z_(HV)_ + β_j,6_ z_(LV)_ + β_j,7_ z_(D)_ + ε_j,2_

GLM9: Step 1, logit(accuracy) = β_0_ + β_1_ z_(HV-LV)_ + β_2_ z_(HV+LV)_ + β_3_ z_(D)_ + ε_1_

Step 2, ε_1_ = β_4_ + β_5_ z_[Shift(D-to-HV)]_ + β_6_ z_[Shift(D-to-LV)]_ + β_7_ z_[Shift(HV-to-D)]_ + β_8_ z_[Shift(LV-to-D)]_ +

β_9_ z_[Shift(LV-to-HV)]_ + β_10_ z_[Shift(HV-to-LV)]_ + ε_2_